# Simultaneous inhibition of bacterial virulence and anti-phage defense systems by synergistic bacteriophage counter-defense proteins

Jingru Zhao[1,2,4], Yuhao Zhu [iD][3,4], Chenchen Wang [iD][2], Fan Tian[2], Jun Deng[3], Jianglin Liao[3], Zhuojun Zhong[3], Jiazhen Liu[3], Nannan Guo[1], Shuai Le [iD][3✉] & Haihua Liang [iD][2✉]

## Abstract

**Bacteriophages have evolved diverse inhibitors targeting key bacterial processes, including virulence and anti-phage defense systems, which could inspire novel antimicrobial strategies and enhance phage therapy approaches. In this study, we characterize Dap2, a protein encoded by a *Pseudomonas aeruginosa* phage PaoP5, which disrupts host virulence by sequestering the type III secretion system (T3SS) transcriptional activator ExsA, thus suppressing bacterial pathogenicity. Furthermore, Dap2 also directly binds the host Lon protease to prevent degradation of the phage-encoded HNH endonuclease. Deletion of *dap2* in PaoP5 strongly impairs phage genome packaging due to insufficient levels of HNH. Finally, Dap2 synergizes with its genomically adjacent partner Dap1, a previously identified HNH-binding protein providing partial Lon resistance, to completely protect HNH against degradation. Together, these findings reveal a dual-function phage protein that simultaneously modulates bacterial virulence and anti-phage immunity, and showcase a synergistic mechanism for complete neutralization of bacterial defense system against which individual components provide only partial protection.**

**Keywords** Bacteriophage Protein Dap2; Anti-phage Immunity; T3SS; Lon Protease; *Pseudomonas aeruginosa*
**Subject Category** Microbiology, Virology & Host Pathogen Interaction

## Introduction

*Pseudomonas aeruginosa* is an opportunistic pathogen capable of causing severe nosocomial infections, and it represents a major cause of morbidity and mortality in patients with cystic fibrosis (Meirelles et al, 2024; Rossi et al, 2021). The pathogenicity of *P. aeruginosa* is coordinated by a range of virulence factors, among

which the type III secretion system (T3SS) serves as a crucial contact-dependent apparatus for injecting effector proteins and plays an essential role in acute infections (Dey et al, 2019; Song et al, 2023). The expression of this highly energy-consuming system is tightly regulated by the transcriptional activator ExsA (Vakulskas et al, 2009). With the increasing prevalence of multidrug-resistant strains, phage therapy is a potential alternative to antibiotics (Li et al, 2023; Li et al, 2025; Pirnay et al, 2024; Yang et al, 2025). A deeper understanding of the molecular mechanisms underlying phage infection and bacterial resistance is expected to provide a theoretical foundation for designing more rational phage-based therapeutic strategies (Strathdee et al, 2023).

Bacteriophages rely on bacterial hosts for replication, driving an ongoing coevolutionary arms race between phage and bacteria (Correa et al, 2021; De Smet et al, 2017; Dion et al, 2020). Thus, phages have developed diverse inhibitors that disrupt critical bacterial processes, such as virulence and anti-phage defense mechanisms (Zhang et al, 2022). These insights may pave the way for innovative antimicrobial therapies and advance phage-based treatments (Wan et al, 2021).

Bacteriophages hijack their bacterial hosts for reproduction by producing early-stage proteins that reprogram key cellular processes, including transcription, DNA replication, translation, and cell division (Roucourt and Lavigne, 2009). Additionally, phages can modulate bacterial virulence. For example, a phage protein PIT2 inhibits the type II secretion system (T2SS) by directly binding LasR (Schroven et al, 2023). These phage-bacterial interactions provide a valuable model for developing small-molecule or peptide-based inhibitors against diverse pathogens (Liu et al, 2004; Zhang et al, 2022).

However, whether phages can regulate the type III secretion system (T3SS) remains unclear. T3SSs are critical virulence determinants in many pathogenic bacteria, enabling them to inject effector proteins directly into host cells and establish trans-kingdom interactions (Deng et al, 2017). Given their essential role in infection, T3SSs are attractive targets for novel therapeutics and vaccines (He et al, 2025). Unlike conventional antibiotics, T3SS inhibitors may reduce selective pressure for resistance since they

[1]College of Life Sciences, Northwest University, Xi'an, ShaanXi, China. [2]Department of Biochemistry, SUSTech Homeostatic Medicine Institute, School of Medicine, Southern University of Science and Technology, Shenzhen 518055 Guangdong, China. [3]Department of Microbiology, College of Basic Medical Sciences, Key Laboratory of Microbial Engineering Under the Educational Committee in Chongqing, Army Medical University, Chongqing, China. [4]These authors contributed equally: Jingru Zhao, Yuhao Zhu. ✉E-mail: leshuai2004@tmmu.edu.cn; lianghh@sustech.edu.cn

disrupt virulence rather than bacterial growth. A deeper structural and functional understanding of T3SSs and their phage protein inhibitors will advance mechanism-based drug development (He et al, 2025; Rasko and Sperandio, 2010).

On the other hand, bacteria have evolved diverse anti-phage defense mechanisms, such as CRISPR-Cas, restriction-modification (RM) systems, CBASS, Pycsar, Gabija, and Thoeris (Bae et al, 2025; Georjon and Bernheim, 2023; LeRoux et al, 2022; Tesson et al, 2024; Vassallo et al, 2022). These systems typically belong to the accessory genome, reflecting their "acquired" rather than "intrinsic" genomic origin. However, the distinction between defense systems becomes more complex when intrinsic mechanisms are considered (Bae et al, 2025). For instance, *Bacillus subtilis* employs SigX, an extracytoplasmic function sigma factor that is transiently upregulated during phage infection. SigX confers resistance by inhibiting phage adsorption to its secondary receptor (Tzipilevich et al, 2022). Unlike acquired systems, SigX is encoded in the core genome of *B. subtilis* (Huang et al, 1997*)*, classifying it as an intrinsic defense mechanism. Similarly, our group previously demonstrated that Lon protease, a housekeeping protein involved in essential bacterial processes (Breidenstein et al, 2012), acts as an intrinsic anti-phage defense factor (Le et al, 2024). Specifically, Lon degrades the HNH endonuclease of phage PaoP5Δ*dap1*, reducing genome packaging efficiency and significantly decreasing progeny phage release (Le et al, 2024). This highlights how core cellular components can also function in phage defense.

In response, phages have evolved sophisticated anti-defense systems (ADSs) to counteract these immune strategies. While numerous ADSs targeting CRISPR-Cas (Bondy-Denomy et al, 2013; Meeske et al, 2020) and RM systems (Karambelkar et al, 2020; Studier and Movva, 1976) have been well characterized, recent studies have uncovered novel phage-encoded countermeasures against CBASS, Pycsar, Gabija, and Thoeris pathways (Hobbs et al, 2022; Mayo-Munoz et al, 2024; Murtazalieva et al, 2024).

Characterized ADSs employ a variety of molecular strategies to neutralize host immunity, such as direct inhibition of defense proteins through binding, enzymatic modification or deactivation of immune components, degradation or sequestration of signaling molecules, metabolic compensation for host-induced depletion of essential molecules, and structural camouflage of phage components to evade detection (Murtazalieva et al, 2024; Niault et al, 2025). To date, approximately 150 distinct ADSs have been functionally characterized. However, the vast diversity of bacterial defense mechanisms necessitates ongoing discovery and mechanistic analysis of ADSs (Duan et al, 2024). Notably, some ADSs confer only partial resistance to bacterial defenses, as illustrated by the phage protein Acb2 (Huiting et al, 2023): its induction only partially rescues phage titers in JBD67Δ*acb2* and JBD18 variants. This suggests that phages may employ multiple counterstrategies against individual immune systems to enhance protection (Murtazalieva et al, 2024).

Here, we establish a systematic framework for identifying virulence-modulating hypothetical proteins in *P. aeruginosa* phage PaoP5 (Shen et al, 2016), with a particular focus on regulators of the type 3 secretion system (T3SS) (Deng et al, 2017; Yip and Strynadka, 2006). We identified the phage protein Orf004, which directly binds to ExsA (Brutinel et al, 2008), the master regulator of T3SS, suppressing T3SS expression and bacterial pathogenicity. Intriguingly, Orf004 also interacts with the bacterial Lon protease

to prevent degradation of the phage-encoded HNH endonuclease. Deletion of *orf004* results in impaired genome packaging and reduced progeny viability. Our findings complement the previously characterized Dap1 protein (defense anti-phage protein 1), which stabilizes HNH through direct binding (Le et al, 2024). Since this ORF (orf004) is located adjacent to the *dap1* gene (Le et al, 2024), we named it Dap2 to maintain a consistent nomenclature based on genomic context. The coordinated action of Dap1 and Dap2 establishes a dual protection mechanism against Lon-mediated antiviral defense: Dap1 physically shields HNH, while Dap2 inhibits protease activity. This synergistic system ensures complete protection of the essential HNH endonuclease from host degradation.

This study highlights the dual functionality of a phage protein in simultaneously regulating bacterial virulence and countering host immunity. Furthermore, it reveals that phages can deploy synergistic anti-defense proteins (Dap1/Dap2) with distinct mechanisms to neutralize single bacterial defense systems, expanding our understanding of phage evolutionary strategies in the ongoing arms race with their hosts.

## Results

### Identification of a phage protein Dap2 that inhibits T3SS

To systematically identify phage-encoded inhibitors of *Pseudomonas aeruginosa* type III secretion system (T3SS), a critical virulence apparatus for host cell invasion (Song et al, 2023), we developed a dual-plasmid fluorescence reporter assay (Fig. 1A). The transcriptional activity of T3SS was monitored by fusing the *exsC* promoter (encoding a core T3SS structural protein) to *luxCDABE* biosensor genes in plasmid pMS402 (Duan et al, 2003). Concurrently, 51 PaoP5 ORFs were individually expressed using the IPTG-inducible vector pME6032, none of which impacted bacterial growth (Tables EV1–EV2). Upon co-transformation of both plasmids into PAO1, luminescence attenuation served as a proxy for T3SS inhibition. Interestingly, IPTG-induced expression of ORF004 (herein designated *Dap2*) demonstrated dose-dependent suppression of luminescence (Fig. EV1A,B), suggesting specific targeting of T3SS regulatory components. To further validate the inhibitory effect of Dap2 on T3SS functionality, we analyzed the promoter activity of key T3SS effector genes (*exoS*, *exoY*, and *exoT*) in WT PAO1/pME6032 versus PAO1 harboring pME6032-*dap2*. The results demonstrated that Dap2 overexpression significantly suppressed promoter activity across all tested genes (Fig. EV1C). Consistent with these transcriptional changes, immunoblot analysis revealed a marked reduction in ExoS protein levels and ExsA (the master T3SS regulator) expression in the *dap2*-overexpressing strain compared to the vector control (Fig. 1B). These findings collectively establish Dap2 as a potent T3SS suppressor and warrant detailed investigation into its molecular mechanism of action.

To gain insight into how Dap2 affects the T3SS, we performed RNA sequencing (RNA-seq) comparing the transcriptional profiles of PAO1/pME6032-*dap2* (*dap2*-overexpressing strain) and PAO1 carrying the empty vector during exponential growth. Overexpression of *dap2* altered the expression of 229 genes (|fold change| > 2, P < 0.05), with 93 upregulated and 136 downregulated genes (Fig. 1C). Strikingly, KEGG pathway enrichment analysis

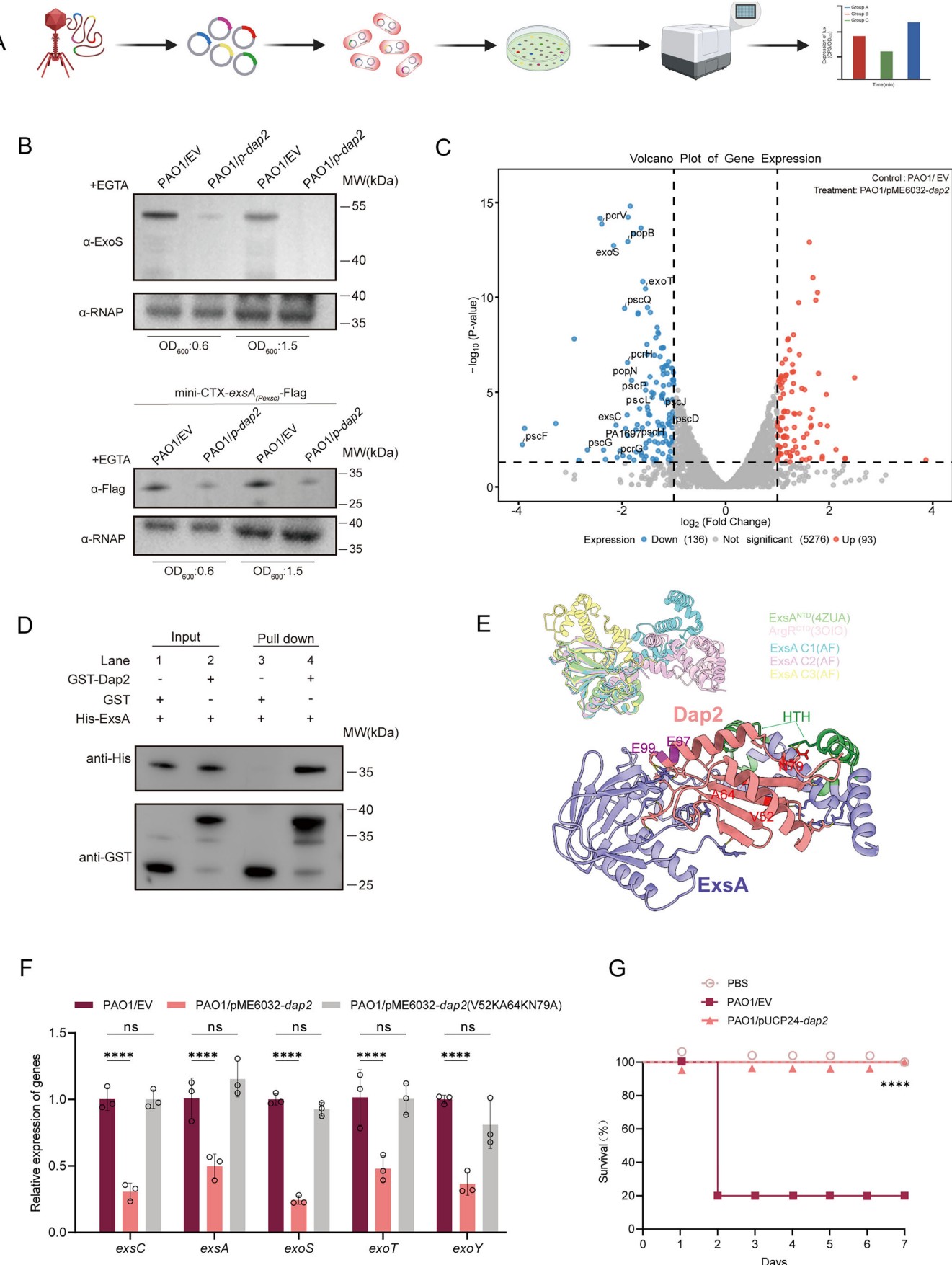

**Figure 1. Phage protein Dap2 inhibits bacterial virulence via interacting with T3SS regulator ExsA.**

(A) Screening strategy for phage-encoded T3SS inhibitors. Schematic of the dual-plasmid screening system in *P. aeruginosa* PAO1. Early-expressed PaoP5 phage genes were cloned into IPTG-inducible plasmid pME6032. A T3SS activity reporter plasmid was constructed by fusing the *exsC* promoter to *luxCDABE* biosensor genes. Subsequently, the luminescence of each clone was measured using the microplate reader. (B) Western blot analysis of ExoS and ExsA protein expression in wild-type and Dap2-overexpressing PAO1 strains. Bacterial cultures were induced with 5 mM EGTA and 20 mM $MgCl_2$, with samples harvested at optical densities ($OD_{600}$) of 0.6 and 1.5. Protein levels were assessed using anti-ExoS and anti-Flag antibodies for ExoS and ExsA detection, respectively. α-RNA polymerase (RNAP) served as the loading control. (C) Volcano plot analysis of differentially expressed genes (DEGs) was performed using RNA-seq data comparing PAO1/pME6032-*dap2* and PAO1/pME6032 strains. Blue and red data points indicate genes with significant upregulation and downregulation, respectively, in PAO1/pEM6032-*dap2* relative to the PAO1/pME6032 control group. Three biological repeats were performed and enrichment significance was calculated using a two-sided Fisher's exact test, with *p* values adjusted for multiple comparisons via the Benjamini-Hochberg method (FDR < 0.05). (D) Direct interaction between GST-Dap2 and ExsA-His$_6$ was confirmed by GST pull-down assay. Purified ExsA-His protein was incubated with either GST-Dap2 or GST (control) protein, and the resulting protein complexes were captured using GST-binding beads. The initial samples (input) and retained proteins (pull-down) were then analyzed by Western blot against GST or His antibody. (E) ExsA-Dap2 complex structure model and the details about interaction interface. Electrostatic potential surfaces of ExsA and Dap2 are basically complementary (bottom right). Different conformations of ExsA were marked with C1 to C3 (upper left). Amino acid residues that, upon directed mutagenesis, affect ExsA-DnaP binding are labeled in red; those affecting ExsA-Lon binding are labeled in magenta. HTH domain referring to DNA recognition and binding was colored in green. (F) Expression of *dap2* triple mutant (pME6032-*dap2*$^{V52K/A64K/N79A}$) failed to inhibit T3SS-related genes. qRT-PCR analysis of T3SS gene expression in the indicated strains. Error bars represent the mean ± SD of three independent experiments. Two-way ANOVA was used to calculate *p* value (PAO1/EV vs PAO1/-*dap2*: *exsC*, *P* < 0.0001; *exsA*, *P* < 0.0001; *exoS*, *P* < 0.0001; *exoT*, *P* < 0.0001; *exoY*, *P* < 0.0001; PAO1/EV vs PAO1/pME6032-*dap2*$^{V52KA64KN79A}$: *exsC*, *P* > 0.9999; *exsA*, *P* = 0.1943; *exoS*, *P* = 0.6203; *exoT*, *P* = 0.9896; *exoY*, *P* = 0.0728). EV, empty vector. ns, no significance, ****$P$ < 0.0001. (G) The expression of *dap2* significantly attenuated the virulence of *P. aeruginosa*. In the infection model, 7-week-old BABL/c female mice (*n* = 10) were intraperitoneally injected with either PAO1/pUCP24 (vector control) or PAO1/pUCP24-dap2 at a dose of 2 × 10$^7$ CFU in 100 μL of phosphate-buffered saline (PBS). Survival data were recorded. Statistical analysis was performed using the log-rank test. (PAO1/EV vs PAO1/pUCP24-*dap2*: *P* < 0.0001). ****$P$ < 0.0001. Source data are available online for this figure.

identified the type III secretion system as the most significantly affected pathway, with multiple T3SS-associated genes clustering in this category (Fig. EV2A,B; Dataset EV1). To confirm the RNA-seq data reliability, we selected four randomly chosen genes (*hemH*, *fhp*, *pcrv*, *pscF*) and five T3SS-related genes for qRT-PCR validation. The qRT-PCR results exhibited strong concordance with RNA-seq trends, showing consistent upregulation or downregulation patterns in response to *dap2* overexpression (Fig. EV2C).

## Dap2 inhibits T3SS via interacting with ExsA

Given that ExsA serves as the master transcriptional regulator of T3SS, and our transcriptomic data revealed global downregulation of T3SS genes by Dap2 overexpression, we posited that Dap2 might directly target ExsA. To test this hypothesis, we first employed the BacterioMatch II Two-Hybrid System to probe Dap2-ExsA interactions. Recombinant strains expressing bait (pBT-Dap2) and prey (pTRG-ExsA) constructs were plated on nonselective versus dual-selective media containing 3-amino-1,2,4-triazole (3-AT) and streptomycin. While the positive control (LGF$_2$/GaI$^{11}$P interaction) showed robust growth under selection, specific colony survival was observed exclusively in the Dap2/ExsA co-expressing strain on selective media, demonstrating direct protein interaction (Fig. EV3A). This physical interaction was further corroborated by affinity pull-down assays, where His-tagged ExsA was specifically pulled down by GST-tagged Dap2, but not by GST alone (Fig. 1D). As anticipated, expression of *dap2* failed to repress *exsC* promoter activity in the Δ*exsA* mutant strain (Fig. EV3B). Collectively, these results establish that Dap2-mediated suppression of T3SS occurs through direct binding to ExsA, thereby inhibiting its regulatory function.

To analyze the potential interaction mode and identify key residues responsible for maintaining the ExsA-Dap2 interaction interface, molecular docking was conducted using Alphafold3 (Abramson et al, 2024). Due to the existence of a hinge composed of the $^{166'}$Ser-Asn-Arg$^{168'}$ triad, ExsA exhibits several conformations across all candidates. And it's inferred that the relative motion

ability of CTD (HTH domain) might contribute to the DNA binding. However, the domains located at each terminus are highly conserved compared to the reported ExsA$^{NTD}$ or its homolog ArgR$^{CTD29}$ (Fig. 1E). Therefore, different conformations of ExsA were docked with Dap2, whose structure was relatively conserved throughout all top candidates. The top ExsA-Dap2 model features an interface area of 2187.5 Å², accompanied by a solvation free energy gain of −13.7 kcal/mol, indicating a relatively hydrophobic interface or positive protein affinity as indicated by PDBePISA analysis. Besides 17 hydrogen bonds and 3 salt bridges, over 400 non-bonded contacts can be observed at the interface, suggesting that hydrophobic van der Waals forces might play a crucial role in maintaining the interface (Fig. 1E, bottom right). To validate the predicted interaction model, we generated site-directed mutants targeting key residues in Dap2 (V52K, A64K, N79A) and assessed their impact on T3SS gene promoter activity. Unlike wild-type Dap2, the triple mutant *dap2* (V52K/A64K/N79A) failed to suppress T3SS genes (Fig. 1F). These collective findings support a model where Dap2 interacts with ExsA to constrain conformational flexibility in its DNA-binding domain, thereby inhibiting ExsA-mediated transcriptional activation.

To further determine whether T3SS genes are suppressed during phage infection, we infected PAO1 with either phage PaoP5 or the PaoP5Δ*dap2* mutant and extracted total RNA at time points between 1 and 30 min post-infection. Quantitative analysis revealed that expression of key T3SS genes (*exsC*, *exoS*) was significantly downregulated in PAO1 infected with wild-type PaoP5, but not in the PaoP5Δ*dap2*-infected group (Fig. EV3C,D). These results confirm that T3SS suppression occurs during phage infection and is dependent on Dap2.

Given the critical role of the T3SS in virulence (Brutinel et al, 2008; Yip and Strynadka, 2006), we investigated the impact of the phage-encoded Dap2 protein on the pathogenicity of *P. aeruginosa* PAO1 using a well-established acute-infection mouse model (Le et al, 2024). The *dap2* gene was cloned into the plasmid pUCP24 to enable constitutive expression of Dap2. The virulence of strains carrying either PAO1/pUCP24 (control) or PAO1/pUCP24-*dap2*

was then evaluated. Female BALB/c mice were intraperitoneally challenged with these strains, and survival was monitored. Strikingly, 80% of mice infected with PAO1/pUCP24 died within 2 days, whereas all mice infected with PAO1/pUCP24-*dap2* survived through day 7 (Fig. 1G). Each experimental group included ten mice, and statistical analysis was performed using the log-rank test. These findings demonstrate that the expression of Dap2 significantly attenuates the pathogenicity of *P. aeruginosa*.

## Knockout of *dap2* significantly impairs the fitness of phage

Our findings highlight the significant influence of the phage-encoded Dap2 protein on *P. aeruginosa* T3SS. To further explore the potential role of Dap2 in enhancing phage fitness, we investigated its transcriptional dynamics during infection. Specifically, *P. aeruginosa* strain PAO1 was infected with phage PaoP5, and samples were collected at 1-, 10-, 15-, and 30-min post-infection. RNA was extracted from these samples, and the expression levels of *dap2* transcripts were quantified using qRT-PCR. The results revealed that *dap2* transcription peaked at 10 min post-infection, while the mRNA level of structural gene *orf053* was highest at 30 min (Fig. EV4A). This temporal expression pattern indicates that *dap2* is an early-expressed gene, likely to play a role in the initial stages of phage infection.

Next, we employed the CRISPR-Cas9 system to delete the *dap2* gene in phage PaoP5, with the knockout another phage gene (*orf014*) which does not affect the plaque formation as a negative control (Fig. 2A). Notably, PaoP5Δ*dap2* produced significantly smaller plaques compared to the wild-type (WT) phage, while PaoP5Δ*orf014* formed plaques similar in size to the WT (Fig. 2A). Furthermore, the expression of *dap2* in *P. aeruginosa* PAO1 restored the formation of large plaques for PaoP5Δ*dap2*, whereas the efficiency of plating (EOP) of PaoP5Δ*dap2* in PAO1/p-dap2 remained comparable to that in PAO1/EV (Fig. 2B). These results demonstrate that *dap2* is important for phage plaque formation, whereas *orf014* does not appear to contribute to phage fitness under the tested laboratory conditions.

To investigate the role of *dap2* in the phage life cycle, we first assessed its impact on the initial adsorption rate, as phage infection begins with host recognition and binding. Ten minutes after mixing phage and bacteria, the adsorption rate of PaoP5Δ*dap2* and the wild-type (WT) phage are $93.23 \pm 0.74\%$ and $94.80 \pm 0.57\%$, respectively, exhibited no statistically significant difference ($P > 0.05$), indicating that *dap2* is not involved in host recognition or binding (Fig. 2C). We then examined the burst size of the two phages, as plaque size is closely linked to the number of progeny produced. The one-step growth curve revealed that PaoP5Δ*dap2* generated significantly fewer progeny compared to the WT phage (Fig. 2D). Specifically, the burst sizes of PaoP5 and PaoP5Δ*dap2* were approximately $100.87 \pm 19.76$ PFU/cell and $10.73 \pm 3.34$ PFU/cell, respectively (Fig. 2E). This indicates that PaoP5Δ*dap2* produced only about 10.63% of the progeny generated by the WT phage.

To determine whether the reduced productivity was linked to the bacterial T3SS, we infected PAO1Δ*exsA* with PaoP5Δ*dap2*. However, the phenotype of small plaque and reduced burst size persisted (Fig. EV4C,D), suggesting that the fitness advantage conferred by Dap2 extends beyond its interaction with ExsA.

## Dap2 inhibits the Lon protease-mediated degradation of the phage HNH protein

To identify the potential anti-defense target of Dap2, we employed a pull-down assay coupled with liquid chromatography-mass spectrometry (LC-MS) to screen for Dap2-interacting proteins. This analysis identified Lon as a binding partner of Dap2 (Figs. 3A and EV5A,B, Dataset EV2). The interaction between Dap2 and Lon was further validated using a bacterial adenylate cyclase two-hybrid (BACTH) assay (Fig. 3B), which confirmed their specific binding. Notably, Dap2 did not interact with Dap1 or HNH under the same conditions. The Dap2-Lon interaction was additionally confirmed through a pull-down assay, reinforcing the specificity of this binding (Fig. 3C).

Previously, we demonstrated that Lon protease directly degrades the phage-encoded HNH endonuclease (Le et al, 2024), a critical component of the phage DNA packaging machinery required for the specific endonuclease activity of large terminase proteins (Kala et al, 2014). Degradation of HNH disrupts phage genome packaging, suggesting that Dap2 may bind to Lon to inhibit its function. To test this hypothesis, we infected wild-type PAO1, Δ*lon*, Δ*lon/p-lon*, and *hnh*-overexpression strain PAO1/*p-hnh* with PaoP5Δ*dap2*. The results showed that PaoP5Δ*dap2* formed larger plaques in Δ*lon* or PAO1/*p-hnh* compared to PAO1 (Fig. 3D,E), while plaque size reverted to smaller dimensions in Δ*lon/p-lon*, supporting the role of Lon in HNH degradation.

Since Lon-mediated degradation of HNH leads to the formation of empty capsids devoid of packaged phage genomes, we examined phage lysates from different strains using transmission electron microscopy (TEM) (Fig. 3F). Negatively stained electron micrographs revealed a significant number of empty phage particles in PaoP5Δ*dap2* lysates, indicative of defective DNA packaging. In contrast, deletion of *lon* or overexpression of *hnh* in PAO1 restored DNA packaging efficiency to levels comparable to the wild-type phage (Figs. 3F and EV6).

Quantification of genome-packaged capsids from three biological replicates showed that only $20.00\% \pm 3.26\%$ of capsids contained genomes in PaoP5Δ*dap2* lysates. However, complementation with *dap2*, overexpression of *hnh*, or deletion of *lon* increased this percentage to $70.67\% \pm 2.49\%$, $69.33\% \pm 0.94\%$, and $81.33\% \pm 0.94\%$, respectively. Conversely, the reintroduction of *lon* into Δ*lon* decreased the percentage of genome-packaged capsids to $20.00 \pm 4.32\%$ (Fig. 3G). These findings demonstrate that Dap2 protects HNH endonuclease from rapid degradation by Lon, thereby ensuring efficient phage genome packaging and maintaining the productivity of phage progeny.

## Dap2 inhibits ExsA and Lon through two distinct domains

To investigate the molecular mechanisms by which Dap2 interacts with ExsA and Lon, we first predicted the structure of Dap2 and ExsA using AlphaFold3. Structures of NTD and CTD of ExsA could be well superimposed with its crystal structure (3OIO) and homolog (4ZUA), respectively, with the overall structure showing several possible conformations (indicated with C1, C2 and C3) (Fig. 1E). The prediction reveals that the inhibition of ExsA and Lon protease is mediated by separate domains of Dap2. Residues critical for ExsA inhibition (yellow; positions 52, 64, and 79) are spatially separated from those involved in Lon protease inhibition gray; positions 97 and 99), with these functional sites located on opposite surfaces of the protein (Fig. 4A).

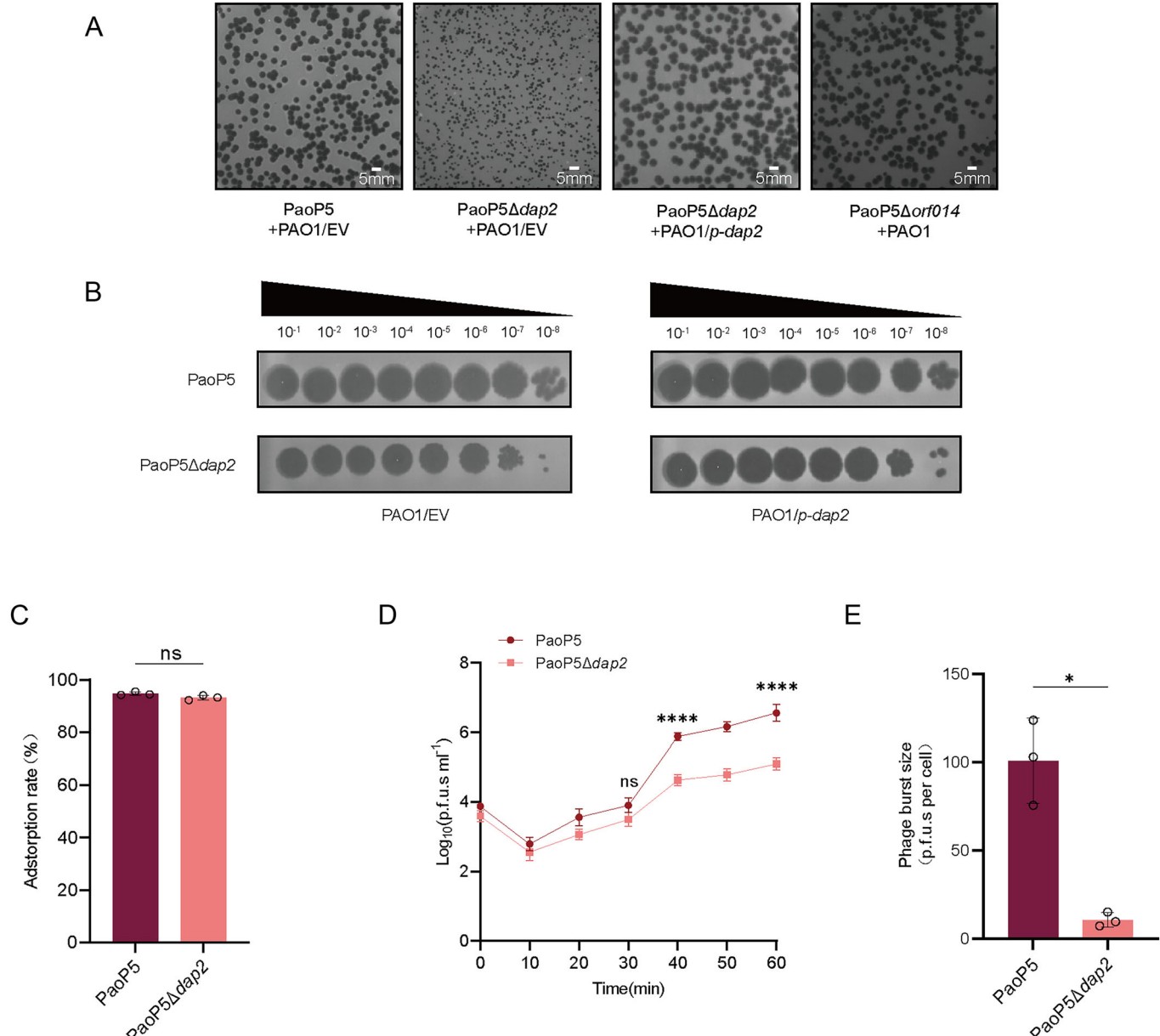

**Figure 2. PaoP5Δ*dap2* forms small plaques and produces fewer progenies.**

(A) PaoP5Δ*dap2* produced significantly smaller phage plaques compared to the wild-type PaoP5, while PaoP5Δ*orf014*, used as a control, exhibited plaque sizes similar to the wild-type. Complementation with Dap2 restored the large plaque phenotype, indicating the functional role of Dap2 in plaque formation. (B) Plaque assays were performed using 10-fold serial dilutions to compare the plating efficiency of wild-type (WT) and mutant phages on PAO1 and PAO1 harboring the *p-dap2* vector. The results demonstrate the impact of Dap2 on phage plaque size but do not affect EOP. (C) Both PaoP5Δ*dap2* and wild-type PaoP5 phages adsorbed to PAO1 with comparable efficiency. Error bars represent the mean ± SD of three independent experiments. Statistical significance was determined using Student's t-test (PaoP5 vs PaoP5Δ*dap2*: $P = 0.1491$). ns, not significant. (D) The one-step growth curve revealed that PaoP5Δ*dap2* produced significantly fewer progeny phages compared to the wild-type PaoP5, highlighting the role of Dap2 in phage replication efficiency. Error bars represent the mean ± SD of three independent experiments. Significance at different time points was assessed by two-way ANOVA (PaoP5 vs PaoP5Δ*dap2*: 30 min, $P = 0.0782$; 40 min, $P < 0.0001$; 60 min, $P < 0.0001$). ns, not significant, ****$P < 0.0001$. (E) The burst size of PaoP5Δ*dap2* and PaoP5 are 10.73 ± 3.34 p.f.u.s per cell and 100.87 ± 19.76 p.f.u.s per cell, respectively. Error bars represent the mean ± SD of three independent experiments. Statistical significance was determined using a two-sided Student's t-test (PaoP5 vs PaoP5Δ*dap2*: $P = 0.0031$). *$P < 0.05$. Source data are available online for this figure.

We then constructed five point mutations in Dap2. In plaque assays, PaoP5Δ*dap2* formed large plaques on PAO1 expressing Dap2(V52K), Dap2(A64K), or Dap2(N79A), but small plaques on strains expressing Dap2(E97A) or Dap2(E99A). This indicates that the V52K, A64K, and N79A mutants retain the ability to inhibit Lon, whereas the E97A and E99A mutants lose this function, suggesting that E97 and E99 residues are essential for Lon inhibition (Fig. 4B,C). This conclusion was further supported by BACTH assays, in which E97A and E99A mutant Dap2 failed to bind Lon (Fig. 4D).

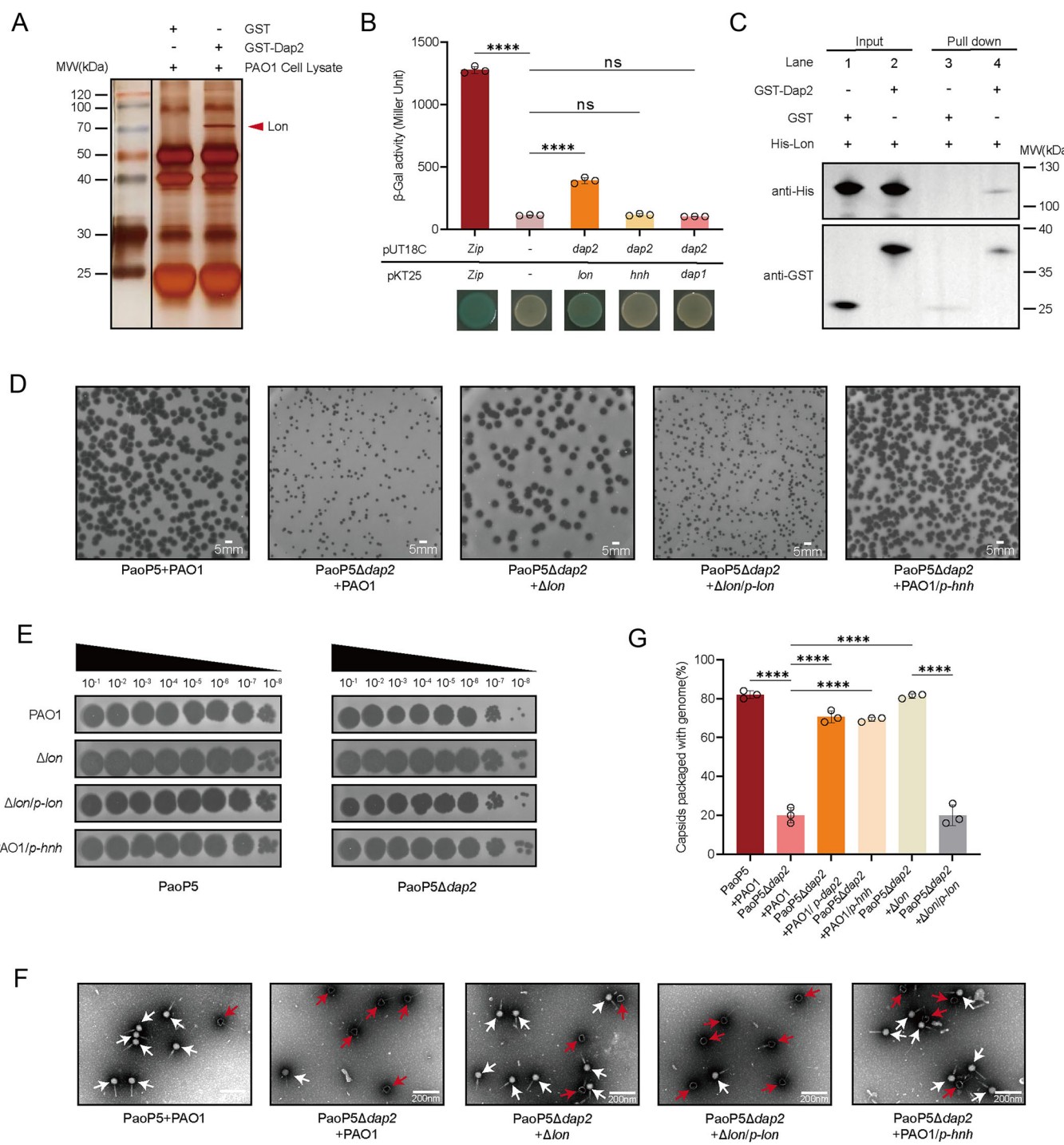

To assess whether Lon inhibition contributes to the reduced virulence of PAO1::Dap2 (Zhou et al, 2018), we infected mice with PAO1::Dap2(E97A) or PAO1::Dap2(E99A). All infected mice survived, indicating that the attenuation of T3SS-dependent virulence is mediated through ExsA inhibition, not through Lon (Fig. 4E).

Given ExsA's role as a transcriptional activator of T3SS through promoter binding, we further performed EMSA assays to evaluate

the effect of Dap2 mutants on ExsA DNA-binding activity. EMSA revealed strong ExsA-dependent DNA binding, evidenced by a retarded band corresponding to ExsA–DNA complexes (Fig. 4F). The N79A, V52K, and A64K mutants lost the ability to inhibit ExsA, allowing ExsA to bind DNA (Fig. 4F,G), whereas the E97A and E99A mutants still suppressed ExsA binding (Fig. 4H). These results confirm that N79, V52, and A64—but not E97 or E99—are essential for inhibiting ExsA.

**Figure 3. Dap2 binds to Lon protease to inhibit its function.**

(A) Identification of Lon as the binding target of Dap2. Proteins specifically retained by GST-Dap2 from cell lysates of *P. aeruginosa* PAO1 with a silver-stained SDS–PAGE gel. The identity of the retained protein (red arrow) was identified through mass spectrometry. (B) The bacterial two-hybrid assay revealed that Dap2 specifically interacts with Lon protease, but not with HNH or Dap1. These interactions were visualized through a drop test on LB agar plates supplemented with x-gal, where the formation of blue colonies signified positive interactions. The interactions were further quantified by measuring β-galactosidase activity. Error bars represent the mean ± SD of three independent experiments. Statistical significance was determined using one-way ANOVA followed by Dunnett's multiple comparison test versus the control group(C-) (C- vs Zip+Zip, $P < 0.0001$; C- vs Dap2+Lon, $P < 0.0001$; C- vs Dap2+HNH, $P = 0.7872$; C- vs Dap2+Dap1, $P = 0.7016$). ns, not significant, ****$P < 0.0001$. (C) Direct interaction between GST-Dap2 and Lon-His$_6$ was confirmed by GST pull-down assay. Purified Lon-His protein was incubated with either GST-Dap2 or GST (control) protein, and the resulting protein complexes were captured using GST-binding beads. The initial samples (input) and retained proteins (pull-down) were then analyzed by Western blot against GST or His antibody. (D) Plaque formation and EOP (E) was observed for PaoP5 infecting PAO1, as well as PaoP5Δ*dap2* infecting PAO1/*p-dap2*, Δ*lon*, Δ*lon*/*p-lon*, and PAO1/*p-hnh*. (F) Transmission electron microscopy (TEM) images of negatively stained phages produced in PAO1, Δ*lon*, Δ*lon*/*p-lon*, and PAO1/*p-hnh* are shown. Red arrows indicate empty capsids, while white arrows point to phages with packaged genomes. (G) The percentage of capsids containing genomes was calculated from three biological replicates. For each condition, 50 particles were counted to determine the presence or absence of genomes. Error bars represent the mean ± SD of three independent experiments. Statistical significance was determined using one-way ANOVA with Dunnett's multiple comparison test (PaoP5Δ*dap2* + PAO1 vs PaoP5 + PAO1, $P < 0.0001$; PaoP5Δ*dap2* + PAO1 vs PaoP5Δ*dap2* + PAO1/*p-dap2*, $P < 0.0001$; PaoP5Δ*dap2* + PAO1 vs PaoP5Δ*dap2* + PAO1/*p-hnh*, $P < 0.0001$; PaoP5Δ*dap2* + PAO1 vs PaoP5Δ*dap2* + Δ*lon*, $P < 0.0001$; PaoP5Δ*dap2*+Δ*lon* vs PaoP5Δ*dap2*+ PAO1Δ*lon*/*p-lon*, $P < 0.0001$). EV, empty vector. ****$P < 0.0001$. Source data are available online for this figure.

## Dap2 and Dap1 cooperate to evade the Lon protease-mediated anti-phage defense

Building on our prior finding that Dap1 sterically shields the HNH endonuclease from Lon protease-mediated degradation (Le et al, 2024), we investigated potential functional synergy between Dap1 and Dap2. The genomic colocalization of these adjacent genes within a conserved operon, coupled with Dap2's direct Lon-binding capacity, prompted us to propose a cooperative defense mechanism ensuring full protection of HNH.

To confirm this hypothesis, we first generated a double knockout of *dap1* and *dap2*. The resulting phage, PaoP5Δ*dap1*-Δ*dap2*, formed tiny, blurred plaques, significantly smaller than those produced by PaoP5Δ*dap1* or PaoP5Δ*dap2* alone (Fig. 5A). TEM analysis revealed that ~90% of the capsids were empty (Fig. 5B), indicating severe defects in genome packaging. Complementation of HNH or deletion of Lon in PAO1 could restore the large plaque size (Fig. EV7A) and increase the genome packaging efficiency (Fig.EV7B,E). While adsorption kinetics remained unaffected in the double mutant (Fig. EV7C), the one-step growth curve demonstrated a markedly reduced burst size for PaoP5Δ*dap1*Δ*dap2* (Fig. 5C). The burst size decreased to $6.86 \pm 1.53$ PFU/cell, only ~6.46% of that observed for the wild-type phage (Fig. EV7D), highlighting the synergistic role of Dap1 and Dap2 in countering Lon-mediated defense.

To further validate this synergy, we performed complementation assays. Individual *dap1* or *dap2* expression partially rescued plaque morphology, genome packaging, and burst size, whereas co-expression fully restored wild-type parameters (Fig. 5A–D). We also conducted in vitro protein degradation assays using purified Lon, HNH, ATP, and an ATP regeneration system (creatine phosphate and creatine phosphokinase) (Herbst et al, 2009). HNH was completely degraded in the presence of Lon and the kinase system but remained stable in the absence of either creatine phosphokinase or Lon (Fig. 5E). Intriguingly, Lon was unable to degrade Dap2, and Dap2 alone provided partial protection against HNH degradation (Fig. 5F). To determine whether Dap2-mediated inhibition is substrate-specific or broadly impairs Lon activity, we also incubated Lon with RhlI in the presence or absence of Dap2 (Yang et al, 2015). We found that Dap2 could inhibit Lon-mediated degradation of RhlI (Fig. EV8). These results demonstrate that

Dap2 exerts a broad inhibitory effect on Lon, rather than being specific to the phage protein HNH.

Moreover, we previously showed that Dap1 also offers partial protection to HNH (Le et al, 2024). Remarkably, when Dap1 and Dap2 were combined, they completely inhibited LON-mediated degradation of HNH (Fig. 5F). These in vitro and in vivo findings demonstrate that Dap1 and Dap2 form a synergistic anti-defense system (ADS) pair, employing distinct mechanisms to neutralize Lon protease: Dap1 shields the target (HNH), while Dap2 directly inhibits the defense protein (Lon). Individually, each protein provides only partial protection, but together, they achieve complete protection, highlighting the evolutionary advantage of this dual-defense strategy.

## Dap2/Dap1 pair is co-localized in phage genomes and cooperates to enhance the efficacy of phage therapy

To explore the prevalence of the *dap1*/*dap2* gene pair, we conducted a homology search in the NCBI database among *P. aeruginosa* phages. As of January 2025, 67 sequenced *Pseudomonas* phages were found to carry *dap2* homologs with a minimal identity of 80%, which were clustered into three distinct clades based on phylogenetic analysis (Fig. 6A).

Interestingly, *dap1* and *dap2* genes consistently co-occur, forming a tandem overlapping gene pair separated by *a* base pair (Fig. 6B). This gene pair was identified in 67 phage genomes (Dataset EV3), suggesting a conserved genetic arrangement. This widespread colocalization indicates that *dap1* and *dap2* function as a synergistic anti-defense system (ADS) pair, both essential for countering the Lon protease, a ubiquitous housekeeping protein in *P. aeruginosa*. The evolutionary conservation of this gene pair underscores its critical role in enhancing phage fitness and efficacy during infection.

Given that phage PaoP5Δ*dap2* produces fewer progeny and PaoP5Δ*dap1*Δ*dap2* generates even fewer, we assessed their therapeutic potential using a mouse infection model (Le et al, 2024). Seven-week-old female BALB/c mice were intraperitoneally injected with either 50 μL of PBS (uninfected control) or 50 μL of *P. aeruginosa* PAO1 ($6 \times 10^7$ CFU/mL). This was followed by administration of 50 μL of phage PaoP5Δ*dap1*Δ*dap2*, PaoP5Δ*dap2*, or wild-type PaoP5 at a multiplicity of infection (MOI) of 10. Wild-

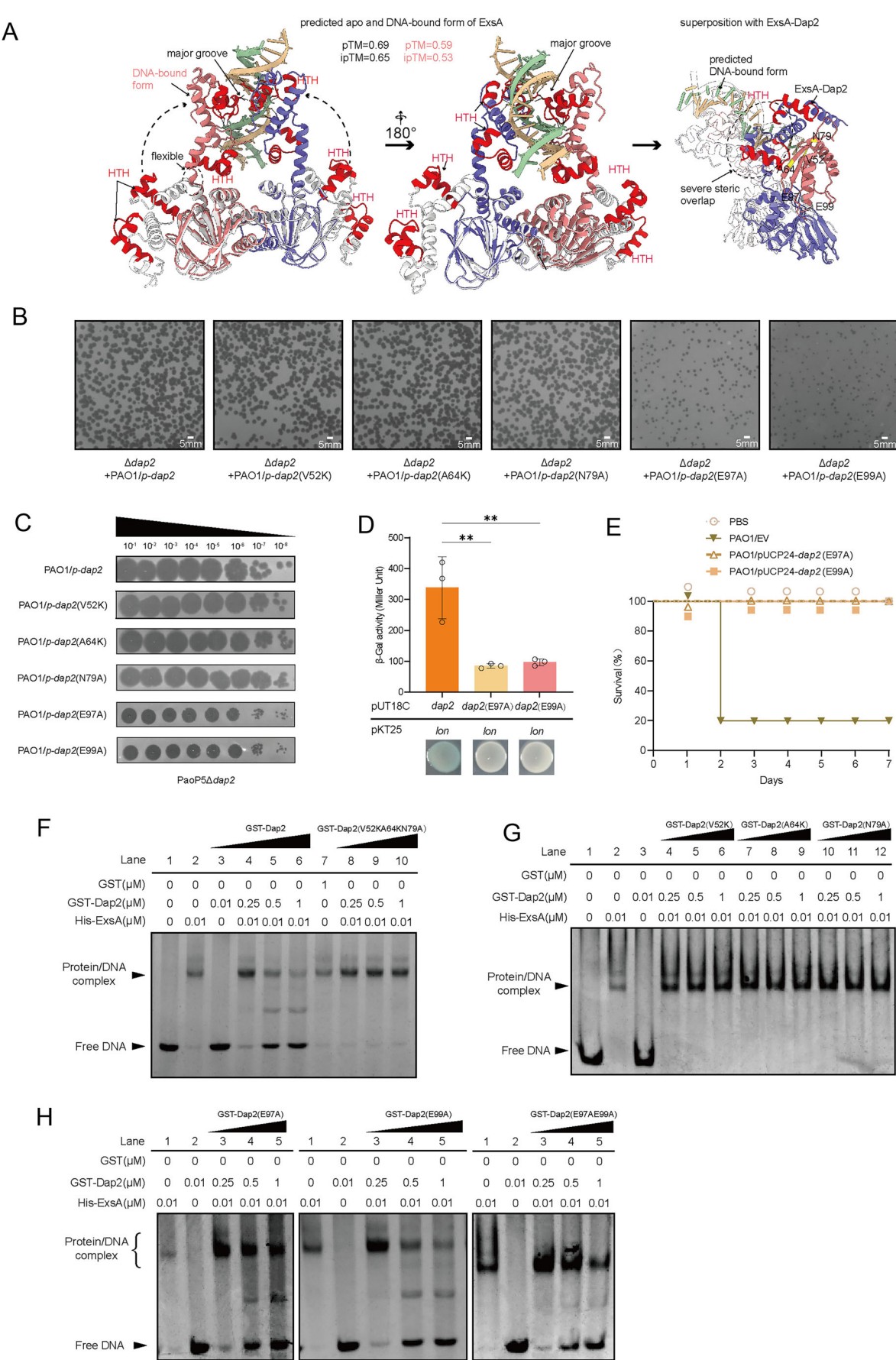

**Figure 4.   Dap2 inhibits ExsA and Lon through two distinct domains.**

(A) Proposed a possible mechanism by which Dap2 inhibits the DNA-binding ability of ExsA, based on predicted protein structures. (B) PaoP5Δ*dap2* exhibited larger plaque morphology in PAO1::*dap2*. This phenotype remained consistent upon complementation with the point-mutant plasmids of dap2 (V52K), (A64K), and (N79A). In contrast, complementation with E97A or E99A mutant Dap2 resulted in smaller plaque sizes. (C) The EOP assay of PaoP5Δ*dap2* was compared on PAO1 overexpressing *dap2* or its point mutants. (D) Bacterial two-hybrid assays demonstrated that Dap2 interacts with the Lon protease; however, the point mutants Dap2(E97A) and Dap2(E99A) lost this interaction. Error bars represent the mean ± SD of three independent experiments. Statistical analysis was performed using one-way ANOVA with Dunnett's multiple comparison test versus the control group (Dap2+Lon vs Dap2$^{E97A}$+Lon, $P = 0.0037$; Dap2+Lon vs Dap2$^{E99A}$ +Lon, $P = 0.0037$). **$P < 0.01$. (E) The *dap2* (E97A) and *dap2*(E99A) mutants also significantly attenuated the virulence of *P. aeruginosa*, indicating that these mutations do not impair Dap2's role in virulence regulation. In the infection model, 7-week-old female BALB/c mice ($n = 10$) were intraperitoneally injected with PAO1 carrying pUCP24 (vector control), pUCP24-*dap2*, pUCP24-*dap2* (E97A), or pUCP24-*dap2* (E99A) at a dose of $8 \times 10^6$ CFU. Significance was determined by the log-rank test (PAO1/EV vs PAO1/pUCP24-*dap2* $^{E97A}$: $P < 0.0001$; PAO1/EV vs PAO1/pUCP24-*dap2*$^{E99A}$: $P < 0.0001$). EV, empty vector. ****$P < 0.0001$. (F–H) EMSA results revealed that Dap2 affects the DNA-binding ability of ExsA. A mixture of 10 ng DNA and His-ExsA was incubated with increasing concentrations of GST-Dap2 or its mutant variants. Data are representative of three independent experiments. The single mutants GST-Dap2 (V52K), (A64K), (N79A), and the triple mutant GST-Dap2 (V52K/A64K/N79A) did not inhibit ExsA–DNA binding, whereas GST-Dap2(E97A), GST-Dap2 (E99A) and the double mutant GST-Dap2(E97A E99A) still disrupted this interaction. Source data are available online for this figure.

type PaoP5 rescued 100% of *P. aeruginosa*-infected mice (Fig. 6C). In contrast, all mice treated with PaoP5Δ*dap2* succumbed to infection within 6 days, while those treated with PaoP5Δ*dap1*Δ*dap2* died even more rapidly, with all mice perishing within 4 days (Fig. 6C).

To further evaluate the therapeutic efficacy of PaoP5Δ*dap2* and PaoP5Δ*dap1*Δ*dap2*, we measured phage titers and bacterial CFUs in the liver post-treatment. As shown in Fig. 5D, significantly higher phage titers were detected in the livers of PaoP5-treated, PAO1-infected mice. The peak titer of PaoP5 reached ~$3.8 \times 10^7$ PFU/g at 8 h post-injection, gradually declining thereafter and clearing within 96 h (Fig. 6D). Concurrently, bacterial CFUs in the liver decreased steadily and were fully cleared by 96 h, resulting in 100% survival of PaoP5-treated mice (Fig. 6E). In contrast, for PaoP5Δ*dap2*-treated, PAO1-infected mice, phage titers continuously decreased, becoming undetectable by 72 h, with a maximum titer of only ~$3.95 \times 10^6$ PFU/g-significantly lower than that observed in PaoP5-treated mice (Fig. 6D). Similarly, PaoP5Δ*dap1*Δ*dap2*-treated, PAO1-infected mice exhibited even lower phage titers during the therapy. Bacterial CFUs in the liver continued to rise post-infection, ultimately leading to the death of all PaoP5Δ*dap1*Δ*dap2*- and PaoP5Δ*dap2*-treated mice. These results demonstrate that PaoP5Δ*dap1*Δ*dap2* produces even fewer progeny, significantly impairing its therapeutic efficacy. Collectively, these findings highlight the critical role of both *dap1* and *dap2* in the effectiveness of phage therapy.

## Discussion

The long-term evolutionary arms race between bacteriophages and their bacterial hosts has driven the diversification of phage-encoded anti-virulence proteins and anti-defense systems (Murtazalieva et al, 2024). We set up a system to identify phage protein that could inhibit T3SS, and identified Dap2 as a direct binder of ExsA, the master regulator of *P. aeruginosa's* T3SS, leading to T3SS suppression and attenuated pathogenicity in a murine acute-infection model (Fig. 1G). The molecular interaction between Dap2 and ExsA provides a mechanistic blueprint for developing small-molecule inhibitors targeting this interface, offering a promising anti-virulence therapeutic strategy (He et al, 2025; Muhlen and Dersch, 2016).

While Dap2 plays a clear biological role in countering Lon protease-mediated anti-phage immunity in lytic phages (Fig. 3), the

biological significance of the interaction between Dap2 and ExsA remains unclear. To explore the biological benefit of T3SS inhibition, we compared phage progeny production in PAO1 and Δ*exsA* strains. And Δ*exsA* did not rescue the small plaque phenotype of PaoP5Δ*dap2*, because larger plaques require a high yield of progeny per generation (Miller et al, 2003), subtle differences in burst size may not be discernible through plaque morphology alone. Burst size analysis demonstrated that PaoP5Δ*dap2* produced ~1.94-fold more progeny when infecting Δ*exsA* mutants compared to wild-type PAO1 (Fig. EV4D). The modest increase in viral yield implies that inhibiting the T3SS may represent a specific phage strategy to conserve host resources, thereby enhancing replication efficiency. Thus, while the effect of a single gene is limited, the cumulative contribution of multiple such phage genes could substantially enhance phage fitness—a possibility that requires further investigation.

Previously, we demonstrated that Lon protease could efficiently serves as an intrinsic anti-phage defense system through cleaving phage HNH endonuclease to inhibit genome packaging of PaoP5Δ*dap1* (Le et al, 2024). Here, we identified Dap2 as an ADS that directly inhibits Lon protease. While ADSs targeting established bacterial defenses like CRISPR-Cas and restriction-modification (R-M) systems are well characterized (Dong et al, 2019), many more ADSs are being discovered recently, includes these countering emerging immune pathways such as CBASS, Pycsar, Gabija, Thoeris, and Hachiman (Leavitt et al, 2022; Mayo-Munoz et al, 2024), which provides new insights into how phage evade bacteria immunity in natural environments.

Despite these advances, critical gaps persist in ADS biology. For example, certain ADSs exhibit incomplete protective capacity, exemplified by the phage protein Acb2, which only partially rescues phage titers in JBD67Δ*acb2* and JBD18 variants (Huiting et al, 2023). This partial efficacy highlights two key insights: the need for quantitative assessments to define ADS protective thresholds, and the potential existence of synergistic ADS pairs that act cooperatively to fully neutralize bacterial defenses (Murtazalieva et al, 2024).

Our study reveals a novel cooperative ADS mechanism. Previously, we demonstrated that Dap1 binds to the phage-encoded HNH endonuclease, providing partial protection against Lon protease-mediated degradation (Le et al, 2024). Here, we show that Dap2 directly binds to and inhibits Lon's proteolytic activity. In vitro reconstitution experiments demonstrated that combined expression of Dap1 and Dap2 completely blocks HNH degradation.

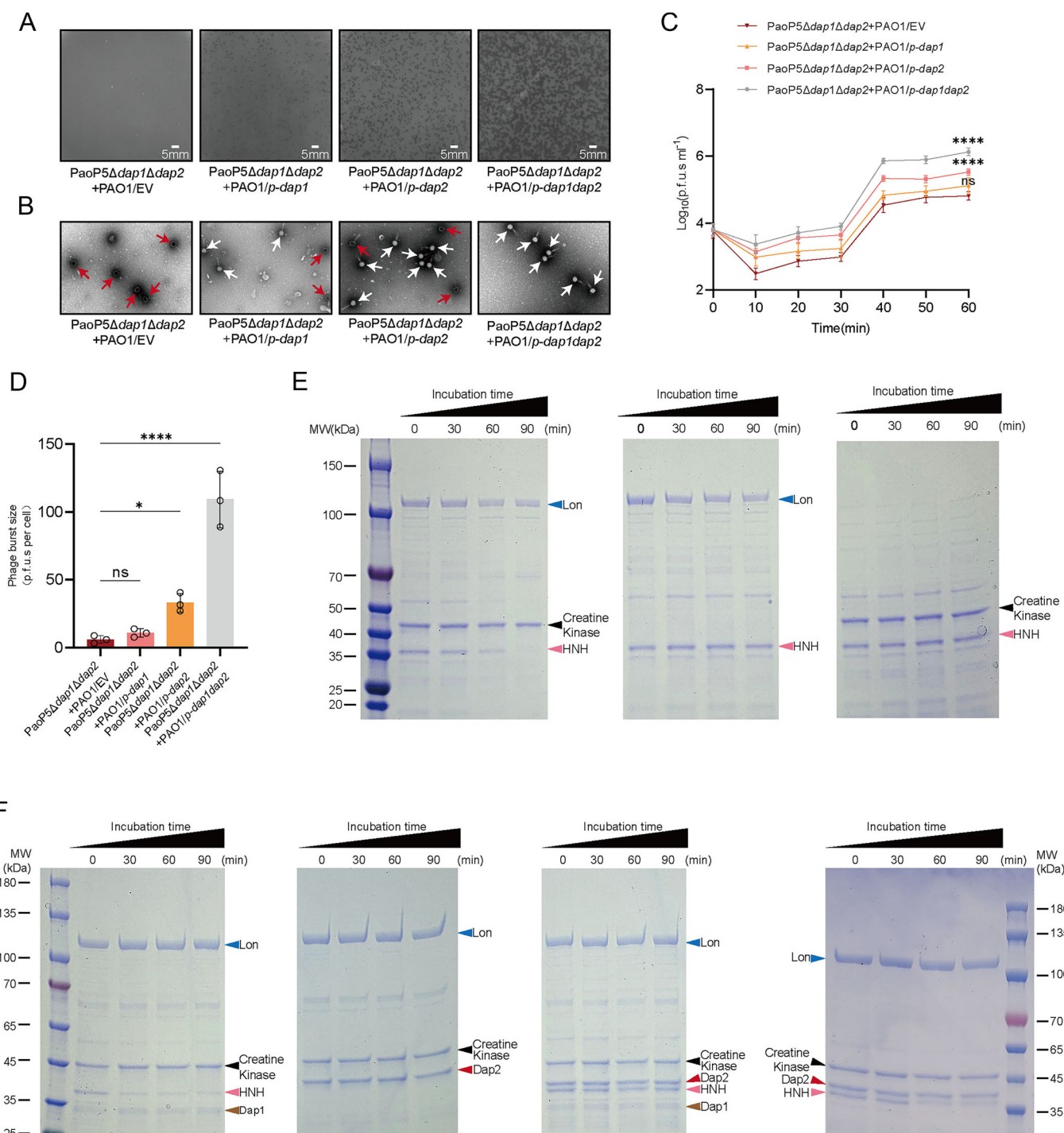

In vivo validation in *P. aeruginosa* PAO1 strains revealed that low-concentration expression of either Dap1 or Dap2 only partially restored plaque formation in PaoP5Δ*dap1*Δ*dap2* phages. Strikingly, co-expression of both proteins at low concentrations fully restored plaque morphology and increased burst size in PAO1::Dap1/Dap2 strains (Fig. 5A). Notably, the PaoP5Δ*Dap1*Δ*dap2* phage exhibited further reduced burst sizes due to HNH deficiency, underscoring the critical role of this ADS pair in phage fitness and therapeutic efficacy.

This synergy represents a compelling example of coordinated ADS action against Lon protease. Our findings suggest an evolutionary strategy where phages deploy complementary ADS combinations when single systems are insufficient. Genomic analysis revealed widespread conservation of *dap1*/*dap2*-like tandem overlapping gene pairs across phages (Fig. 6B), indicating their essential role in countering host immunity. This genomic colocalization further supports the hypothesis that synergistic ADS

**Figure 5. Dap2 and Dap1 cooperate to defend against Lon protease.**

(A) Plaque formation of PaoP5Δ*dap1*Δ*dap2* was assessed on PAO1, PAO1/*p-dap1*, PAO1/*p-dap2*, and PAO1/*p-dap1dap2*, demonstrating the impact of Dap1 and Dap2 complementation on phage infectivity. (B) Transmission electron microscopy (TEM) images of negatively stained PaoP5Δ*dap1*Δ*dap2* phages produced in PAO1, PAO1/*p-dap1*, PAO1/*p-dap2*, and PAO1/*p-dap1dap2* are shown. Red arrows highlight empty capsids, while white arrows indicate phages with packaged genomes. (C) The one-step growth curve revealed that PaoP5Δ*dap1*Δ*dap2* produced fewer progeny phages in PAO1 compared to the wild-type. However, complementation with both Dap1 and Dap2 significantly restored the burst size, underscoring their collective role in phage replication. Error bars represent the mean ± SD of three independent experiments. Significance at different time points was assessed by two-way ANOVA (60 min: PaoP5Δ*dap1*Δ*dap2* + PAO1/EV vs PaoP5Δ*dap1*Δ*dap2* + PAO1/*p-dap1*, $P = 0.0992$; PaoP5Δ*dap1*Δ*dap2* + PAO1/EV vs PaoP5Δ*dap1*Δ*dap2* + PAO1/*p-dap2*, $P < 0.0001$; PaoP5Δ*dap1*Δ*dap2* + PAO1/EV vs PaoP5Δ*dap1*Δ*dap2* + PAO1/*p-dap1dap2*, $P < 0.0001$). EV, empty vector. ns, not significant, ****$P < 0.0001$. (D) The burst size of PaoP5Δ*dap1*Δ*dap2* infecting PAO1, PAO1/*p-dap1*, PAO1/*p-dap2*, and PAO1/*p-dap1dap2* was 5.96 ± 1.97 PFU/cell, 10.71 ± 2.16 PFU/cell, 33.00 ± 4.86 PFU/cell, and 109.41 ± 14.71 PFU/cell, respectively. Error bars represent the mean ± SD of three independent experiments. Statistical significance was determined using Student's t-test (PaoP5Δ*dap1*Δ*dap2* + PAO1/EV vs PaoP5Δ*dap1*Δ*dap2* + PAO1/*p-dap1*, $P = 0.0992$; PaoP5Δ*dap1*Δ*dap2* + PAO1/EV vs PaoP5Δ*dap1*Δ*dap2* + PAO1/*p-dap2*, $P < 0.0001$; PaoP5Δ*dap1*Δ*dap2* + PAO1/EV vs PaoP5Δ*dap1*Δ*dap2* + PAO1/*p-dap1dap2*, $P < 0.0001$). EV, empty vector. ns, not significant, *$P < 0.05$, ****$P < 0.0001$. (E, F) In vitro proteolysis assays demonstrated that the combination of Dap1 and Dap2 completely prevented Lon-mediated degradation of HNH, whereas either protein alone only slowed the process. Complete substrate stabilization was achieved throughout the 90-min reaction when both proteins were present. Protein samples collected at specified time points were analyzed by 12% SDS–PAGE and visualized via Coomassie staining. All experiments were repeated at least three times with consistent results, and representative data are shown. Source data are available online for this figure.

pairs are evolutionarily favored, and investigating genes adjacent to known ADS loci may uncover novel defense-countering systems.

Phages represent promising therapeutic alternatives for treating bacterial infections (Li et al, 2023; Mu et al, 2025). Clinicians typically select phages that form large plaques and exhibit high lytic efficiency for treatment. However, the genetic basis underlying these desirable traits remains poorly understood. Our study identifies Dap2/Dap1 as key genetic determinants that enhance phage lytic efficiency by inhibiting the bacterial defense protein Lon. These findings have important implications for synthetic phage design: to optimize therapeutic efficacy, engineered phages should retain or incorporate such anti-defense genes in their genomes.

This study elucidates the dual functionality of a phage protein Dap2, which simultaneously targets the virulence regulator ExsA and the Lon protease. Furthermore, we unveil a synergistic ADS pair (Dap1/Dap2) that employs distinct mechanisms to fully neutralize Lon-mediated defense, exemplifying a sophisticated evolutionary strategy (Fig. 7). These findings advance our understanding of phage counter-defense tactics and highlight the potential of dual-function phage proteins as blueprints for novel antimicrobial and anti-virulence therapies.

## Methods

### Reagents and tools table

| Reagent/Resource | Reference or Source | Identifier or Catalog Number |
|---|---|---|
| **Experimental models** | | |
| List of *P. aeruginosa* and *E. coli* strains | This paper | Table EV1 |
| List of *Pseudomonas aeruginosa* phage PaoP5 | This paper | Table EV1 |
| BALB/c mice (female) | Jiangsu Huachuang Xinnuo Medical Technology Co., Ltd. | SCXK(Su) 2020-0009 |
| **Recombinant DNA** | | |
| List of plasmids | This paper | Table EV1 |

| Reagent/Resource | Reference or Source | Identifier or Catalog Number |
|---|---|---|
| **Antibodies** | | |
| Mouse anti DDDDK-Tag mAb | Abclonal | AE005 |
| Mouse anti-His-Tag mAb | Abclonal | AE003 |
| Mouse anti-GST-Tag mAb | Abclonal | AE001 |
| HRP-conjugated Goat anti-Mouse IgG (H + L) | Abclonal | AS003 |
| Purified anti-*E. coli* RNA Polymerase α Antibody | BioLegend | 663104 |
| **Oligonucleotides and other sequence-based reagents** | | |
| oligonucleotide sequence information | This paper | Table EV2 |
| **Chemicals, Enzymes and other reagents** | | |
| Lysogeny Broth (LB) medium | servicebio | G3101 |
| Kanamycin | biosharp | BS152-1g |
| Tetracycline | biosharp | BS172-5g |
| Gentamicin | Beyotime | ST2398-1g |
| Ethylenebis (oxyethylenenitrilo) tetraacetic (EGTA) | Solarbio | E8050-5g |
| Magnesium chloride hexahydrate (MgCl$_2$·6H$_2$O) | Solarbio | M8161-100g |
| FuturePAGE™ HP 4–12% 15 Wells | ACE Biotechnology | F15412MGel |
| MES-SDS Running Buffer | ACE Biotechnology | BR0002-01 |
| PVDF membranes (0.45 μm) | Servicebio | G6047-50-0.45 |
| 1*TBST Buffer | Solarbio | T1085 |
| ATP solution(100 mM) | Solarbio | A8270 |
| Dithiothreitol (DTT) | Solarbio | D8220 |
| Calcium Chloride, dihydrate (CaCl$_2$·2H$_2$O) | Solarbio | C8370 |
| Sodium Chlorid (NaCl) | Solarbio | S8212 |
| Ethylenediaminetetraacetic Acid (EDTA) | Solarbio | E8040 |

| Reagent/Resource | Reference or Source | Identifier or Catalog Number |
|---|---|---|
| Tris (Hydroxymethyl) Aminomethane | Solarbio | T8061 |
| Imidazole | Solarbio | I8093 |
| Skim Milk | Solarbio | D8341 |
| 2X Universal SYBR Green Fast qPCR Mix | Abclonal | RK21203 |
| 2×Es Taq MasterMix (Dye) | cwbio | CW0690S (1 ml) |
| PrimeSTAR® Max DNA Polymerase | TaKaRa | R045Q |
| PrimeScript™ RT reagent Kit with gDNA Eraser (Perfect Real Time) | TaKaRa | RR047A |
| Affinity Prestained Protein Ladder (10-250KD) | Affinity | KF8007 |
| Glycine | Solarbio | G8200 |
| Sucrose | Macklin | 57-50-1 |
| L- (+)-Arabinose | BBI | 5328-37-0 |
| Agarose, Regular | BBI | A620014 |
| Creatine Phosphate Disodium Salt Tetrahydrate | Sigma | 27920-1 G |
| Creatine Phosphokinase from rabbit muscle | Sigma | C3755-500UN |
| HisTrap HP 5 mL | Cytiva | 17524802 |
| GSTrap™ HP 5 mL | Cytiva | 17528201 |
| **Software** | | |
| GraphPad Prime V9 | N/A | https://www.graphpad-prism.cn/ |
| SnapGene software V8.0.3 | N/A | https://www.snapgene.cn/ |
| **Other** | | |
| ÄKTA pure | GE Healthcare/Cytiv | 29018224 |

## Bacterial strains, phages, and culture conditions

The bacterial strains, phages, and plasmids used in this research are listed in Table EV1. Both *E. coli* and *P. aeruginosa* strains were grown in Lysogeny Broth (LB) medium at 37 °C with shaking at 220 rpm. Antibiotics were added as required at the following concentrations: for *E. coli*, carbenicillin (100 µg/mL), kanamycin (50 µg/mL), and tetracycline (10 µg/mL); for *P. aeruginosa* PAO1, carbenicillin (300 µg/mL), tetracycline (100 µg/mL), and gentamicin (50 µg/mL). Bacteriophages were propagated in LB medium at 37 °C alongside their respective host bacteria.

## Construction of a promoter activity detection system

The promoterless *luxCDABE* reporter cluster in plasmid pMS402 was utilized to generate promoter-*luxCDABE* reporter fusions, following established methodology (Duan et al, 2003). Target promoter regions

were PCR-amplified and inserted into the pMS402 multiple cloning site. Recombinant plasmids were subsequently introduced into *Pseudomonas aeruginosa* PAO1 via electroporation. Bacterial luciferase activity, reflecting promoter-driven gene expression, was quantified in liquid cultures as light emission (counts per second, cps) using a Synergy 2 Multimode Microplate Reader (BioTek). Simultaneous measurements of luminescence and bacterial growth (optical density at 600 nm, $OD_{600}$) were recorded at 30-min intervals over a 24-hour incubation period. All data acquisition and analysis were performed using the instrument's native software package.

## Construction of plasmids

To construct the pME6032-*dap2* plasmid, the DNA fragment was amplified by PCR using the primer pair *orf004*-F and *orf004*-R (Table EV2). The PCR product was digested with EcoRI and HindIII restriction enzymes and then ligated into the similarly digested pME6032 vector, generating plasmid pME6032-*dap2*. This plasmid was transformed into the PAO1 strain, and transformants were selected on tetracycline-supplemented plates. Other plasmids were constructed using the same method.

To construct pHERDB20T-*dap2* plasmid, the *orf004* gene was amplified using the primers pHERD-*orf004*-F and pHERD-*orf004*-R, and the resulting PCR product was ligated into the pHERDB20T vector. The plasmid was then electroporated into the PAO1 strain, and transformants were selected on gentamicin-containing plates. The plasmid pHERDB20T-*dap1dap2* was constructed similarly using the primers listed in Table EV2. The plasmid pUCP24-*dap2* was generated following the same procedure as described above.

To construct the pUCP24-*dap2*(V52K) mutant plasmid, a site-directed mutagenesis PCR was performed using the PaoP5 genome as the template. The target sequence was amplified in two separate fragments using primer pairs *orf004*(V52K)-F/*orf004*-DTB-R and *orf004*(V52K)-R/*orf004*-DTB-F from Table EV2, with the mutation site incorporated into the primer sequences. The PCR products were analyzed by agarose gel electrophoresis to confirm their expected sizes, and the two target fragments were gel-purified. These purified fragments were then used as templates in an overlap extension PCR with the flanking primers orf004-DTB-F and orf004-DTB-R. The resulting full-length product was again verified by agarose gel electrophoresis and subsequently gel-purified, yielding the final mutated DNA fragment. Other mutations were introduced into the plasmid with the same procedures using the primers listed in Table EV2.

## Western blot analysis

Overnight cultures of test strains were subcultured (1:100 dilution) in fresh LB medium supplemented with 5 mM EGTA and 20 mM $MgCl_2$, then grown to mid-log phase ($OD_{600} = 0.6$). Equal protein amounts (20 µg) were resolved by 12% SDS-PAGE and electro-transferred to PVDF membranes (Millipore). Membranes were blocked with 5% non-fat milk in TBST (50 mM Tris-HCl, 150 mM NaCl, 0.1% Tween-20, pH 7.6) for 1 h at 25 °C, followed by overnight incubation at 4 °C with mouse anti-Flag monoclonal antibody and rabbit anti-RNA polymerase α subunit as a loading control. After washing, membranes were incubated with HRP-conjugated goat anti-mouse IgG or anti-rabbit IgG for 1 h at 25 °C, and protein signals were visualized using an ECL Plus kit (Amersham Biosciences) on an ImageQuant LAS 4000 imager.

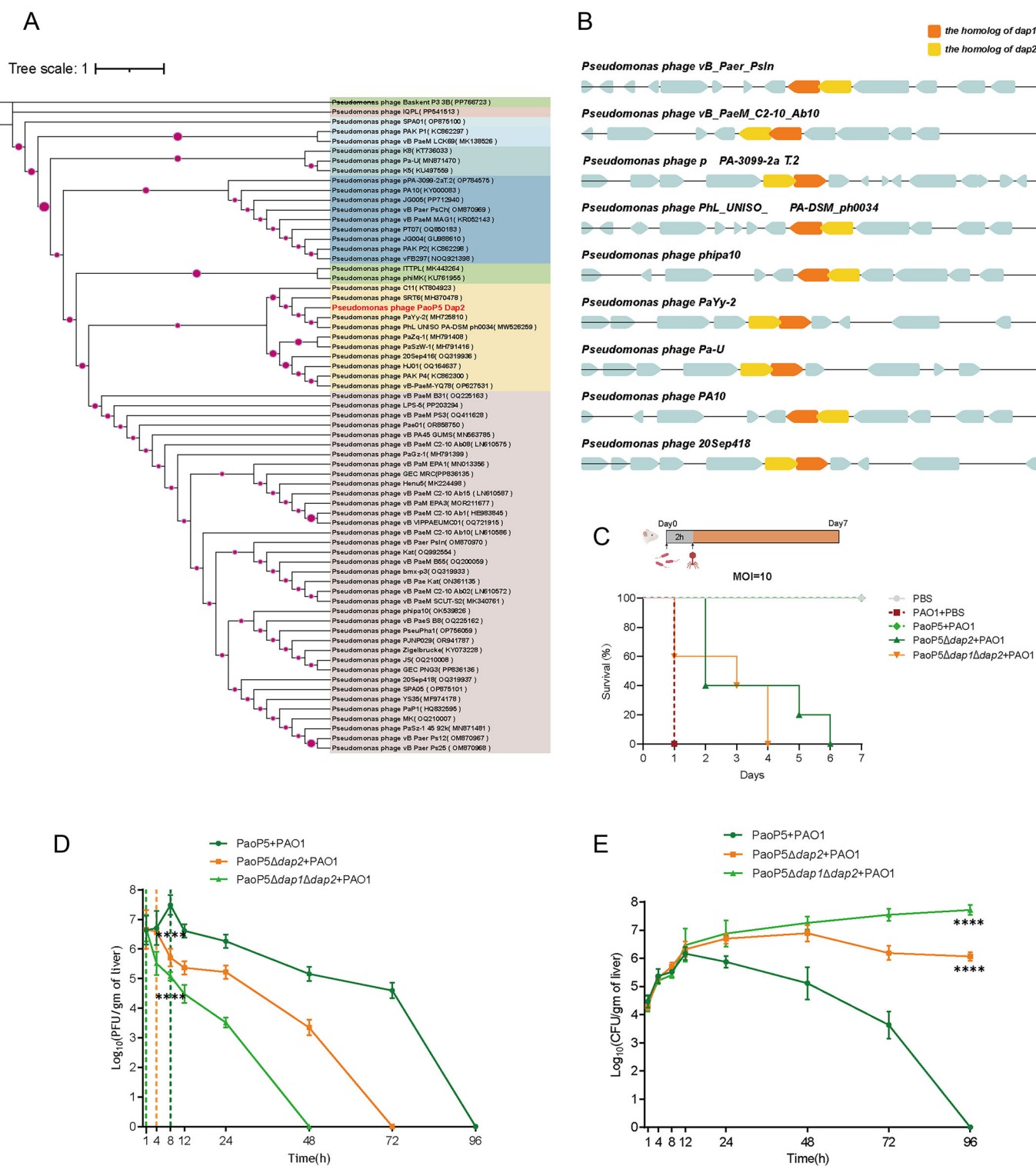

## GST Pull-down assay

In GST pull-down experiments, 100 μg of purified GST or GST-Dap2 was incubated with 25 μL pre-washed MagneGST™ glutathione beads (Promega) for 2 h at 4 °C, followed by three washes with GST binding buffer to remove unbound proteins.

Subsequently, 20 μg of His₆-ExsA or His₆-Lon was added to the bead complexes for 4 h at 4 °C, after which nonspecific interactions were eliminated by washing with GST washing buffer. Retained proteins and input controls were resolved via SDS-PAGE, transferred to membranes, and subjected to immunoblotting using anti-GST (Immunoway) and anti-His (TransGen Biotech)

**Figure 6. Dap2/Dap1 pair is co-localized in phage genomes and cooperates to enhance the efficacy of phage therapy.**

(A) A phylogenetic tree of Dap2 homologs with a minimum identity of 80% was constructed. The tree was generated using neighbor-joining analysis in MEGA 7, with evolutionary distances calculated using the p-distance method. The phylogenetic trees were visualized using iTOL. (B) Dap1 and Dap2 homologs, which are tandem overlapping genes, were identified in 67 phage genomes using BLAST. Nine representative phage genomes containing Dap1 and Dap2 homologs are illustrated. (C) Survival of 7-week-old BALB/c mice (n = 5) following intraperitoneal injection with PAO1 and phage PaoP5Δ*dap2*, PaoP5Δ*dap1*Δ*dap2*, or PaoP5 at an MOI of 10. (D) Phage titers and (E) bacterial CFUs were measured in the livers of mice intraperitoneally injected with PAO1 and phage PaoP5Δ*dap2*, PaoP5Δ*dap1*Δ*dap2*, or PaoP5 at an MOI of 10. The maximum titer of PaoP5 was detected at 8 h post-injection (~ $3.8 \times 10^7$ PFU/g). The phage titer gradually decreased after 24 h and was cleared within 96 h. Concurrently, bacterial CFUs in the liver decreased steadily and were fully cleared within 96 h. In contrast, for PaoP5Δ*dap1*Δ*dap2*-treated, PAO1-infected mice, the phage titer continuously decreased and became undetectable by 48 h, with a maximum titer of only ~$4.3 \times 10^5$ PFU/g-significantly lower than that observed in PaoP5-treated mice. Error bars represent the mean ± SD of the CFU or PFU from four independent mice. Significance versus the control group at each time point was assessed by two-way ANOVA with Dunnett's multiple comparisons test (phage titers, 8 h: PaoP5 + PAO1 vs PaoP5Δ*dap2* + PAO1, P < 0.0001; PaoP5 + PAO1 vs PaoP5Δ*dap1*Δ*dap2* + PAO1, P < 0.0001; bacterial CFUs, 96 h: PaoP5 + PAO1 vs PaoP5Δ*dap2* + PAO1, P < 0.0001; PaoP5 + PAO1 vs PaoP5Δ*dap1*Δ*dap2* + PAO1, P < 0.0001). ****P < 0.0001. Source data are available online for this figure.

antibodies, with protein signals visualized using an ECL Plus Kit (GE Healthcare, USA).

## RNA-seq and data analysis

RNA-seq was performed as previously described (Zhong et al, 2020). The PAO1/EV and PAO1/pME6032-*dap2* strains were grown in LB medium at 37 °C until the $OD_{600}$ reached ~0.6. Total RNA was promptly isolated using TRIzol reagent (Invitrogen) following the manufacturer's protocol. Ribosomal RNA was removed using the Ribo-Zero rRNA depletion kit. Subsequently, cDNA libraries were prepared and sequenced on an Illumina HiSeq 2500 platform. Each RNA-seq experiment was performed in triplicate. Sequencing reads were aligned to the *P. aeruginosa* reference genome (NC_002516.2, sourced from the National Center for Biotechnology Information) using Bowtie2, and only uniquely mapped reads were retained for further analysis. Differentially expressed genes (DEGs) were identified with DESeq2, applying a Benjamini-Hochberg-adjusted P < 0.05 and |log2 fold change| > 1 as significance thresholds. The RNA-seq data have been deposited in the BioProject database under accession number PRJNA1232226.

## Real-time quantitative PCR (RT-qPCR)

To validate the RNA-seq data, PAO1 and PAO1/*p-dap2* samples were prepared as described above. To validate the expression of *dap2*, 10 mL of bacterial PAO1 culture ($OD_{600}$ = 0.6) was infected with phage at an MOI of 10, and the culture grew at 37 °C with shaking. For RNA extraction, 1 mL of the culture was taken at given time points. Three biological repeats were performed. Total RNA was extracted from each sample by RNAprep Pure Cell/Bacteria Kit (TIANGEN, China), rRNA was removed, and cDNA was generated using the PrimerScriptTM RT reagent kit with gDNA eraser (Takara, Japan). RT-qPCR was performed using 2X Universal SYBR Green Fast qPCR Mix (ABclone, China). The primers used in this study are listed in Table EV1. The 16S rRNA gene was used as the reference gene for normalization, and the expression of each gene was compared using the delta-delta Ct method.

## Bacterial two-hybrid assays

The bacterial two-hybrid assay was performed as previously reported (Zhao et al, 2016). Briefly, DNA fragments encoding Dap2 were PCR-amplified and cloned into the bait vector pBT to generate pBT-Dap2. The pTRG-ExsA vectors were constructed and co-transformed with pBT-Dap2 into the *E. coli* reporter strain.

Transformants were selected on LB agar containing 5 mM 3-amino-1,2,4-triazole (3-AT) and incubated at 30 °C for 48 h. Positive colonies were subsequently streaked onto dual-selection media supplemented with 5 mM 3-AT and 12.5 µg/mL streptomycin.

## Modeling for ExsA-Dap2 complex using molecular docking

The initial models of each protein with an average pLDDT values of ~80 were generated by AlphaFold server (Abramson et al, 2024). Based on the predicted models, molecular docking was carried out to analyze the potential interaction modes between ExsA and Dap2, followed by one round of water refinement as implemented at the HADDOCK2.4 server [https://wenmr.science.uu.nl/haddock2.4/] (van Zundert et al, 2016; Yan et al, 2020). The interaction interface of ExsA-Dap2 hetero-dimer candidate complex was further analyzed and evaluated by PDBePISA server [https://www.ebi.ac.uk/pdbe/pisa/] (Vangone et al, 2011). Carry incidentally, AF3 failed to predict the complex structure due to low ipTM value.

## Electrophoretic mobility shift assay (EMSA)

EMSAs were conducted as described (Yang et al, 2023) to evaluate Dap2-ExsA interaction and its effect on ExsA DNA-binding activity. The *exsC* promoter region (amplified from *Pseudomonas aeruginosa* PAO1 genomic DNA) was incubated with an equimolar amount of His$_6$-ExsA in binding buffer (20 mM Tris-HCl pH 7.5, 200 mM KCl, 2 mM EDTA, 2 mM DTT, 200 µg/mL BSA, 5 µg/mL salmon sperm DNA) for 10 min at 25 °C, followed by addition of increasing concentrations of GST-Dap2 or GST-Dap2 point mutant for additional 10 min. Protein-DNA complexes were resolved on 6% native polyacrylamide gels in 0.5× TBE buffer (45 mM Tris-borate, 1 mM EDTA, pH 8.3) at 90 V for 1 h, stained with SYBR™ Gold nucleic acid gel stain (Invitrogen), and imaged using a Tanon-5500 phosphorimager. All assays were performed in triplicate with three biological replicates. The interaction between the mutated Dap2 proteins and ExsA was assessed as described.

## Mouse infection and phage therapy experiment

The Animal Research Ethics Committee of Army Medical University reviewed, approved, and supervised all animal protocols (AMUMEC20250001, AMUWEC20250096). For the mouse infection study, PAO1/pUCP24, PAO1/pUCP24-*dap2*, PAO1/pUCP24-*dap2*(E97A) and PAO1/pUCP24-*dap2*(E99A) strains were cultured

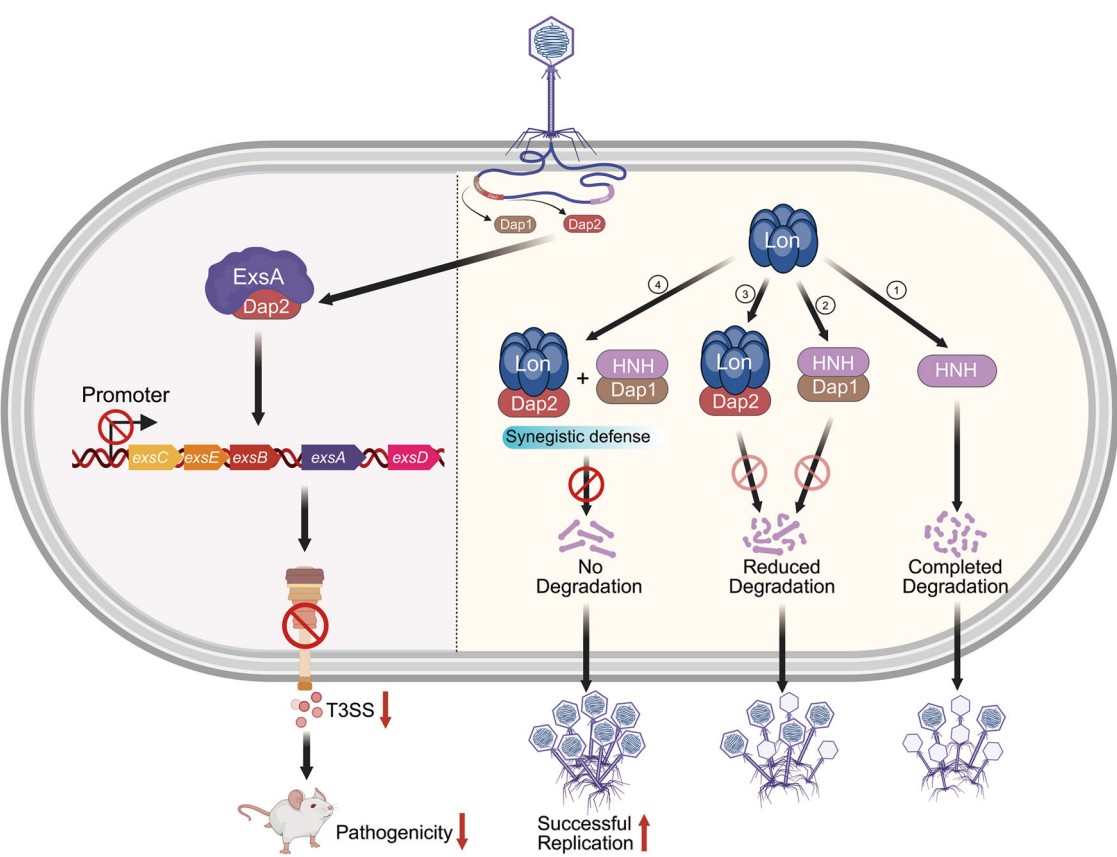

**Figure 7. Proposed model depicting the dual mechanisms by which Dap2 suppresses bacterial T3SS and collaborates with Dap1 to counteract Lon-mediated anti-phage defense.**

The phage-encoded protein Dap2 interacts with ExsA, the master regulator of the T3SS, thereby inhibiting T3SS expression and significantly attenuating the virulence of *P. aeruginosa*. Furthermore, during PaoP5 infection of PAO1, Dap1 binds to the HNH endonuclease, protecting it from Lon-mediated degradation, while Dap2 directly interacts with the Lon protease, effectively inhibiting its activity. Individually, Dap1 or Dap2 offers only partial protection against Lon-mediated anti-phage defense. However, their combined action completely neutralizes Lon protease activity, ensuring the preservation of the HNH endonuclease and facilitating efficient phage genome packaging. This synergistic interplay between Dap1 and Dap2 guarantees the production of sufficient phage progeny, exemplifying a refined evolutionary adaptation to overcome host immune defenses.

in LB medium at 37 °C until reaching the early stationary phase. Bacterial cells were harvested, washed, and resuspended in PBS to an $OD_{600}$ of 0.6. Each mouse group, consisting of 10 BALB/c female mice (7 weeks old), was intraperitoneally injected with 100 μL of bacterial suspension (~2 × 10^7 CFU) or PBS (negative control). Mice were monitored every 24 h for 7 days. Survivors at the end of the experiment were euthanized.

In the phage therapy experiment, 7-week-old BALB/c female mice were intraperitoneally injected with 50 μL of bacterial suspension (~6 × 10^7 CFU) or PBS (negative control). Two hours post-infection, 50 μL of phages (PaoP5, PaoP5Δ*dap1*, or PaoP5Δ-*dap1*Δ*dap2*, ~6 × 10^8 PFU) were administered intraperitoneally. Each group included 5 mice, which were monitored for 7 days. Survivors at the end of the observation period were euthanized.

To quantify phage and bacterial loads in the liver, measurements were taken up to 96 h post-phage administration, as phages become undetectable after 4 days. A total of 32 mice were treated with PaoP5, and 80 mice were treated with PaoP5Δ*dap1* or PaoP5Δ-*dap1*Δ*dap2* to account for potential mortality during the experiment. Only surviving mice were used for quantification. At 1 h, 4 h,

8 h, 12 h, 24 h, 48 h, 72 h, and 96 h post-phage administration, 4 mice from each group were euthanized. Liver tissues were homogenized as previously described (Dhungana et al, 2021). For phage titration, homogenized tissues were centrifuged at $10,000 \times g$ for 10 min, and phage titers in the supernatant were determined using the EOP assay. Bacterial counts were obtained by spreading 100 μL of 10-fold serial dilutions of tissue samples onto LB agar plates in duplicate, followed by incubation at 37 °C for 16 h.

## Plaque assay and efficiency of plating (EOP) assay

The phage plaque assay was performed following established protocols (Zhong et al, 2024). Briefly, 200 μL of log-phase bacterial culture ($OD_{600}$ of 0.6) was mixed with 100 μL of diluted phage solution (~100 pfu) and 4 mL of 0.4% LB agar. The mixture was then overlaid onto LB agar plates and incubated overnight at 37 °C until plaques became visible.

To determine the efficiency of plating (EOP) of phages on different bacterial strains, a mixture of 4 mL of 0.4% LB agar and 200 μL of log-phase bacterial culture ($OD_{600}$ of 0.6) was poured

onto LB agar plates to create a double-layer agar. Subsequently, 2 μL of serial 10-fold dilutions of the phage solution were spotted onto the prepared plates. The plates were incubated at 37 °C for 18 h to allow plaque formation.

## Knockout of phage genes using CRISPR-Cas9 system

The CRISPR-Cas9 plasmid pPTCS was utilized to knock out phage genes following established methods (Yang et al, 2023). To delete *dap2* in phage PaoP5, a spacer was created by annealing the primers 004-G2-F/R (Table EV2) and ligated into the Eco31I-digested pPTCS plasmid. For the recombination template, primers Δ004-LA-F/R and Δ004-RA-F/R were used to amplify two fragments upstream and downstream of the *dap4* gene, respectively. These fragments were ligated into the multiple cloning site of pTCPLS using Gibson Assembly. The resulting plasmid was introduced into PAO1 to generate PAO1/pTCPLS-Δg004G2D, which was then infected with $10^5$ PFU of phages. Survivors were selected using a plaque assay, and the mutant phage was confirmed by PCR and Sanger sequencing. The double deletion of *dap1* and *dap2* in phage PaoP5 was performed using the same approach, with the primers listed in Table EV2.

## Bacteriophage adsorption assay

A phage adsorption assay was conducted as previously described (Cen et al, 2022). Phages PaoP5Δ*dap1*Δ*dap2*, PaoP5Δ*dap2*, or PaoP5 were mixed with *P. aeruginosa* PAO1 ($OD_{600}$ of 0.6) at a multiplicity of infection (MOI) of 0.01. The mixture was incubated at 37 °C for 10 min and then centrifuged. Phage titers in the initial phage solution (t1) and the supernatant (t2) were determined using the double-agar layer method. The adsorption rate was calculated using the formula (t1-t2)/t1. Statistical significance was assessed using Student's t-test, with three biological replicates performed for each condition.

## One-step growth curve of phages

The one-step growth curve for PaoP5Δ*dap1*Δ*dap2*, PaoP5Δ*dap2*, or PaoP5 was conducted as previously described (Le et al, 2024). Briefly, phages were mixed with 1 mL of log-phase bacterial culture at an MOI of 0.01 and incubated at 37 °C for 10 min. The mixture was centrifuged at $10,000 \times g$ for 1 min, and the pellet was resuspended in 100 mL of LB medium. The suspension was incubated at 37 °C for 60 min, with 0.2 mL of the culture sampled every 10 min. Each sample was centrifuged at $10,000 \times g$ for 1 min, and the phage titer in the supernatant was immediately measured using the EOP assay. The burst size was calculated as the ratio of the total number of phages released at the end of the growth cycle (40 min) to the total number of infected bacteria. Each experiment was performed in triplicate for every phage variant.

## Co-immunoprecipitation (Co-IP) assay

The GST Co-IP assay was performed as described previously (Si et al, 2017). Briefly, 100 μg of purified GST or an equal amount GST-Dap2 was mixed with 50 μL of pre-washed MagneGST glutathione beads slurry (Progma) in GST binding buffer (50 mM Tris, 150 mM NaCl, pH 8.0) and incubated for 2 h at 4 °C, after

which unbound GST proteins were washed away. Then, the PAO1 cell lysates were incubated with the protein-binding beads for an additional 6 h at 4 °C. The beads were then washed with GST washing buffer (50 mM Tris, 500 mM NaCl, pH 8.0) sufficiently and the beads bound proteins were dissolved in SDS sample buffer. After SDS-PAGE, proteins were visualized by silver staining (Beyotime, China). The samples from the gels (molecular weight from 26 to 100 kDa) retained by beads coated with GST-Dap2, but not GST alone, were analyzed by Matrix-assisted laser desorption/ionization (MALDI) mass spectrometry.

## Transmission electron microscopy (TEM)

The phages PaoP5Δ*dap1*Δ*dap2*, PaoP5Δ*dap2*, or PaoP5 were propagated on various bacterial strains, including PAO1, PAO1/*p-hnh*, PAO1/*p-dap2*, PAO1/*p-dap1dap2*, Δ*lon*, or Δ*lon/p-lon*. The phage lysates were analyzed using transmission electron microscopy (TEM) following a standard protocol (Le et al, 2024). Briefly, the lysate was applied to carbon-coated copper grids for 10 min, followed by negative staining with a 2% phosphotungstic acid solution for 30 s. Phage particles were then examined using TEM, and the percentage of phages with genome-containing capsids was determined by averaging counts from over 50 particles across three independent biological replicates.

## Protein expression and purification

To construct the pGEX-6p-1-*dap2*, pGEX-6p-1-*dap2*(V52K), pGEX-6p-1-*dap2*(A64K), pGEX-6p-1-*dap2*(N79A), pGEX-6p-1-*dap2*(V52KA64KN79A), pGEX-6p-1-*dap2*(E97A), pGEX-6p-1-*dap2*(E99A), pGEX-6p-1-*dap2*(E97AE99A), pET28a-*dap1*, pET28a-*hnh*, and pET28a-*lon* plasmids, the fragments of *dap1*, *dap2*, *hnh*, and *lon* were amplified by PCR with the corresponding primers listed in Table EV2, respectively. Subsequently, the amplified fragment was digested with corresponding enzymes and ligated into pET28a, or pGEX-6p-1 vector, respectively. The resultant plasmids were introduced into *E. coli* BL21(DE3). After this, the strains were cultured at 37 °C in LB medium. When the $OD_{600}$ reached 0.6, 0.5 mM isopropyl β-D-1-thiogalactopyranoside (IPTG) was added to induce protein production. The cultures were incubated at 16 °C for an additional 20 h.

The cells were collected via centrifugation and resuspended in buffer A [20 mM tris-HCl (pH 8.0), 300 mM NaCl, and 25 mM imidazole]. The cells were lysed by high-pressure homogenization and centrifuged. The supernatant was applied to a 5-mL HisTrap HP column (Cytiva), and the target protein was eluted via AKTA purifier (GE Healthcare) using buffer B [20 mM tris-HCl (pH 8.0), 300 mM NaCl, and 500 mM imidazole]. The target protein was collected and applied to a HiLoad 16/600 Superdex 75-pg gel filtration column (Cytiva) equilibrated with buffer composed of 20 mM tris-HCl (pH 8.0), 200 mM NaCl, and 2 mM dithiothreitol (DTT). The purified proteins were concentrated and stored at −80 °C.

## Protein degradation assay

The HNH endonuclease degradation assay was carried out as previously described (Yang et al, 2015). Briefly, 100 μM of Dap2 and/or 100 μM of Dap1 were combined with 100 μM of HNH

endonuclease and incubated on ice for 20 min. The reaction mixture was then added to 30 μg of Lon in a 50 μL buffer containing 4 mM ATP, 50 mM Tris-HCl (pH 8.0), 10 mM MgCl$_2$, 1 mM DTT, 80 μg/mL creatine phosphokinase (Sigma, #C3755), and 50 mM creatine phosphate (Sigma, #27920). The mixture was incubated at 37 °C for the specified duration, and a 10 μL aliquot was collected at each time point. SDS-PAGE loading buffer was added to the aliquot, followed by heating at 100 °C for 15 min. The degradation of HNH was analyzed using 12% SDS-PAGE. The extent of HNH protein degradation was assessed as described above.

## Phylogenetic analysis

Homologs of *dap1* and *dap2* were identified by searching against 26,648 phage genomes retrieved from the INPHARED database on January 20, 2025. The search was performed using MMseqs2 (Steinegger and Söding, 2017), and hits with an e-value below $10^{-5}$ were retained. Sequence alignment and phylogenetic trees were conducted using MEGA7 (Kumar et al, 2016), and the resulting trees were visualized using iTOL (Letunic and Bork, 2019).

## Statistical analyses

The statistical analysis employed in this study involved the utilization of the Student's *t*-test to compare data from two distinct groups. A significance level of $P < 0.05$ was adopted to determine statistical significance (Bao et al, 2020; Li et al, 2025; Liu et al, 2025).

# Data availability

The analyzed data and raw RNA-seq readings of PAO1 expressing *dap2* were uploaded to the NCBI GEO (PRJNA1232226).

The source data of this paper are collected in the following database record: biostudies:S-SCDT-10_1038-S44318-026-00740-0.

# Peer review information

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

## Acknowledgements

This study was supported by grants from the Guangdong Major Project of Basic Research (grant no. 2025B0303000005), the National Natural Science Foundation of China (32470196, 32170188 to HL, 32570191 to SL), the Shenzhen Science and Technology Program (20231120104808001), and Key University Laboratory of Metabolism and Health of Guangdong, SUST. The funders had no role in study design, data collection and interpretation, or the decision to submit the work for publication.

## Author contributions

Jingru Zhao: Resources; Data curation; Software; Validation; Investigation; Methodology; Writing—original draft. Yuhao Zhu: Data curation; Software; Validation; Methodology. Chenchen Wang: Software; Methodology. Fang Tian: Formal analysis; Methodology. Jun Deng: Formal analysis. Jianglin Liao: Investigation; Methodology. Zhuojun Zhong: Software. Jiazhen Liu: Methodology. Nannan Guo: Software; Visualization. Shuai Le: Conceptualization; Supervision; Writing—review and editing. Haihua Liang: Conceptualization; Supervision; Funding acquisition; Writing—review and editing.

Source data underlying figure panels in this paper may have individual authorship assigned. Where available, figure panel/source data authorship is listed in the following database record: biostudies:S-SCDT-10_1038-S44318-026-00740-0.

## Disclosure and competing interests statement

The authors declare no competing interests.

# Expanded View Figures

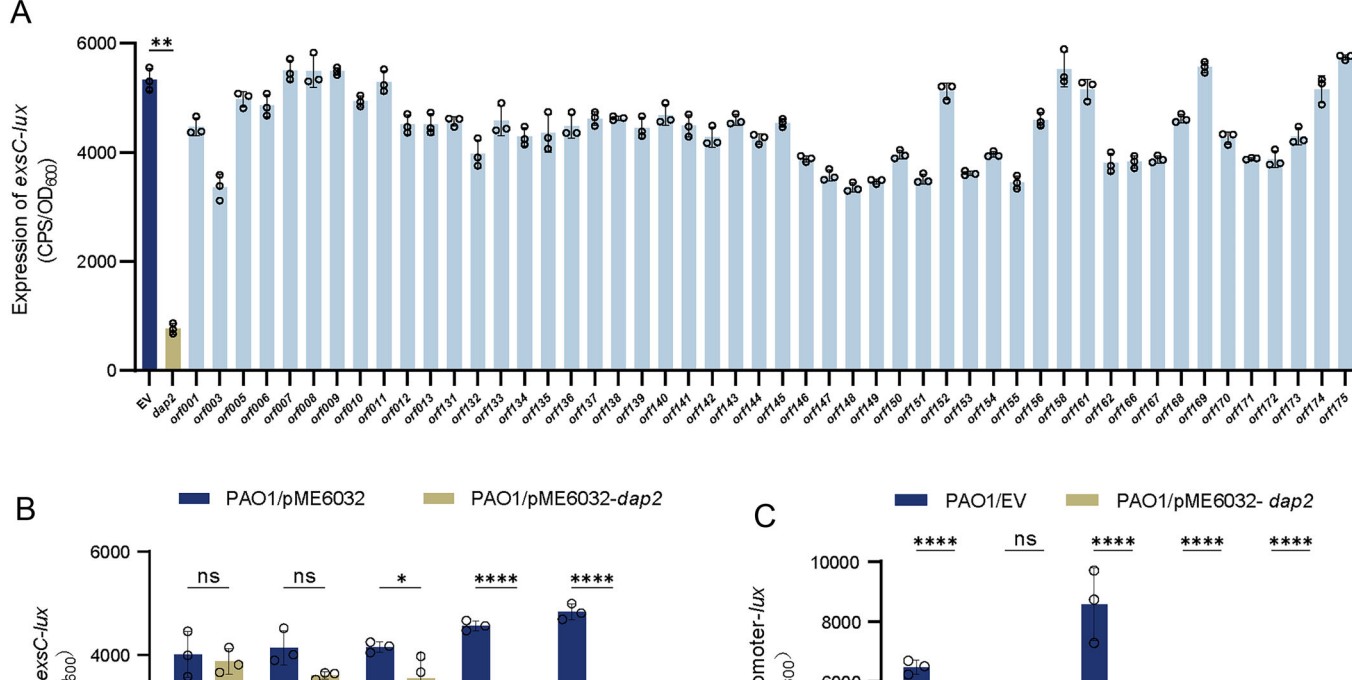

**Figure EV1. Overexpression of *dap2* inhibited the activity of T3SS-related genes.**

(A) The promoter activity of *exsC* was assessed in strains carrying either *exsC-lux* or 51 hypothetical ORFs derived from PaoP5. These constructs were individually expressed using the pME6032 vector and cultured in LB medium supplemented with 5 mM EGTA, 20 mM $MgCl_2$, and 0.5 mM IPTG. Promoter activity measurements were performed following 6 h of cultivation. Error bars represent the mean ± SD of three independent experiments. Statistical significance was determined using a two-sided Student's t-test (PAO1/EV vs PAO1/-*dap2*: $P < 0.0001$). EV, empty vector. **$P < 0.01$. (B) The expression of *exsC* in PAO1/pME6032 and PAO1/pME6032-*dap2* cultured in T3SS inducing medium supplemented with different concentrations of IPTG. Error bars represent the mean ± SD of three independent experiments. Two-way ANOVA was used to calculate p value (PAO1/EV vs PAO1/-*dap2*: 0 mM IPTG, $P = 0.9760$; 0.005 mM IPTG, $P = 0.0994$; 0.05 mM IPTG, $P = 0.0480$; 0.5 mM IPTG, $P < 0.0001$; 1 mM IPTG, $P < 0.0001$). ns, not significant, *$P < 0.05$, ****$P < 0.0001$. (C) The promoter activities of pKD-*exsA*, pKD-*exsC*, pKD-*exoS*, pKD-*exoT*, and pKD-*exoY* were comparatively analyzed in PAO1/EV and PAO1-pME6032-*dap2* strains under T3SS-inducing conditions supplemented with 0.5 mM IPTG. Following 6-hour cultivation, transcriptional activity measurements revealed differential expression patterns. (A–C) Error bars represent the mean ± SD of three independent experiments. Statistical significance was determined using a two-sided Student's t-test (PAO1/EV vs PAO1/-*dap2*: *exsC*, $P < 0.0001$; *exsA*, $P = 0.9983$; *exoS*, $P < 0.0001$; *exoT*, $P < 0.0001$; *exoY*, $P < 0.0001$). EV, empty vector. ns, not significant. ****$P < 0.0001$. Source data are available online for this figure.

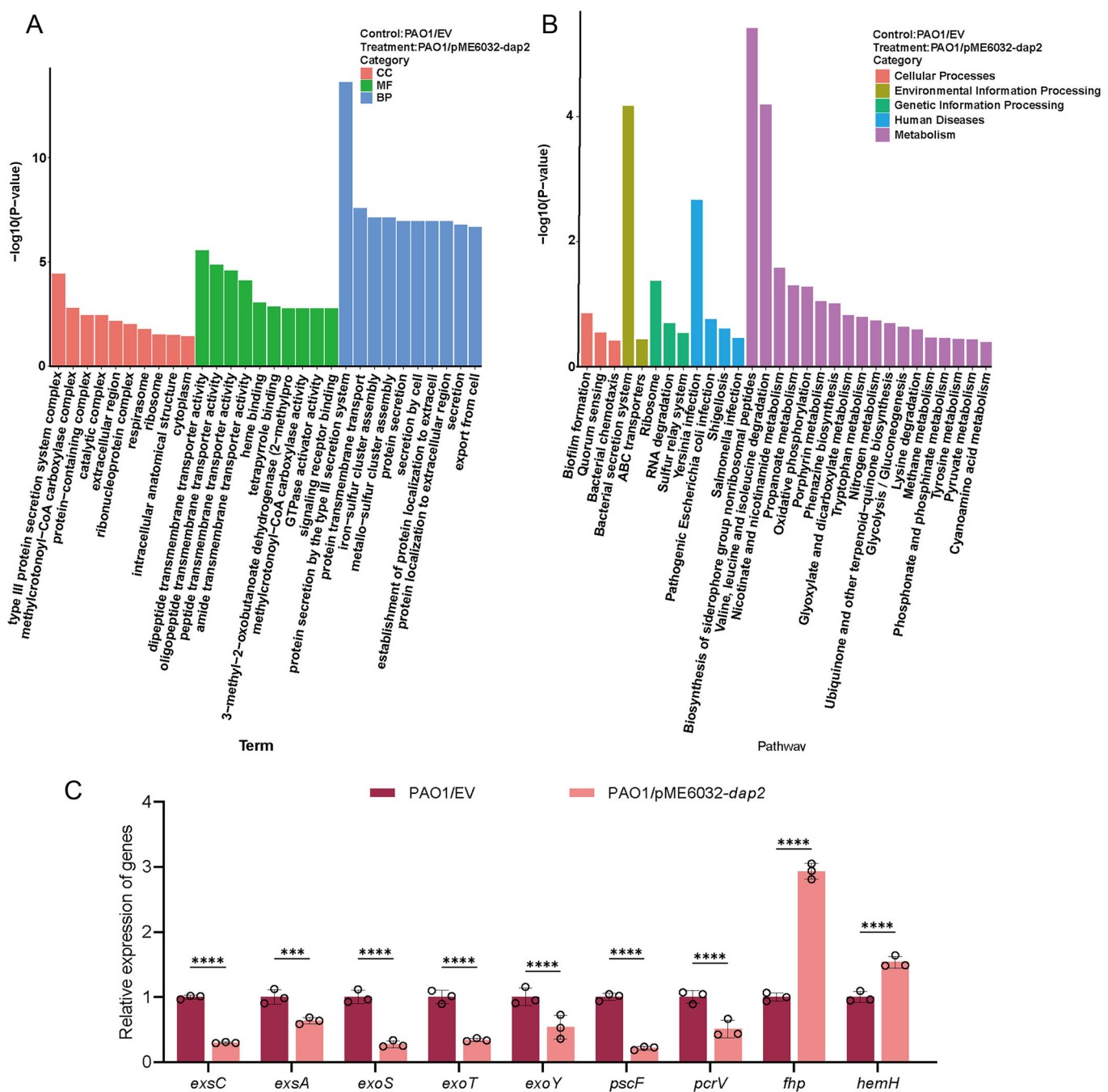

**Figure EV2.   RNA-seq analysis between PAO1 and PAO1/pME6032-*dap2*.**

(A) The GO enrichment of the differentially expressed genes in PAO1/EV and PAO1/pME6032-*dap2* strains, which is classified according to molecular function (MF), biological process (BP), and cellular component (CC), and the top 10 enriched GO was shown. (B) The DEGs are classified based on the KEGG analysis, and the top 30 enriched pathways, including Quorum Sensing and biofilm formation, are displayed. (C) Comparative qRT-PCR analysis of target gene expression in PAO1/EV versus PAO1/pME6032-*dap2* strains under T3SS-inducing conditions. Quantitative data are presented as mean ± SD of three independent experiments. Statistical significance was determined by two-way ANOVA (PAO1/EV vs PAO1/-*dap2*: *exsC*, $P < 0.0001$; *exsA*, $P = 0.0002$; *exoS*, $P < 0.0001$; *exoT*, $P < 0.0001$; *exoY*, $P < 0.0001$; *pscF*, $P < 0.0001$; *pcrV*, $P < 0.0001$; *fhp*, $P < 0.0001$; *hemH*, $P < 0.0001$). EV, empty vector. ***$P < 0.001$, ****$P < 0.0001$. Source data are available online for this figure.

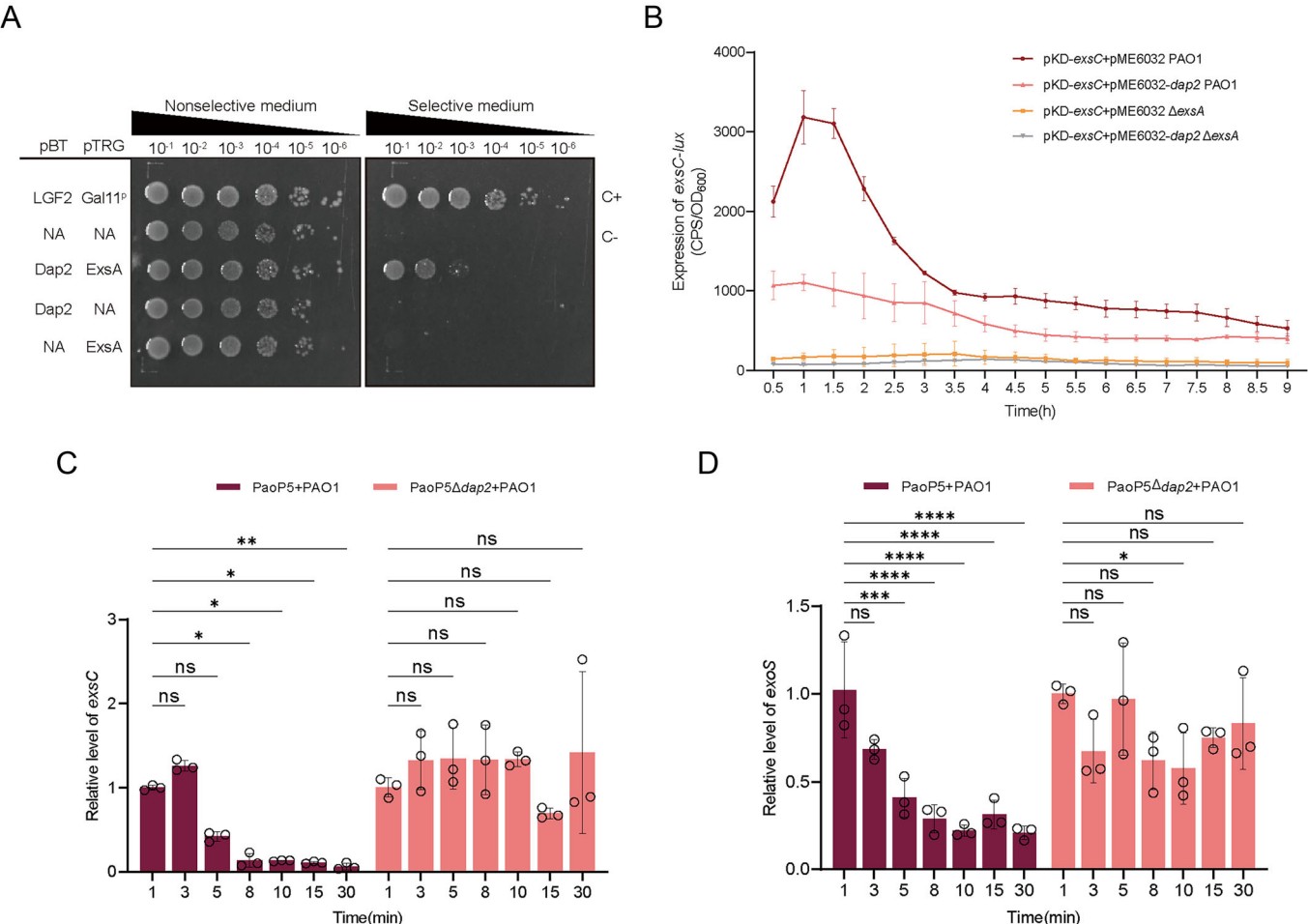

**Figure EV3.  Verification of the interaction between Dap2 and ExsA in vitro and in vivo.**

(A) *E. coli* two-hybrid assay reveals an interaction between Dap2 and ExsA. The recombinant strains harboring different plasmids were separately streaked on nonselective and dual-selective media. The strain expressing LGF2 and GAI11P was used as a positive control. (B) Expression of *dap2* did not suppress *exsC* activity in Δ*exsA* mutant strain. *P. aeruginosa* carrying pKD-*exsC* with pME6032-*dap2* was cultivated in T3SS-inducing medium. The promoter activity was detected at specified time points. Error bars represent mean ± SD of three biological replicates. (C, D) Expression of exsC and exoS during phage PaoP5 and PaoP5Δ*dap2* infection at the given time points. Error bars represent the mean ± SD of three independent experiments. Statistically significant variations were demonstrated by two-way ANOVA (PaoP5 + PAO1:0 min vs 3 min, $P = 0.8997$; 0 min vs 5 min, $P = 0.1769$; 0 min vs 8 min, $P = 0.0135$; 0 min vs 10 min, $P = 0.0129$; 0 min vs 15 min, $P = 0.0105$; 0 min vs 30 min, $P = 0.0065$; PaoP5Δ*dap2* + PAO1: 0 min vs 3 min, $P = 0.7734$; 0 min vs 5 min, $P = 0.7226$; 0 min vs 8 min, $P = 0.7645$; 0 min vs 10 min, $P = 0.7462$; 0 min vs 15 min, $P = 0.8022$; 0 min vs 30 min, $P = 0.5398$). EV, empty vector. ns, not significant, *$P < 0.05$, **$P < 0.01$, ***$P < 0.001$, ****$P < 0.0001$. Source data are available online for this figure.

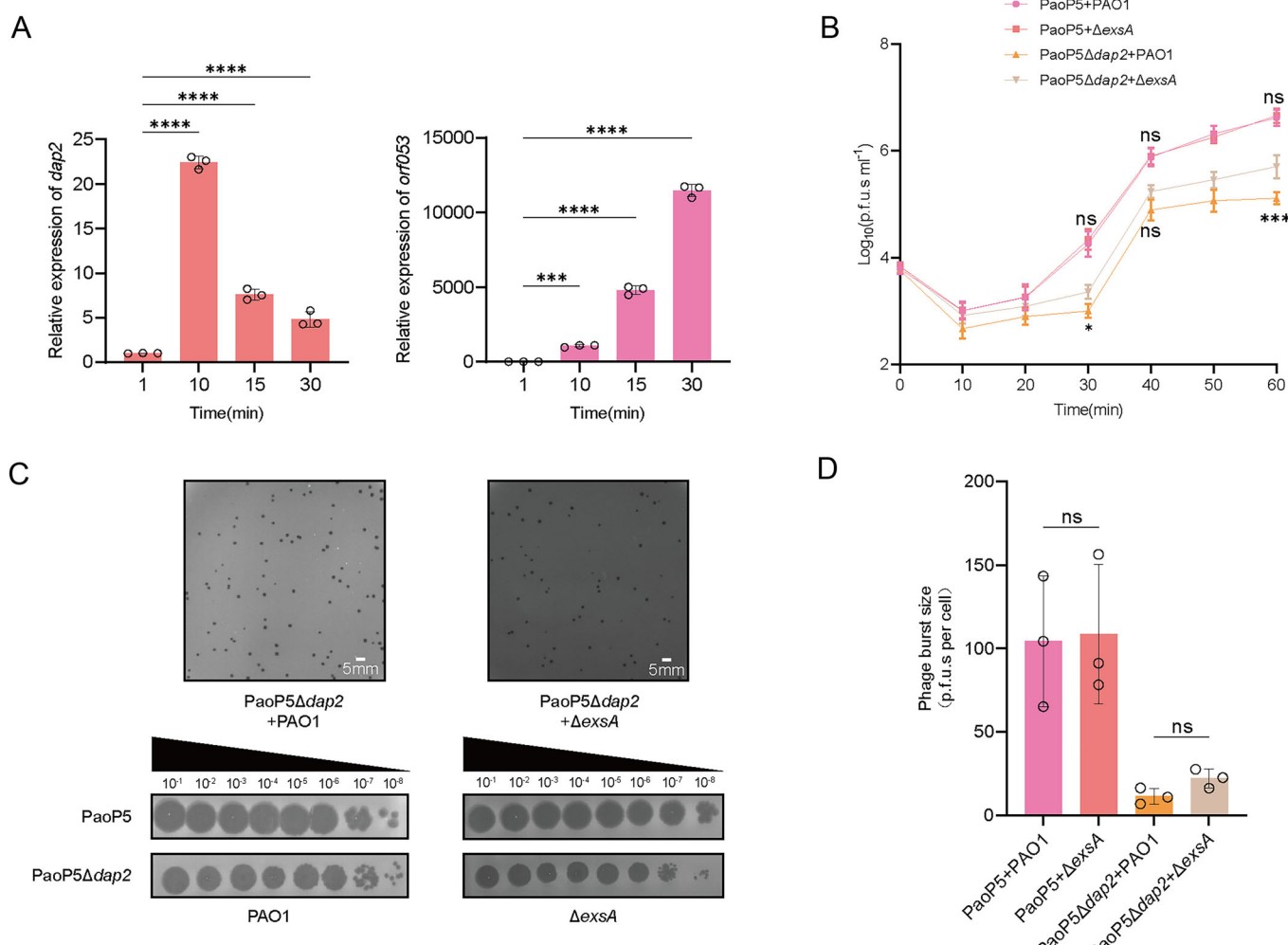

**Figure EV4. Dap2 is an early-expressed gene and deletion of *exsA* did not affect phage plaque formation.**

(A) qRT-PCR analysis of *dap2* and *orf53* expression at 1-, 10-, 15-, and 30-min after phage infection. *dap2* was expressed immediately after entering the host and the expression decreased after 10 min. Error bars represent the mean ± SD of three independent experiments. Statistically significance was assessed by one-way ANOVA (*dap2*:1 min vs 10 min, $P < 0.0001$; 1 min vs 15 min, $P < 0.0001$; 1 min vs 30 min, $P < 0.0001$; *orf053*: 1 min vs 10 min, $P = 0.0008$; 1 min vs 15 min, $P < 0.0001$; 1 min vs 30 min, $P < 0.0001$). ***$P < 0.001$, ****$P < 0.0001$. (B) The one-step growth curve of phage PaoP5 or PaoP5Δ*dap2* infecting PAO1 or Δ*exsA*. Error bars represent the mean ± SD of three independent experiments. Statistical significance was determined using two-way ANOVA (PaoP5 + PAO1 vs PaoP5+Δ*exsA*: 30 min, $P = 0.0782$; 40 min, $P < 0.0001$; 60 min, $P < 0.0001$; PaoP5Δ*dap2* + PAO1 vs PaoP5Δ*dap2*+Δ*exsA*: 30 min, $P = 0.0782$;40 min, $P < 0.0001$; 60 min, $P < 0.0001$). ns, not significant, *$P < 0.05$, ***$P < 0.001$. (C) Phage PaoP5Δ*dap2* was mixed with PAO1 or Δ*exsA*, and used double-layer agar plates to observe the plaque number and size. Data are representative of three independent replicates. (D) The burst size of wild-type PaoP5 infecting PAO1 and PAO1Δ*exsA* was 104.35 ± 27.67 PFU/cell and 108.70 ± 29.65 PFU/cell, respectively. In contrast, the burst size of PaoP5Δ*dap2* infecting PAO1 and PAO1Δ*exsA* was significantly reduced to 11.36 ± 3.89 PFU/cell and 22.08 ± 3.99 PFU/cell, respectively. Error bars represent the mean ± SD of three independent experiments. Statistical significance was determined using one-way ANOVA by Dunnett's multiple comparison test (PaoP5 + PAO1 vs PaoP5+Δ*exsA*, $P = 0.9972$; PaoP5Δ*dap2* + PAO1 vs PaoP5Δ*dap2*+Δ*exsA*, $P = 0.9613$). ns, not significant, *$P < 0.05$. Source data are available online for this figure.

## A

| Accession # | Molecular weight | Cover percentage | Peptides | m/z | PTM |
|---|---|---|---|---|---|
| tr\|Q9I5F9\|Q9I5F9_PSEAE | 88575 | 81% | K.KGVAMTGELTLTGQVLPIGGVR.E | 1099.1 | |
| | | | K.GVAM(+15.99)TGELTLTGQVLPIGGVR.E | 1043 | Oxidation(M) |
| | | | K.VLFVC(+57.02)TANTLDSIPGPLLDR.M | 1101.1 | Carbamidomethylation |
| | | | K.E(-18.01)SAEIAYSYIGSHLKK.Y | 593.3 | Pyro-glu from E |
| | | | K.ELLPLNPLYSEELKN(+.98)YLNR.F | 773.7 | Deamidation(NQ) |
| | | | R.Q(-17.03)KIFELILPEANR.G | 777 | Pyro-glu from |

## B

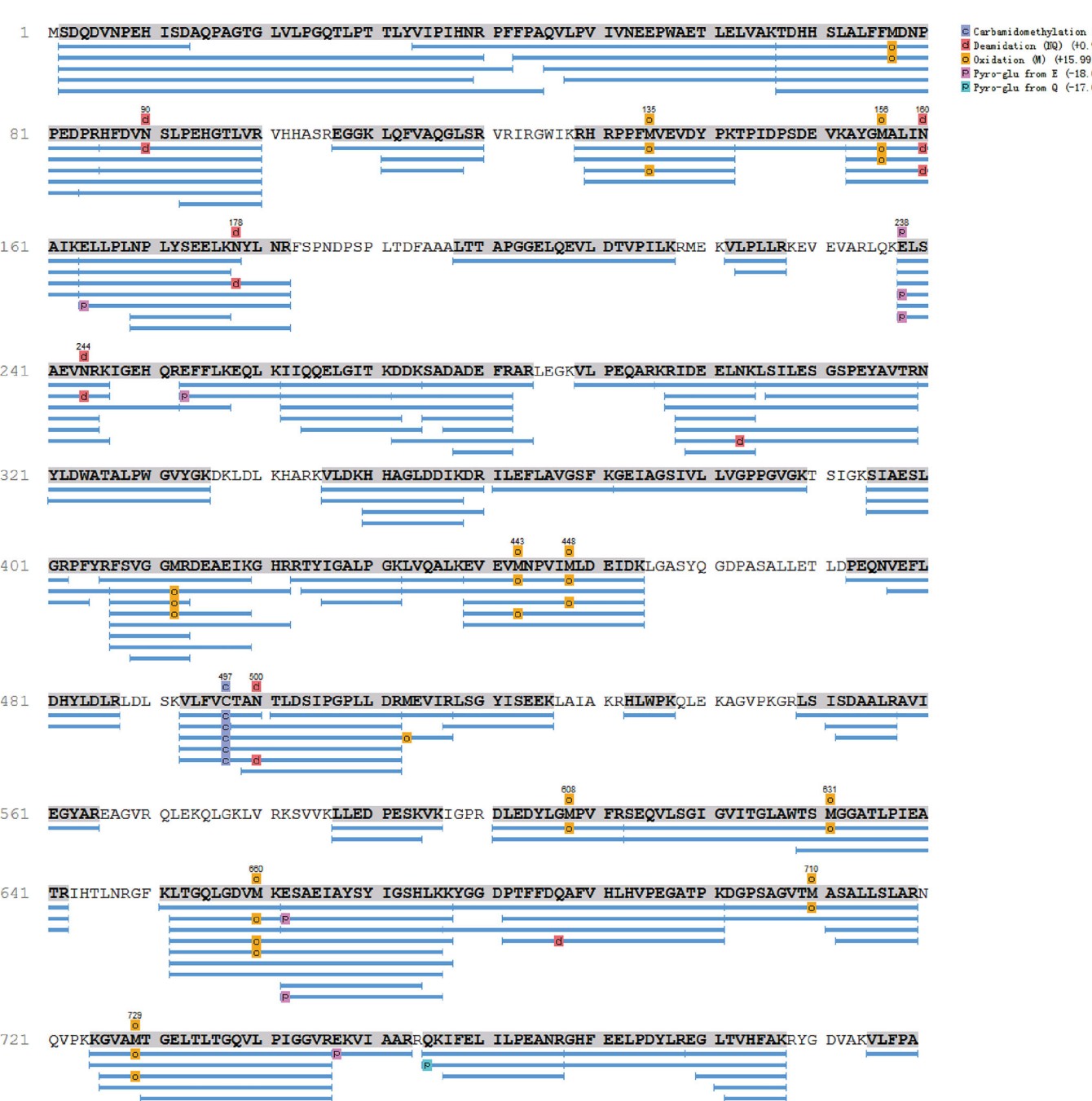

**Figure EV5. Identification of Lon protease that interacts with Dap2.**

(A) The specific protein bands identified in the Co-IP experiment were excised and subjected to mass spectrometry (MS) analysis. The MS results confirmed Lon as a novel interaction partner of Dap2, as indicated by the peptide matches and corresponding scores listed in the search results. (B) The coverage map displays peptide detection across the amino acid sequence of Lon, with peaks indicating identified regions.

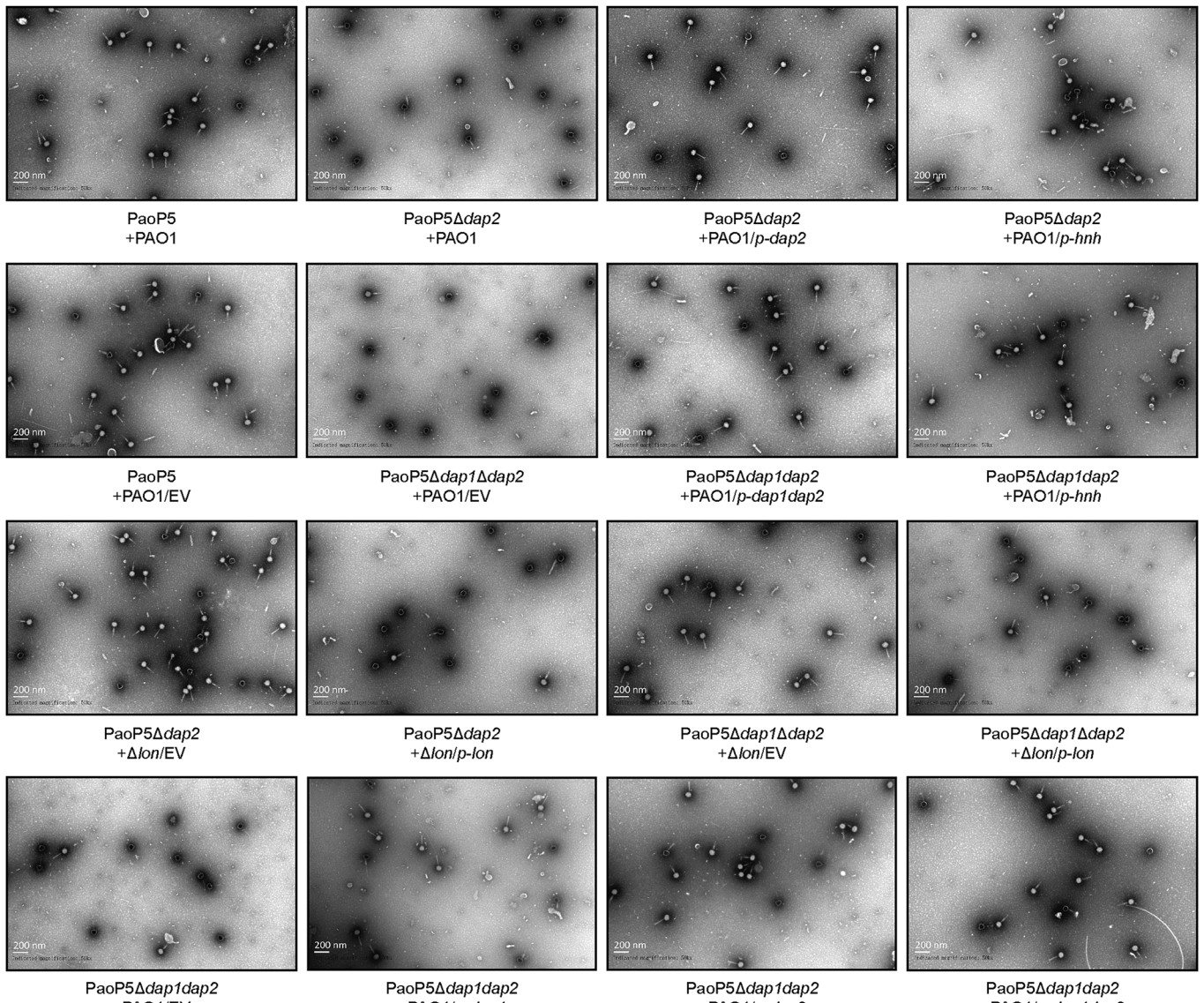

**Figure EV6. Representative transmission electron micrographs of negatively stained phages produced in PAO1, PAO1/p-hnh, PAO1/p-dap2, PAO1/p-dap2dap2, Δlon, and Δlon/p-lon.**

The empty capsids are black, and phages with white heads are packaged with the genomes. Source data are available online for this figure.

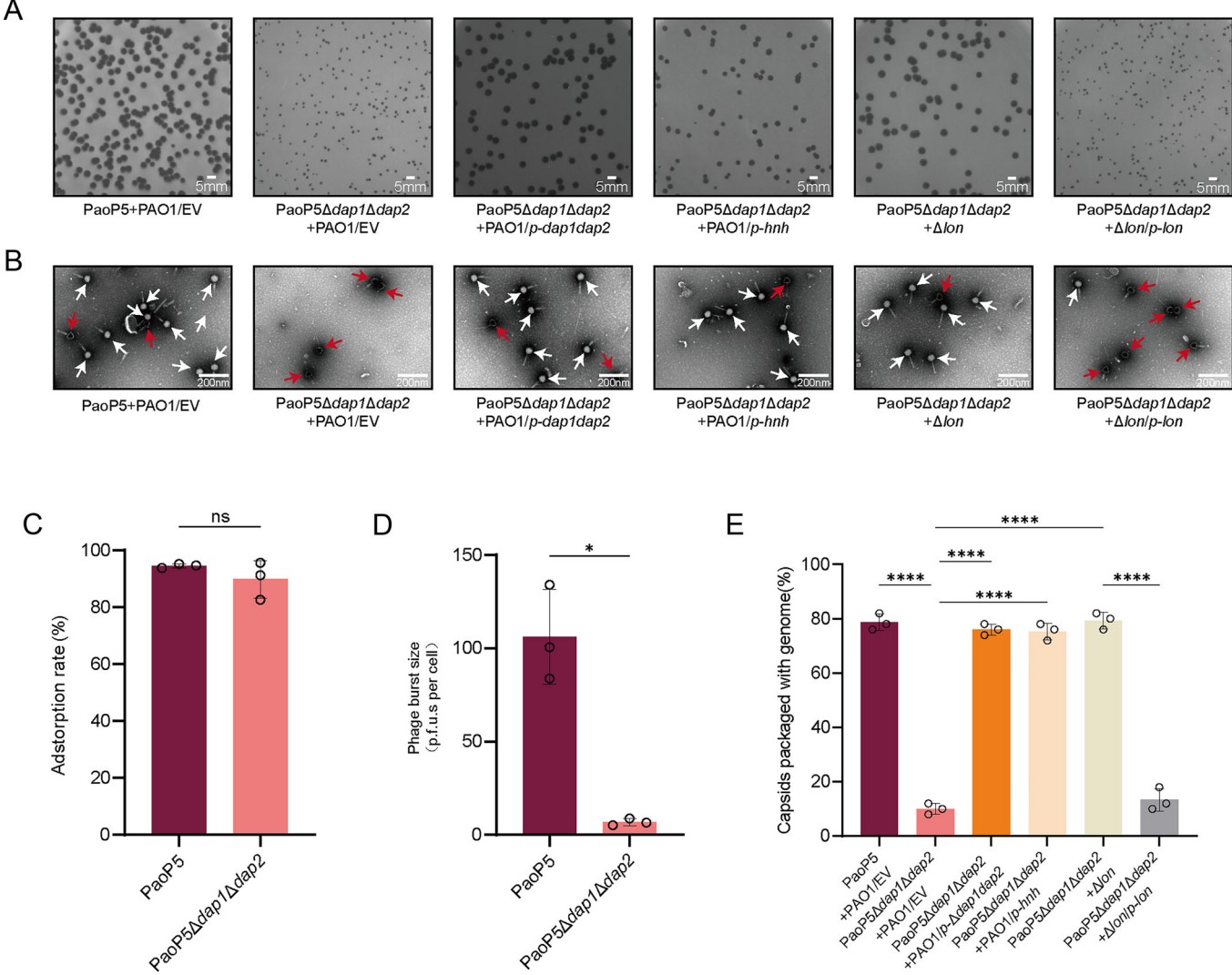

**Figure EV7. Impact of Lon and HNH on phage PaoP5Δdap1Δdap2.**

(A) The plaques of PaoP5 or PaoP5Δdap1Δdap2 infecting PAO1/EV, PAO1/p-dap1dap2, PAO1/p-hnh, Δlon, and Δlon/p-lon. (B) Representative TEM of negatively stained phages produced in PAO1/EV, PAO1/p-dap1dap2, PAO1/p-hnh, Δlon, and Δlon/p-lon. The red arrows indicate the empty capsids and the phages in which the genome is packaged are indicated by the white arrows. (C) Both PaoP5Δdap1Δdap2 and PaoP5 phages adsorbed to PAO1 efficiently. Error bars represent the mean ± SD of three independent experiments. Statistical significance was determined by an unpaired, two-tailed Student's *t* test (PaoP5 vs PaoP5Δdap1Δdap2: *P* = 0.3680). ns, not significant. (D) The burst size of phage PaoP5 or PaoP5Δdap1Δdap2 infecting PAO1. respectively. Error bars represent the mean ± SD of three independent experiments. Statistical significance was determined using a two-sided Student's t-test (PaoP5 vs PaoP5Δdap1Δdap2: *P* = 0.0229). *P < 0.05. (E) Phages were cultured in the indicated strains, and the percentage of capsids packaged with genomes was calculated from three biological repeats. Three biological repeats were performed, and 50 particles were counted for the presence or absence of the genome. Error bars represent the mean ± SD of three independent experiments. Significance versus the control group was assessed by one-way ANOVA with Dunnett's multiple comparisons test. *P* value was calculated based on one-way ANOVA Dunnett's multiple comparison test (PaoP5Δdap1Δdap2 + PAO1 vs PaoP5 + PAO1, *P* < 0.0001; PaoP5Δdap1Δdap2 + PAO1 vs PaoP5Δdap1Δdap2 + PAO1/p- dap1dap2, *P* < 0.0001; PaoP5Δdap1Δdap2 + PAO1 vs PaoP5Δdap1Δdap2 + PAO1/p-hnh, *P* < 0.0001; PaoP5Δdap1Δdap2 + PAO1 vs PaoP5Δdap1Δdap2+Δlon, *P* < 0.0001; PaoP5Δdap1Δdap2+ Δlon vs PaoP5Δdap1Δdap2+Δlon/p-lon, *P* < 0.0001). EV, empty vector. ****P < 0.0001. Source data are available online for this figure.

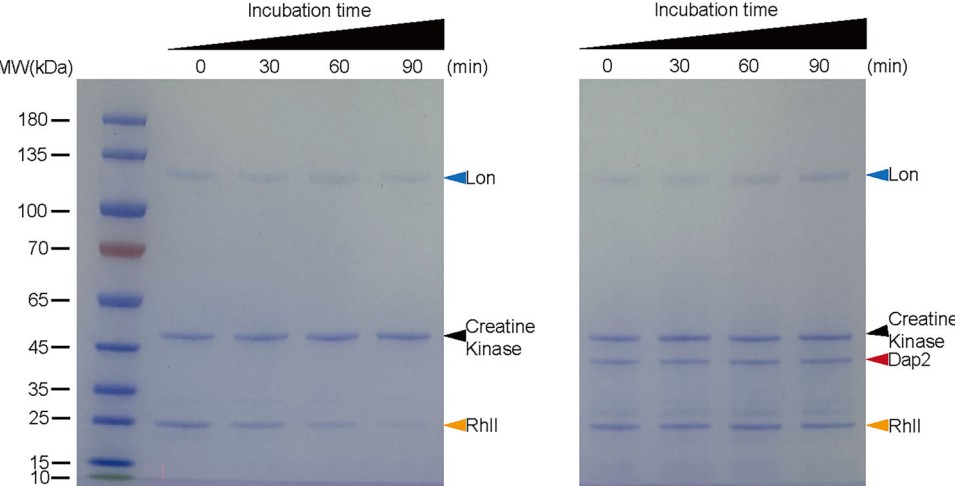

**Figure EV8.  In vitro proteolysis assays showed that Lon effectively degraded RhlI, and this degradation was suppressed upon the addition of Dap2.**

Protein samples collected at designated time points were analyzed using 12% SDS–PAGE and visualized by Coomassie blue staining. All experiments were independently repeated at least three times with consistent outcomes, and representative results are presented.

