## [Peer Review File · The EMBO Journal]

Simultaneous inhibition of bacterial virulence and anti-phage defense systems by synergistic bacteriophage counter-defense proteins

Jingru Zhao, Yuhao Zhu, Chen Wang, Fang Tian, Jun Deng, Jianglin Liao, Zhuojun Zhong, Jiazhen Liu, Nannan Guo, Shuai Le, and Haihua Liang

Corresponding authors: Haihua Liang (lianghh@sustech.edu.cn) , Shuai Le (leshuai2004@tmmu.edu.cn)

Review Timeline:

Submission Date:	28th Jun 25
Editorial Decision:	31st Jul 25
Appeal:	18th Aug 25
Editorial Decision:	4th Sep 25
Revision Received:	26th Nov 25
Editorial Decision:	16th Jan 26
Revision Received:	1st Feb 26
Accepted:	17th Feb 26

Editor: Ieva Gailite

Transaction Report:

Dear Dr. Liang,

Thank you for submitting your manuscript for consideration by The EMBO Journal. We have now received a full set of reviewer reports on your study, which are included below for your information. Based on these comments, we unfortunately had to conclude that the study is not a sufficiently strong candidate for publication in The EMBO Journal.

As you will see, while reviewers #1 and #3 appreciate the findings, reviewer #2 finds that the broader novelty of the study is limited by the previous publication from your team characterizing the role of Dap1 in Lon inhibition. Furthermore, all reviewers find that the proposed role of Dap2 in regulation of T3SS activity needs further substantiation and differentiation from its effect on Lon protease, and that the current characterisation of these interactions is rather preliminary. Additionally, reviewer #1 raises an important point that the effect of Dap1 on T3SS regulation could be mediated indirectly via Lon inhibition.

Since extensive research work beyond a single major revision round and with an unclear outcome appears to be needed to address the open questions, I am afraid that I cannot explicitly invite a revised version of the manuscript. However, if you find the majority of the concerns addressable, please contact me for a further discussion.

Alternatively, I have discussed your manuscript and referee comments with my colleague Achim Breiling at our sister journal EMBO Reports. I am glad to say that he is interested in considering your manuscript for publication after a revision. EMBO Reports would not request that point 1 of referee #1 is addressed experimentally. But point 2 of referee #1 is important and needs to be fully addressed. Moreover, as EMBO reports emphasises novel functional insight over detailed mechanistic insight, Achim will not require addressing points regarding more mechanism experimentally (i.e. points 3 of ref. #1 or 1 of ref. #2). However, it will be necessary that during revision all other points regarding the main conclusions of the study, and all technical concerns, or points regarding the experimental designs, model systems used, or data presentation, are addressed.

You do not need to revise the manuscript prior to transfer. Once you have initiated the transfer, Achim will send you an invitation to revise, outlining the scope of revision. Achim would be happy to discuss with you the transfer process or the revision plan, if further input is needed - you can reach him at a.breiling@emboreports.org.

If you find the transfer option of interest, you please use the link below to transfer the manuscript:

Link Not Available

Thank you in any case for the opportunity to consider this manuscript. I am sorry that I could not communicate more positive news, and I sincerely hope that you will find the transfer option of interest.

With kind regards,

Ieva Gailite

Referee #1:

The manuscript titled "Bacteriophage Dap2 inhibits bacterial T3SS and synergizes with Dap1 to evade anti-phage immunity" characterizes the dual function of the Dap2 protein from *Pseudomonas aeruginosa* phage PaoP5. First, Dap2 binds to the T3SS regulator ExsA, leading to transcriptional downregulation of T3SS-related genes and attenuated bacterial virulence in a mouse infection model. Second, Dap2 functions as an anti-defense protein by inhibiting the bacterial Lon protease, preventing degradation of a phage-encoded HNH endonuclease critical for efficient genome packaging. The authors further demonstrate that Dap2 acts synergistically with Dap1, a previously identified HNH-binding protein, to fully neutralize Lon protease-mediated anti-phage defense. The experiments are well-executed, and the findings are interesting. This study highlights the importance of coordinated anti-defense strategies, such as the Dap1/Dap2 system, in enhancing phage fitness and informing the development

of engineered phage therapeutics.

To further strengthen the manuscript, I offer the following major and minor suggestions:

Major Suggestion:

1. Biological rationale for T3SS inhibition remains unclear. The functional significance of Dap2-mediated inhibition of the T3SS via ExsA interaction is not immediately apparent. It is unclear why phage PaoP5 would benefit from suppressing host virulence in an infection context. One possibility is that certain ExsA-regulated genes impair phage infection or replication, making T3SS activity deleterious to phage fitness. To directly test this hypothesis, the authors should compare phage fitness parameters—including burst size, efficiency of plating (EOP), and plaque morphology—between wild-type *P. aeruginosa* PAO1 and a Δ exsA mutant, using both wild-type and Δ dap2 phage strains. In addition, it would be valuable to assess whether Dap2-mediated T3SS suppression enhances phage therapeutic outcomes in vivo by comparing bacterial clearance or survival in mouse infection models using wild-type versus Δ exsA hosts. These experiments would clarify the evolutionary or functional rationale for T3SS inhibition by Dap2.
2. Potential indirect effect of Lon inhibition on T3SS. Previous studies have shown that Lon protease can influence T3SS regulation in *P. aeruginosa* (e.g., PMID: 29362236, 28509355, 27657922). Since Dap2 inhibits Lon to protect the HNH endonuclease, the observed T3SS inhibition may result from Lon inactivation rather than direct Dap2-ExsA interaction. To differentiate between these possibilities, the authors should construct a Dap2 mutant (Dap2mut) that fails to bind Lon but retains ExsA binding. Expressing this mutant in *P. aeruginosa* and assessing virulence phenotypes in a mouse infection model will clarify whether Lon inhibition is responsible for T3SS suppression. If Lon inhibition is dispensable for the observed effect, Dap2mut and wild-type Dap2 should yield similar phenotypes.
3. Mechanistic basis of Lon inhibition by Dap2 is insufficiently defined. It remains unclear how Dap2 binds Lon and inhibits its proteolytic activity. Does Dap2 inhibit Lon globally or selectively block degradation of the HNH endonuclease? The authors should consider molecular docking analyses and introduce site-directed mutations in Dap2 and/or Lon to map the interaction interface. These mutants should be assessed via binding assays, plaque assays, and EOP assays to determine whether loss of interaction correlates with reduced HNH protection and phage fitness. Furthermore, a known Lon substrate should be included in the degradation assay to evaluate whether Dap2 inhibition is substrate-specific or broadly impairs Lon activity.

Minor Suggestion:

1. Line 360: "Figure 6b" should be corrected to "Figure 5?".
2. Figure 1b: The upper blot is labeled as anti-Flag, but based on the figure legend, it should be labeled anti-ExoS.
3. Figure 1d: The pull-down results are difficult to interpret. The input levels of His-ExsA appear unequal between samples. The observed reduction in pull-down signal may simply reflect lower input of His-ExsA rather than a difference in binding. Please ensure comparable input levels or normalize accordingly.
4. Figure 1e and Lines 194-198: For readers unfamiliar with ExsA, please provide additional details on its conformational changes. Also, clearly indicate the locations of the DNA-binding domain and Dap2-binding site on the ExsA structure.
5. Figure 2d: The x-axis label is missing.
6. Figure 3d: Ensure consistent nomenclature throughout the manuscript. Clarify whether Orf050 refers to the HNH endonuclease (hnh), and standardize the naming accordingly.
7. Figure 4c: Please include statistical analyses (e.g., error bars, p-values) to support the conclusions.
8. Figure 4f: Quantify the degree of HNH degradation in the protease assay and include this quantification in the figure.
9. Figure 4f: In the third gel panel, the Dap1 label points to a band that is inconsistent with the corresponding band in the first gel. Please clarify the labeling and ensure consistency.
10. Lines 280-284: The strain PAO1/p-hnh is discussed without prior introduction. Please provide context and experimental rationale when first mentioned.

Referee #2:

During bacteriophage infection, factors produced by the phage modulate activities of host processes, including transcription, translation, anti-phage immunity, and virulence. The authors of this manuscript previously discovered that Dap1, a protein produced by *Pseudomonas* phage PaoP5, prevents degradation by host Lon protease of a phage nuclease HNH, which is required for efficient packaging of phage genomes into nascent capsids.

Here, the authors identify a second factor (Dap2) in the same phage that directly inhibits Lon-mediated degradation and is also required for HNH protection. Separately from this activity, they find that Dap2 interacts with ExsA, a transcriptional activator of the T3SS required for *Pseudomonas* virulence in mammals. They go on to show that phage-treatment of mice infected with *Pseudomonas* have enhanced survival in a Dap1/Dap2-dependent manner.

The major shortcomings of the manuscript are that (i) the findings are relatively preliminary and need further substantiation, see details below, and (ii) they do not make a very significant advance over those previously reported by these authors: that Dap1 protects HNH from Lon-mediated degradation.

Detailed critiques:

1. The paper makes an abrupt pivot from the study of the role of Dap2 in T3SS inhibition to the study of Dap2 in Lon inhibition. The two activities appear genetically separable, but it is unclear whether the same parts of the Dap2 protein are responsible for inhibiting ExsA and Lon.
2. Does Dap2 affect the transcriptome in the absence of ExsA and/or Lon?
3. Is T3SS expression affected during phage infection? It seems as though all the related analyses were performed with plasmid

expression of Dap2.

4. The levels of HNH (and preferably additional known Lon substrates as well) during phage infection need to be measured to substantiate the conclusion that Dap2 protects HNH from Lon-mediated degradation *in vivo*.
5. Though the TEM is suggestive of a packaging defect in Δ dap2 phage, it is not a conclusive demonstration that these phages are devoid of DNA. DNA extracted from capsids should be quantified by qPCR or similar methodology. Also, even if the phages lack DNA, it could stem from inappropriate genome injection rather than inefficient packaging. These possibilities have not been distinguished by the existing analyses.

Minor:

1. The name "Dap2" stands for Defense anti-phage 2. This is not a very logical name for this protein, since it is not a defense or anti-phage protein but an inhibitor thereof.
2. Fig 1b: *exoS*-flag should be labeled.
3. Fig 1d: why is His-ExsA missing from the input? It is difficult to make a conclusion about the interaction without this control. Considering that Dap2 reduces ExsA levels in Fig. 1b, there should be less ExsA in the Dap2 input, not more.
4. Line 179-180: This conclusion cannot be made with confidence, because in the Δ exsA background, there is little or no *exsC* transcription for Dap2 to inhibit.
5. Fig S3c-d: His-ExsA purification has a substantial number of contaminating bands. It would help if a nonspecific DNA substrate of identical length were included to confirm the observed binding to the *ExsC* promoter is specific.
6. Fig 1f: is the Dap2 triple mutant stable *in vivo*?

Referee #3:

General Summary and Opinion

This manuscript reports the characterization of Dap2, a protein encoded by *Pseudomonas aeruginosa* phage PaoP5, which inhibits the bacterial type III secretion system (T3SS) by binding the master regulator ExsA and simultaneously inhibits Lon protease to protect the phage-encoded HNH endonuclease. Dap2 and its neighboring gene product Dap1, previously shown to bind HNH directly, together provide full protection from Lon-mediated degradation and enable efficient phage replication. The authors demonstrate that the Dap1/Dap2 pair enhances phage fitness and therapeutic efficacy in a murine infection model. Overall, the work is carefully executed and addresses an important question in phage-host interactions. The dual role of Dap2 in both virulence suppression and anti-defense is interesting, and the mechanistic dissection of Dap2's interaction with ExsA and Lon adds insight into phage counter-defense strategies. However, the rationale for why the authors targeted virulence phenotypes, such as T3SS, is poorly developed, and several experimental details require clarification or quantification.

Major Concerns

1. Clarify the study rationale and connection to virulence modulation

The stated goal (line 112) is to identify phage-encoded virulence modulators, yet the introduction is largely focused on anti-phage defense. The rationale for why phages would inhibit virulence systems like T3SS is not well articulated. The discussion (line 391) admits this biological significance is unclear. Given that virulence inhibition does not appear to meaningfully increase phage yield (e.g., Fig. S4d), a clearer argument should be made for why the authors pursued this hypothesis and how it connects to phage fitness.

2. Background on *Pseudomonas aeruginosa*

Given the central role of this pathogen, a brief overview of its relevance, especially in the context of phage therapy, T3SS, and intrinsic defenses like Lon, would benefit readers less familiar with the system.

3. Clarify luminescence readout (line 142)

The assay using the *exsC* promoter-lux fusion should be described as a proxy for transcriptional activation of T3SS genes, not T3SS activity *per se*. Similarly, clarify whether the *exoS* promoter reporter (line 179-180) is analogous.

4. *In vivo* plasmid retention (Fig. 1g)

The dramatic reduction in mouse mortality with Dap2-expressing strains implies *in vivo* repression of T3SS. However, the authors should confirm that plasmids were retained and consider the possibility that plasmid loss during infection could have affected the results were retained and actively expressed during infection. Was plasmid maintenance tested or monitored *in vivo*? Were bacterial burdens in tissue quantified in this experiment?

5. Quantify plaque size and clarify EOP (Fig. 2)

Data related to plaque morphology and size (e.g., Figs. 2a, 2b, 4a) are used to make claims of impaired fitness, but these should be quantified using plaque diameter or area measurements and statistical analysis. In Fig. 2b, dilution spot assays are shown, but these are not equivalent to EOP values. The authors should include proper EOP calculations (e.g., PFU on test strain divided by PFU on reference strain).

6. Introduce HNH earlier

HNH is referenced multiple times before its function is explained (lines 276-280). A brief introduction to HNH as a genome packaging endonuclease and Lon target earlier in the manuscript would improve clarity.

7. Delayed burst time not apparent (Fig. 4c)

The claim that PaoP5 Δ dap1 Δ dap2 shows delayed burst timing needs clarification. The growth curves in Fig. 4c do not obviously support this. Please define the criteria for assessing burst time and describe the basis for this conclusion.

8. Fig. 4f gel labeling

The gel images in Fig. 4f are not clearly labeled, making interpretation difficult. Labeling each gel and clarifying what the asterisk indicates would help. The identity of bands, especially HNH, should be explicitly marked in each gel.

9. Phylogenetic tree scale (Fig. 5a)

The tree lacks a scale bar. Readers cannot assess phylogenetic distance or confidence without it.

10. Figure citation correction (line 360)

The reference to Fig. 6b should be corrected to Fig. 5c.

11. Statistical analysis missing (Fig. 5d,e)

No statistical tests are reported for the phage/bacterial titers in mouse liver. This is critical for evaluating the claim of differential therapeutic efficacy.

12. Expand discussion with field context

The discussion primarily rephrases results without situating them in broader context and is also redundant with material presented in the introduction. It would be valuable to speculate why phages might inhibit virulence regulators like ExsA or T3SS- perhaps to reduce host inflammation, slow immune clearance, or alter bacterial physiology in ways that promote phage spread. The authors should also consider broader implications of cooperative anti-defense systems in phage evolution.

Minor Concerns

- ADS abbreviation: Consider avoiding this abbreviation unless space is a concern-it adds unnecessary jargon.
- Line 108: Include a reference when Acb2 is first mentioned as an example of partial defense activity.
- Line 147: Clarify that the comparison is between Dap2 expression and an empty vector control.
- Line 241: Refer readers to the figure panel where the relevant plaque images or data are shown.
- Line 266: "Degradation on" should be revised to "degradation by."
- Line 286: The claim about defective packaging in Δ dap2 lysates should be supported by a figure reference.
- Line 373: Refer readers to the figure/data showing phage titers in liver and related outcomes.

Suggestions

- Consider analyzing whether co-expression of Dap2 and Dap1 offers any fitness advantage in strains lacking Lon. This could help disentangle whether their function is specific to Lon inhibition or broader.
- Discuss whether the Dap1/Dap2 pairing is representative of a broader strategy, and whether other such synergistic/overlapping systems exist in known phages.

** As a service to authors, EMBO Press provides authors with the possibility to transfer a manuscript that one journal cannot offer to publish to another EMBO publication or the open access journal Life Science Alliance launched in partnership between EMBO Press, Rockefeller University Press and Cold Spring Harbor Laboratory Press. The full manuscript and if applicable, reviewers' reports, are automatically sent to the receiving journal to allow for fast handling and a prompt decision on your manuscript. For more details of this service, and to transfer your manuscript please click on Link Not Available. **

Dear Dr. Gailite,

We sincerely appreciate your thoughtful comments and suggestions. After carefully evaluating all points raised, we believe all reviewers' concerns can be satisfactorily addressed through revision. We would therefore like to request permission to resubmit a revised manuscript.

To address the major concerns, we propose the following:

As you will see, while reviewers #1 and #3 appreciate the findings, reviewer #2 finds that the broader novelty of the study is limited by the previous publication from your team characterizing the role of Dap1 in Lon inhibition.

We sincerely appreciate the reviewer's positive feedback and insightful comments. We fully agree that the inhibition of Lon by Dap2 shares some conceptual overlap with previous findings involving Dap1. However, the underlying mechanisms are distinct: whereas Dap1 protects the HNH domain from Lon-mediated degradation, Dap2 directly binds to and inhibits Lon itself.

Importantly, the Lon inhibition by Dap2 represents only one aspect of our study's novelty. Our work further reveals that:

- 1、 Dap2 suppresses T3SS activity, expanding its functional repertoire beyond Lon inhibition.
- 2、 Dap2 collaborates with Dap1 to enhance anti-defense activity, a synergistic interaction that aligns with observations from other reviewers and provides new insights into phage counter-defense strategies.

Together, these findings significantly advance our understanding of phage-bacterial interactions and underscore the multifaceted nature of phage-encoded defense inhibitors.

Furthermore, all reviewers find that the proposed role of Dap2 in regulation of T3SS activity needs further substantiation and differentiation from its effect on Lon protease, and that the current characterisation of these interactions is rather preliminary. Additionally, reviewer #1 raises an important point that the effect of Dap1 on T3SS regulation could be mediated indirectly via Lon inhibition.

We agree with the reviewers that elucidating how Dap2 regulates both T3SS and Lon (as raised in point 3 of reference #1 and point 1 of reference #2) represents a key mechanistic question. In particular, determining whether T3SS suppression is Lon-dependent (point 2 of reviewer #1) warrants careful investigation. These are excellent suggestions, and we are prepared to complete these experiments within two months.

Our preliminary work has already generated several Dap2 mutants with distinct

phenotypic profiles.

Notably, AlphaFold3 structural predictions reveal that ExsA and Lon protease inhibition are mediated by distinct domains of Dap2. Residues critical for ExsA inhibition (red; positions 52, 64, 79) are spatially separated from those involved in Lon protease inhibition (blue; positions 97, 99), with these functional sites located on opposite surfaces of the protein. This structural arrangement supports the existence of independent regulatory mechanisms for these two inhibitory functions.

EMSA assays demonstrate that the N79A, V52K, and A64K mutants lose their ability to inhibit ExsA, allowing ExsA to maintain DNA binding activity. Interestingly, plaque assays reveal that while the N79A mutant fails to inhibit ExsA, it retains the capacity to restore the large plaque phenotype, suggesting this residue is not involved in Lon protease inhibition.

Conversely, the E97A/E99A double mutant of Dap1 maintains ExsA binding capability but loses Lon protease inhibitory function, resulting in a small plaque phenotype. This reciprocal pattern of functional loss further corroborates the model of distinct inhibitory domains.

Dap2 binds ExsA, inhibits ExsA binds free DNA

N79A, V52K, A64K mutants failed to binds to ExsA, thus ExsA still binds to DNA

E97A E99A double mutants still binds to ExsA

To strengthen these findings, we plan to:

1. Generate additional targeted mutations based on structural predictions
2. Conduct comprehensive phenotypic characterization
3. Specifically test the Lon-dependence of T3SS suppression through genetic and biochemical approaches

This systematic approach will provide definitive evidence regarding whether Dap2's effects on T3SS are mediated through Lon or represent an independent function.

(3) As to the point 1 of referee #1, (Biological rationale for T3SS inhibition remains unclear) Biological rationale for T3SS inhibition remains unclear. The functional significance of Dap2-mediated inhibition of the T3SS via ExsA interaction is not immediately apparent. It is unclear why phage PaoP5 would benefit from suppressing host virulence in an infection context. One possibility is that certain ExsA-regulated genes impair phage infection or replication, making T3SS activity deleterious to phage fitness. To directly test this hypothesis, the authors should compare phage fitness parameters-including burst size, efficiency of plating (EOP), and plaque morphology-between wild-type *P. aeruginosa* PAO1 and a Δ exsA mutant, using both wild-type and Δ dap2 phage strains. In addition, it would be valuable to assess whether Dap2-mediated T3SS suppression enhances phage therapeutic outcomes in vivo by comparing bacterial clearance or survival in mouse infection models using wild-type versus Δ exsA hosts. These experiments would clarify the evolutionary or functional rationale for T3SS inhibition by Dap2.

These data are presented in Supplementary Figure S4. T3SS inhibition appears to modestly enhance phage progeny production, as evidenced by the burst size of PaoP5 Δ dap2 in Δ exsA (22.08 ± 3.99 PFU/cell) compared to PAO1 (11.32 ± 1.42 PFU/cell);

P=0.0357). However, this fitness advantage is relatively minor, which is why we included it in the supplementary materials. We propose that while T3SS inhibition may provide a slight fitness benefit, the cumulative effect of multiple phage proteins with similar functions could collectively contribute to greater phage fitness.

Regarding the animal model experiments, we found that T3SS inhibition by Dap2 significantly attenuates *P. aeruginosa* virulence, similar to $\Delta exsA$ strains. Consequently, comparing phage therapy efficacy between PAO1 and $\Delta exsA$ in animal models may not yield meaningful conclusions. The biological significance of T3SS inhibition for phage fitness appears limited to a modest (~2-fold) increase in progeny production - a difference too subtle to be detected by plaque assays and unlikely to manifest in animal experiments.

“Since larger plaques require a high yield of progeny per generation, subtle differences in burst size may not be discernible through plaque morphology alone. To further explore this, we infected PAO1 and PAO1 $\Delta exsA$ with PaoP5 $\Delta dap2$. Interestingly, the burst size of PaoP5 infecting PAO1 or PAO1 $\Delta exsA$ are 108.26 ± 6.65 PFU/cel and 103.91 ± 11.08 PFU/cel, respectively, which is not significantly different (P=0.7014), the burst size of PaoP5 $\Delta dap2$ in PAO1 $\Delta exsA$ was 22.08 ± 3.99 PFU/cell compared to 11.32 ± 1.42 PFU/cell in PAO1 (P=0.0357) (Fig. S4 D), indicating that the phage's inhibition of T3SS serves as a specific strategy to reduce host energy consumption, thereby enhancing progeny production. This provides direct evidence of the fitness benefits conferred by Dap2 through T3SS suppression. However, the small plaque phenotype of PaoP5 $\Delta dap2$ was not reversed in PAO1 $\Delta exsA$, suggesting that Dap2 may also function as an anti-defense system, targeting an additional bacterial defense mechanism beyond T3SS. This dual role highlights the multifaceted nature of Dap2 in promoting phage fitness.

FIG S4

d

(b) The one-step growth curve of phage PaoP5 or PaoP5Δdap2 infecting PAO1 or ΔexsA. (a-b) Error bars indicate the mean ± SD (n=3) and statistical significance was determined using a two-sided Student's t-test.

(c) Phage PaoP5Δdap2 was mixed with PAO1 or ΔexsA, and used double-layer agar plates to observe the plaque number and size. Data are representative of three independent replicates. (d) The burst size of wild-type PaoP5 infecting PAO1 and PAO1ΔexsA was 1108.26 ± 6.65 PFU/cell and 103.91 ± 11.08 PFU/cell, respectively. In contrast, the burst size of PaoP5Δdap2 infecting PAO1 and PAO1ΔexsA was significantly reduced to 11.32 ± 1.42 PFU/cell and 22.08 ± 3.99 PFU/cell, respectively. Error bars represent the mean ± SD of three biological replicates. Statistical significance was determined using Student's t.

(4) The other minor points should be validated as suggested soon.

Overall, we are confident that most concerns can be adequately addressed, and we respectfully request the opportunity to resubmit after careful revision.

Dear Dr. Liang,

Thank you for contacting me to discuss the previous decision and for submitting a preliminary revision plan for your manuscript. I apologise for the unusual delay in replying to you due to the currently high number of manuscript submissions to our office.

I have now looked through your revision plan. I appreciate that you may be able to alleviate many of the reviewers' concerns regarding the differentiation of the role of Dap2 in regulation of T3SS vs Lon protease function. However, I also find that the outcome of the proposed experiments remains open-ended. Therefore, I do not find myself in a position to commit to inviting a revision.

Nevertheless, if you find that the outcome of these experiments substantively supports your findings, I would be willing to reconsider the manuscript, while treating it as a new submission. In this case, the manuscript would have to be re-assessed for novelty at the time of its resubmission. Depending on the added findings, I would send it back for re-assessment to the original reviewers. Due to the substantial amount of new results that would have to be added, I would allow them to make new comments on the data, which might then have to be further addressed.

I appreciate that you contacted us for further discussion of your work, and I hope that the proposed approach sounds reasonable to you.

With kind regards,

Ieva

** As a service to authors, EMBO Press provides authors with the possibility to transfer a manuscript that one journal cannot offer to publish to another EMBO publication or the open access journal Life Science Alliance launched in partnership between EMBO Press, Rockefeller University Press and Cold Spring Harbor Laboratory Press. The full manuscript and if applicable, reviewers' reports, are automatically sent to the receiving journal to allow for fast handling and a prompt decision on your manuscript. For more details of this service, and to transfer your manuscript please click on Link Not Available. **

Dear editor and reviewers:

Thank you for the good comments concerning our manuscript entitled "Bacteriophage Dap2 inhibits bacterial T3SS and synergizes with Dap1 to evade anti-phage immunity" (ID: EMBOJ-2025-121748). We have carefully addressed your concerns and provide our point-by-point responses below.

Editor concerns:

As you will see, while reviewers #1 and #3 appreciate the findings, reviewer #2 finds that the broader novelty of the study is limited by the previous publication from your team characterizing the role of Dap1 in Lon inhibition.

We sincerely appreciate the reviewer's positive feedback and insightful comments. We agree that the inhibition of Lon by Dap2 shares some conceptual overlap with previous findings involving Dap1. However, the underlying mechanisms are distinct: whereas Dap1 protects the HNH domain from Lon-mediated degradation, Dap2 directly binds to and inhibits Lon itself.

Moreover, the Lon inhibition by Dap2 represents only one aspect of our study's novelty. Our work further reveals that: (1) Dap2 suppresses T3SS activity, expanding its functional repertoire beyond Lon inhibition. (2) Dap2 collaborates with Dap1 to enhance anti-defense activity, a synergistic interaction that aligns with observations from other reviewers and provides new insights into phage counter-defense strategies.

Together, these findings significantly advance our understanding of phage-bacterial interactions and underscore the multifaceted nature of phage-encoded defense inhibitors.

Furthermore, all reviewers find that the proposed role of Dap2 in regulation of T3SS activity needs further substantiation and differentiation from its effect on Lon protease, and that the current characterisation of these interactions is rather preliminary. Additionally, reviewer #1 raises an important point that the effect of Dap1 on T3SS regulation could be mediated indirectly via Lon inhibition.

We agree with the reviewers that elucidating how Dap2 regulates both T3SS and Lon (as raised in point 3 of reference #1 and point 1 of reference #2) represents a key mechanistic question. In particular, determining whether T3SS suppression is Lon-dependent (point 2 of reviewer #1) warrants careful investigation. These are excellent suggestions. We have done lots of experiments to address this question.

Table: The experiments to prove that Dap2 inhibits ExsA and Lon through two distinct domains

Dap2	Keys residues for binding to ExsA				Keys residues for binding to Lon	
Mutation	V52K	A64K	N79A	Mutation	E97A	E99A
Plaque Size	PaoP5Δdap2 forms Large plaques in PAO1::Dap2 V52K, A64K, and N79A, indicating the mutant Dap2 still inhibits Lon (FIG 4b,c)			Plaque Size	PaoP5Δdap2 forms small plaques in PAO1::Dap2 E97A, E99A, indicating the mutant Dap2 failed to inhibit Lon (FIG 4b,c)	
EMSA	Failed to inhibit the binding of ExsA to DNA, indicating these residues are essential for binding to ExsA (FIG 4f, g)			EMSA	Still inhibit the binding of ExsA to DNA, indicating these residues are nonessential for binding to Lon (FIG 4h)	
RT-QPCR	Triple mutant failed to inhibit the expression of T3SS genes (FIG 1f)			BACTCH	White, failed to bind to Lon (FIG 4d)	
				Mice infection	Mice infected with PAO1::dap2E97A or PAO1::dap2E99A survived, indicating the inhibition of bacterial virulence is independent of Lon (FIG 4e)	

Figure 4

Dap2 inhibits ExsA and Lon through two distinct domains

To investigate the molecular mechanisms by which Dap2 interacts with ExsA and Lon, we first predicted the structure of Dap2 and ExsA using AlphaFold3. Structures of NTD and CTD of ExsA could be well superimposed with its crystal structure (3OIO) and homolog (4ZUA), respectively, with the overall structure showing several possible conformations (indicated with C1, C2 and C3). The prediction reveals that the inhibition of ExsA and Lon protease is mediated by separate domains of Dap2. Residues critical for ExsA inhibition (red; positions 52, 64, and 79) are spatially separated from those involved in Lon protease inhibition (magenta; positions 97 and 99), with these functional sites located on opposite surfaces of the protein (Fig. 4A).

We then constructed five point mutations in Dap2. In plaque assays, *PaoP5* Δ *dap2* formed large plaques on PAO1 expressing Dap2(V52K), Dap2(A64K), or Dap2(N79A), but small plaques on strains expressing Dap2(E97A) or Dap2(E99A). This indicates that the V52K, A64K, and N79A mutants retain the ability to inhibit Lon, whereas the E97A and E99A mutants lose this function, suggesting that E97 and E99 residues are essential for Lon inhibition (Fig. b,c). This conclusion was further supported by BACTCH assays, in which E97A and E99A mutant Dap2 failed to bind Lon (Fig. 4d).

To assess whether Lon inhibition contributes to the reduced virulence of PAO1::Dap2(48), we infected mice with PAO1::Dap2(E97A) or PAO1::Dap2(E99A). All infected mice survived, indicating that the attenuation of T3SS-dependent virulence is mediated through ExsA inhibition, not through Lon (Fig. 4e).

We further performed EMSA assays to evaluate the effect of Dap2 mutants on ExsA DNA-binding activity. The N79A, V52K, and A64K mutants lost the ability to inhibit ExsA, allowing ExsA to bind DNA (Fig. 4f,g), whereas the E97A and E99A mutants still suppressed ExsA binding (Fig. 4h). These results confirm that N79, V52, and A64—but not E97 or E99—are essential for inhibiting ExsA .

This systematic approach will provide definitive evidence that Dap2's effects on T3SS and Lon are mediated through two independent domains.

Referee #1:

The manuscript titled "Bacteriophage Dap2 inhibits bacterial T3SS and synergizes with Dap1 to evade anti-phage immunity" characterizes the dual function of the Dap2 protein from *Pseudomonas aeruginosa* phage PaoP5. First, Dap2 binds to the T3SS regulator ExsA, leading to transcriptional downregulation of T3SS-related genes and attenuated bacterial virulence in a mouse infection model. Second, Dap2 functions as an anti-defense protein by inhibiting the bacterial Lon protease, preventing degradation of a phage-encoded HNH endonuclease critical for efficient genome packaging. The authors further demonstrate that Dap2 acts synergistically with Dap1, a previously identified HNH-binding protein, to fully neutralize Lon protease-mediated anti-phage defense. The experiments are well-executed, and the findings are interesting. This study highlights the importance of coordinated anti-defense strategies, such as the Dap1/Dap2 system, in enhancing phage fitness and informing the development of engineered phage therapeutics.

To further strengthen the manuscript, I offer the following major and minor suggestions:

Thank you so much for your positive comments!

Major Suggestion:

1. Biological rationale for T3SS inhibition remains unclear. The functional significance of Dap2-mediated inhibition of the T3SS via ExsA interaction is not immediately apparent. It is unclear why phage PaoP5 would benefit from suppressing host virulence in an infection context. One possibility is that certain ExsA-regulated genes impair phage infection or replication, making T3SS activity deleterious to phage fitness. To directly test this hypothesis, the authors should compare phage fitness parameters—including burst size, efficiency of plating (EOP), and plaque morphology—between wild-type *P. aeruginosa* PAO1 and a Δ exsA mutant, using both wild-type and Δ dap2 phage strains.

Thank you so much for your suggestions. We included these data in Supplementary Figure S4. T3SS inhibition appears to modestly enhance phage progeny production, as evidenced by the burst size of PaoP5 \$\Delta\$ dap2 in \$\Delta\$ exsA (22.08 \$\pm\$ 3.99 PFU/cell) compared to PAO1 (11.32 \$\pm\$ 1.42 PFU/cell; P=0.0357). However, this fitness advantage is relatively minor, which is why we included it in the supplementary materials. We propose that while T3SS inhibition may provide a slight fitness benefit, the cumulative effect of multiple phage proteins with similar functions could collectively contribute to greater phage fitness.

Regarding the animal model experiments, we found that T3SS inhibition by Dap2 significantly attenuates *P. aeruginosa* virulence, similar to \$\Delta\$ exsA strains. Consequently, comparing phage therapy efficacy between PAO1 and \$\Delta\$ exsA in animal models may

not yield meaningful conclusions. The biological significance of T3SS inhibition for phage fitness appears limited to a modest (~2-fold) increase in progeny production - a difference too subtle to be detected by plaque assays and unlikely to manifest in animal experiments.

Since larger plaques require a high yield of progeny per generation, subtle differences in burst size may not be discernible through plaque morphology alone. To further explore this, we infected PAO1 and PAO1 Δ exsA with PaoP5 Δ dap2. Interestingly, the burst size of PaoP5 infecting PAO1 or PAO1 Δ exsA are 108.26 ± 6.65 PFU/cel and 103.91 ± 11.08 PFU/cel, respectively, which is not significantly different ($P=0.7014$), the burst size of PaoP5 Δ dap2 in PAO1 Δ exsA was 22.08 ± 3.99 PFU/cell compared to 11.32 ± 1.42 PFU/cell in PAO1 ($P=0.0357$) (Fig. S4 D), indicating that the phage's inhibition of T3SS serves as a specific strategy to reduce host energy consumption, thereby enhancing progeny production. This provides direct evidence of the fitness benefits conferred by Dap2 through T3SS suppression. However, the small plaque phenotype of PaoP5 Δ dap2 was not reversed in PAO1 Δ exsA, suggesting that Dap2 may also function as an anti-defense system, targeting an additional bacterial defense mechanism beyond T3SS. This dual role highlights the multifaceted nature of Dap2 in promoting phage fitness.

Fig. S4 (b) The one-step growth curve of phage PaoP5 or PaoP5 Δ dap2 infecting PAO1

or Δ exsA. (a-b) Error bars indicate the mean \pm SD (n=3) and statistical significance was determined using a two-sided Student's t-test.

(c) Phage PaoP5 Δ dap2 was mixed with PAO1 or Δ exsA, and used double-layer agar plates to observe the plaque number and size. Data are representative of three independent replicates.

(d) The burst size of wild-type PaoP5 infecting PAO1 and PAO1 Δ exsA was 1108.26 ± 6.65 PFU/cell and 103.91 ± 11.08 PFU/cell, respectively. In contrast, the burst size of PaoP5 Δ dap2 infecting PAO1 and PAO1 Δ exsA was significantly reduced to 11.32 ± 1.42 PFU/cell and 22.08 ± 3.99 PFU/cell, respectively. Error bars represent the mean \pm SD of three biological replicates. Statistical significance was determined using Student's t.

In addition, it would be valuable to assess whether Dap2-mediated T3SS suppression enhances phage therapeutic outcomes in vivo by comparing bacterial clearance or survival in mouse infection models using wild-type versus Δ exsA hosts. These experiments would clarify the evolutionary or functional rationale for T3SS inhibition by Dap2.

Thanks for your suggestions! Assessing whether Dap2-mediated T3SS suppression enhances phage therapeutic outcomes in vivo is indeed valuable. However, in our mouse infection model, even at a high dose (8×10^6 PFU), all mice survived infection with the Δ exsA strain, as the disruption of T3SS significantly attenuates its virulence. This makes it challenging to compare the effects of WT phage versus Dap2 mutant phage in the context of Δ exsA-infected mice.

Fig: Δ exsA is significantly attenuated so it is not possible to compare the phage therapeutic efficacy in PAO1 and PAO1 Δ exsA

2. Potential indirect effect of Lon inhibition on T3SS. Previous studies have shown that Lon protease can influence T3SS regulation in *P. aeruginosa* (e.g., PMID: 29362236, 28509355, 27657922). Since Dap2 inhibits Lon to protect the HNH endonuclease, the observed T3SS inhibition may result from Lon inactivation rather than direct Dap2-ExsA interaction. To differentiate between these possibilities, the authors should construct a Dap2 mutant (Dap2mut) that fails to bind Lon but retains ExsA binding. Expressing this mutant in *P. aeruginosa* and assessing virulence phenotypes in a mouse infection model will clarify whether Lon inhibition is responsible for T3SS suppression. If Lon inhibition is dispensable for the observed effect, Dap2mut and wild-type Dap2 should yield similar phenotypes.

3. Mechanistic basis of Lon inhibition by Dap2 is insufficiently defined. It remains unclear how Dap2 binds Lon and inhibits its proteolytic activity. Does Dap2 inhibit Lon globally or selectively block degradation of the HNH endonuclease? The authors should consider molecular docking analyses and introduce site-directed mutations in Dap2 and/or Lon to map the interaction interface. These mutants should be assessed via binding assays, plaque assays, and EOP assays to determine whether loss of interaction correlates with reduced HNH protection and phage fitness.

Thanks for your suggestions! We cited these papers and performed extensive assays to prove the detailed mechanism.

Table: The experiments to prove that Dap2 inhibits ExsA and Lon through two distinct domains

Dap2	Keys residues for binding to ExsA				Keys residues for binding to Lon	
Mutation	V52K	A64K	N79A	Mutation	E97A	E99A
Plaque Size	PaoP5Δdap2 forms Large plaques in PAO1::Dap2 V52K, A64K, and N79A, indicating the mutant Dap2 still inhibits Lon (FIG 4b,c)			Plaque Size	PaoP5Δdap2 forms small plaques in PAO1::Dap2 E97A, E99A, indicating the mutant Dap2 failed to inhibit Lon (FIG 4b,c)	
EMSA	Failed to inhibit the binding of ExsA to DNA, indicating these residues are essential for binding to ExsA (FIG 4f,g)			EMSA	Still inhibit the binding of ExsA to DNA, indicating these residues are nonessential for binding to Lon (FIG 4h)	
RT-QPCR	Triple mutant failed to inhibit the expression of T3SS genes (FIG 1f)			BACTCH	White, failed to bind to Lon (FIG 4d)	
				Mice	Mice infected with PAO1::	

		infection	dap2E97A or PAO1 : : dap2E99A survived, indicating the inhibition of bacterial virulence is independent of Lon (FIG 4e)
--	--	-----------	---

Figure 4

Dap2 inhibits ExsA and Lon through two distinct domains

To investigate the molecular mechanisms by which Dap2 interacts with ExsA and Lon, we first predicted the structure of Dap2 and ExsA using AlphaFold3. Structures of NTD and CTD of ExsA could be well superimposed with its crystal structure (3OIO) and homolog (4ZUA), respectively, with the overall structure showing several possible conformations (indicated with C1, C2 and C3). The prediction reveals that the inhibition of ExsA and Lon protease is mediated by separate domains of Dap2. Residues critical for ExsA inhibition (red; positions 52, 64, and 79) are spatially separated from those involved in Lon protease inhibition (magenta; positions 97 and 99), with these functional sites located on opposite surfaces of the protein (Fig. 4A).

We then constructed five point mutations in Dap2. In plaque assays, *PaoP5Δdap2* formed large plaques on PAO1 expressing Dap2(V52K), Dap2(A64K), or Dap2(N79A), but small plaques on strains expressing Dap2(E97A) or Dap2(E99A). This indicates that the V52K, A64K, and N79A mutants retain the ability to inhibit Lon, whereas the E97A and E99A mutants lose this function, suggesting that E97 and E99 residues are essential for Lon inhibition (Fig. b,c). This conclusion was further supported by BACTCH assays, in which E97A and E99A mutant Dap2 failed to bind Lon (Fig. 4d).

To assess whether Lon inhibition contributes to the reduced virulence of PAO1::Dap2(48), we infected mice with PAO1::Dap2(E97A) or PAO1::Dap2(E99A). All infected mice survived, indicating that the attenuation of T3SS-dependent virulence is mediated through ExsA inhibition, not through Lon (Fig. 4e).

We further performed EMSA assays to evaluate the effect of Dap2 mutants on ExsA DNA-binding activity. The N79A, V52K, and A64K mutants lost the ability to inhibit ExsA, allowing ExsA to bind DNA (Fig. 4f,g), whereas the E97A and E99A mutants still suppressed ExsA binding (Fig. 4h). These results confirm that N79, V52, and A64—but not E97 or E99—are essential for inhibiting ExsA .

Furthermore, a known Lon substrate should be included in the degradation assay to

evaluate whether Dap2 inhibition is substrate-specific or broadly impairs Lon activity.

Thanks for your suggestions! We included this data in FIG S8.

To determine whether Dap2-mediated inhibition is substrate-specific or broadly impairs Lon activity, we also incubated Lon with RhII in the presence or absence of Dap2. We found that Dap2 could inhibit Lon-mediated degradation of RhII (Fig. S8). These results demonstrate that Dap2 exerts a broad inhibitory effect on Lon, rather than being specific to the phage protein HNH.

Minor Suggestion:

1. Line 360: "Figure 6b" should be corrected to "Figure 5?".

Thanks for your suggestions! We corrected it.

2. Figure 1b: The upper blot is labeled as anti-Flag, but based on the figure legend, it should be labeled anti-ExoS.

Thanks for your suggestions! We replaced anti-Flag with anti-ExoS.

3. Figure 1d: The pull-down results are difficult to interpret. The input levels of His-

ExsA appear unequal between samples. The observed reduction in pull-down signal may simply reflect lower input of His-ExsA rather than a difference in binding. Please ensure comparable input levels or normalize accordingly.

Thanks for your suggestions! We repeated this experiment and added equal input levels of His-ExsA and made the correction in Fig1d.

4. Figure 1e and Lines 194-198: For readers unfamiliar with ExsA, please provide additional details on its conformational changes. Also, clearly indicate the locations of the DNA-binding domain and Dap2-binding site on the ExsA structure.

Thanks for your suggestions!! We made corrections and clearly indicated these locations.

“FIG 1e: ExsA-Dap2 complex structure model and the details about interaction interface. Electrostatic potential surfaces of ExsA and Dap2 are basically

complementary (bottom right). Different conformations of ExsA were marked with C1 to C3 (upper left). Amino acid residues that, upon directed mutagenesis, affect ExsA-DnaP binding are labeled in red; those affecting ExsA-Lon binding are labeled in magenta. HTH domain referring to DNA recognition and binding was colored in green.”

5. Figure 2d: The x-axis label is missing.

Thanks for your suggestions! We added the label in Figure 2d.

6. Figure 3d: Ensure consistent nomenclature throughout the manuscript. Clarify whether Orf050 refers to the HNH endonuclease (hnh), and standardize the naming accordingly.

Thanks for your suggestions! We used HNH throughout the figures.

7. Figure 4c: Please include statistical analyses (e.g., error bars, p-values) to support the conclusions.

As suggested, we have incorporated error bars and p-values, calculated by 2way ANOVA, into the figure (FIG 5C).

8. Figure 4f: Quantify the degree of HNH degradation in the protease assay and include this quantification in the figure.

Thank you for your suggestions. We quantified the extent of HNH degradation using ImageJ; however, given the large number of figures already included, we felt that the degradation of HNH was clearly visible in the gel image and decided not to include the quantification data in the manuscript.

9. Figure 4f: In the third gel panel, the Dap1 label points to a band that is inconsistent with the corresponding band in the first gel. Please clarify the labeling and ensure consistency.

Thanks, we made the corrections.

10. Lines 280-284: The strain PAO1/p-hnh is discussed without prior introduction. Please provide context and experimental rationale when first mentioned.

Thank you so much for your suggestions, we made the correction.

“To test this hypothesis, we infected wild-type PAO1, Δlon , $\Delta lon/p-lon$, and *hnh*-overexpression strain PAO1/p-hnh with PaoP5 $\Delta dap2$. The results showed that PaoP5 $\Delta dap2$ formed larger plaques in Δlon or PAO1/p-hnh compared to PAO1 (Fig. 3d-e)”

Referee #2:

During bacteriophage infection, factors produced by the phage modulate activities of host processes, including transcription, translation, anti-phage immunity, and virulence. The authors of this manuscript previously discovered that Dap1, a protein produced by Pseudomonas phage PaoP5, prevents degradation by host Lon protease of a phage nuclease HNH, which is required for efficient packaging of phage genomes into nascent capsids.

Here, the authors identify a second factor (Dap2) in the same phage that directly inhibits Lon-mediated degradation and is also required for HNH protection. Separately from this activity, they find that Dap2 interacts with ExsA, a transcriptional activator of the T3SS required for Pseudomonas virulence in mammals. They go on to show that phage-treatment of mice infected with Pseudomonas have enhanced survival in a Dap1/Dap2-dependent manner.

The major shortcomings of the manuscript are that (i) the findings are relatively preliminary and need further substantiation, see details below, and (ii) they do not make a very significant advance over those previously reported by these authors: that Dap1 protects HNH from Lon-mediated degradation.

We sincerely appreciate the insightful comments. We did additional works to substantially enhance the strength of the findings, as listed below.

We fully agree that the inhibition of Lon by Dap2 shares some conceptual overlap with previous findings involving Dap1. However, the underlying mechanisms are distinct:

whereas Dap1 protects the HNH domain from Lon-mediated degradation, Dap2 directly binds to and inhibits Lon itself.

Moreover, the Lon inhibition by Dap2 represents only one aspect of our study's novelty. Our work further reveals that: 1、 Dap2 suppresses T3SS activity, expanding its functional repertoire beyond Lon inhibition. 2、 Dap2 collaborates with Dap1 to enhance anti-defense activity, a synergistic interaction that aligns with observations from other reviewers and provides new insights into phage counter-defense strategies.

Together, these findings significantly advance our understanding of phage-bacterial interactions and underscore the multifaceted nature of phage-encoded defense inhibitors.

Detailed critiques:

1. The paper makes an abrupt pivot from the study of the role of Dap2 in T3SS inhibition to the study of Dap2 in Lon inhibition. The two activities appear genetically separable, but it is unclear whether the same parts of the Dap2 protein are responsible for inhibiting EsxA and Lon.

Thanks for your suggestions! Please see the response to reviewer 1.

2. Does Dap2 affect the transcriptome in the absence of EsxA and/or Lon?

Thank you for these interesting suggestions. Given that the deletion of EsxA and Lon significantly perturbs global gene expression, we feel that an RNA-seq study in this background would be challenging to interpret and unlikely to yield definitive conclusions. The primary goal of the current study is to establish the dual function of Dap2 in directly inhibiting EsxA and Lon. Therefore, we plan to address the broader transcriptomic implications in a future study.

3. Is T3SS expression affected during phage infection? It seems as though all the related analyses were performed with plasmid expression of Dap2.

Thanks for your suggestions, it is very important to test whether the inhibition of T3SS expression is true during phage infection.

“To further determine whether T3SS genes are suppressed during phage infection, we infected PAO1 with either phage PaoP5 or the PaoP5 Δ dap2 mutant and extracted total RNA at time points between 1 and 30 minutes post-infection. Quantitative analysis revealed that expression of key T3SS genes (exsC,exoS) was significantly downregulated in PAO1 infected with wild-type PaoP5, but not in the PaoP5 Δ dap2-

infected group (Fig. S1, D-E). These results confirm that T3SS suppression occurs during phage infection and is dependent on Dap2.”

4. The levels of HNH (and preferably additional known Lon substrates as well) during phage infection need to be measured to substantiate the conclusion that Dap2 protects HNH from Lon-mediated degradation in vivo.

We appreciate this suggestion. Investigating HNH levels in vivo is indeed an interesting idea. To that end, we attempted to construct phages with N-terminal or C-terminal HIS-tagged HNH using our well-established PaoP5 genome engineering system. Despite extensive efforts, we were unable to obtain recombinant phages with either tag. We therefore regret that we could not directly quantify HNH protein levels during infection. Nonetheless, our genetic and phenotypic data (Fig. 3D), combined with the in vitro protein degradation assay (Fig. 5F), robustly support the conclusion that LON mediates the degradation of HNH.

Fig: protocol to engineer phages with the *hnh* gene fused to 5' HIS or 3' HIS

5. Though the TEM is suggestive of a packaging defect in $\Delta dap2$ phage, it is not a conclusive demonstration that these phages are devoid of DNA. DNA extracted from capsids should be quantified by qPCR or similar methodology. Also, even if the phages lack DNA, it could stem from inappropriate genome injection rather than inefficient packaging. These possibilities have not been distinguished by the existing analyses. As suggested, we extracted total DNA from lysates of PAO1 infected with either PaoP5 or PaoP5 $\Delta dap2$. Using quantitative PCR (qPCR) with primers targeting the phage gene *orf057*, we observed a significantly lower level of phage DNA in the PaoP5 $\Delta dap2$ -infected group compared to the wild-type PaoP5 group, which is consistent with our expectations.

Minor:

1. The name "Dap2" stands for Defense anti-phage 2. This is not a very logical name for this protein, since it is not a defense or anti-phage protein but an inhibitor thereof.

Thank you for raising this point. The name "Dap2" was assigned because this gene is located next to dap1 in the genome, following our previous nomenclature. We agree that the name "Defense anti-phage 2" is not an ideal descriptor for an inhibitor, and we have added clarification in the manuscript to prevent any misunderstanding.

“ Since this ORF (orf004) is located adjacent to the dap1 gene, we named it dap2 to maintain a consistent nomenclature based on genomic context. “

2. Fig 1b: exoS-flag should be labeled.

We replaced anti-Flag with anti-ExoS.

3. Fig 1d: why is His-ExsA missing from the input? It is difficult to make a conclusion about the finteraction without this control. Considering that Dap2 reduces ExsA levels in Fig. 1b, there should be less ExsA in the Dap2 input, not more.

Thanks! We repeated this experiment and added equal input levels of His-ExsA and made the correction in Fig1d.

4. Line 179-180: This conclusion cannot be made with confidence, because in the Δ exsA background, there is little or no exsC transcription for Dap2 to inhibit.

Thanks for your suggestions, we agree that there is no exsC transcription in Δ exsA background, so we deleted this sentence.

5. Fig S3c-d: His-ExsA purification has a substantial number of contaminating bands. It would help if a nonspecific DNA substrate of identical length were included to confirm the observed binding to the ExsC promoter is specific.

Thanks for your suggestions, we tried several times to purify ExsA, but the contaminating bands still persist. While our ExsA preparations were not entirely pure, we have multiple lines of evidence supporting the specificity of its binding. The well-characterized function of ExsA (Brutinel et al., 2008) is corroborated by our internal controls: the inability of a Dap2 mutant to disrupt the ExsA-DNA interaction (Fig. 4f) and functional validation via RT-qPCR (Fig. 1f). Together, these data confirm that the binding to the exsC promoter is specific and is not compromised

6. Fig 1f: is the Dap2 triple mutant stable in vivo?

The plasmid PME6032 is a widely used plasmid in *P. aeruginosa* and is quite stable in *P. aeruginosa*. We selected 5 colonies from each transfer, and the Dap2 triple mutant

is stable in vivo.

Referee #3:

General Summary and Opinion

This manuscript reports the characterization of Dap2, a protein encoded by *Pseudomonas aeruginosa* phage PaoP5, which inhibits the bacterial type III secretion system (T3SS) by binding the master regulator ExsA and simultaneously inhibits Lon protease to protect the phage-encoded HNH endonuclease. Dap2 and its neighboring gene product Dap1, previously shown to bind HNH directly, together provide full protection from Lon-mediated degradation and enable efficient phage replication. The authors demonstrate that the Dap1/Dap2 pair enhances phage fitness and therapeutic efficacy in a murine infection model.

Overall, the work is carefully executed and addresses an important question in phage-host interactions. The dual role of Dap2 in both virulence suppression and anti-defense is interesting, and the mechanistic dissection of Dap2's interaction with ExsA and Lon adds insight into phage counter-defense strategies. However, the rationale for why the authors targeted virulence phenotypes, such as T3SS, is poorly developed, and several experimental details require clarification or quantification.

Thank you so much for your positive comments!

We introduced the importance of phage derived anti-virulence studies and pointed out that T3SS is a very important virulence factor but phage proteins inhibiting T3SS is not well studied while inhibiting other virulence had been investigated, thus we tried to identify phage protein that inhibit T3SS. Othe other experimental details are clarified in the following response.

Major Concerns

1. Clarify the study rationale and connection to virulence modulation

The stated goal (line 112) is to identify phage-encoded virulence modulators, yet the

introduction is largely focused on anti-phage defense. The rationale for why phages would inhibit virulence systems like T3SS is not well articulated. The discussion (line 391) admits this biological significance is unclear. Given that virulence inhibition does not appear to meaningfully increase phage yield (e.g., Fig. S4d), a clearer argument should be made for why the authors pursued this hypothesis and how it connects to phage fitness.

Thank you for your suggestions, we described the rationale for identifying phage inhibitors of T3SS in introduction.

“However, whether phages can regulate the type III secretion system (T3SS) remains unclear. T3SSs are critical virulence determinants in many pathogenic bacteria, enabling them to inject effector proteins directly into host cells and establish trans-kingdom interactions(19). Given their essential role in infection, T3SSs are attractive targets for novel therapeutics and vaccines(20). Unlike conventional antibiotics, T3SS inhibitors may reduce selective pressure for resistance since they disrupt virulence rather than bacterial growth. A deeper structural and functional understanding of T3SSs and their phage protein inhibitors will advance mechanism-based drug development(20, 21).”

We also discussed the effect of inhibiting T3SS for the fitness of phage in discussion.

“The modest increase in viral yield implies that inhibiting the T3SS may represent a specific phage strategy to conserve host resources, thereby enhancing replication efficiency. Thus, while the effect of a single gene is limited, the cumulative contribution of multiple such phage genes could substantially enhance phage fitness—a possibility that requires further investigation.”

2. Background on *Pseudomonas aeruginosa*

Given the central role of this pathogen, a brief overview of its relevance, especially in the context of phage therapy, T3SS, and intrinsic defenses like Lon, would benefit readers less familiar with the system.

Thanks! we added the description in introduction.

“*Pseudomonas aeruginosa* is an opportunistic pathogen capable of causing severe nosocomial infections, and it represents a major cause of morbidity and mortality in patients with cystic fibrosis (1, 2). The pathogenicity of *P. aeruginosa* is coordinated by a range of virulence factors, among which the type III secretion system (T3SS) serves as a crucial contact-dependent apparatus for injecting effector proteins and plays an essential role in acute infections (3, 4). The expression of this highly energy-consuming

system is tightly regulated by the transcriptional activator ExsA (5). With the increasing prevalence of multidrug-resistant strains, phage therapy is a potential alternative to antibiotics (6-9). A deeper understanding of the molecular mechanisms underlying phage infection and bacterial resistance is expected to provide a theoretical foundation for designing more rational phage-based therapeutic strategies (10).”

3. Clarify luminescence readout (line 142)

The assay using the *exsC* promoter-lux fusion should be described as a proxy for transcriptional activation of T3SS genes, not T3SS activity per se. Similarly, clarify whether the *exoS* promoter reporter (line 179-180) is analogous.

In T3SS research, a well-established and reliable method involves the use of promoters from key regulatory elements, such as the *exsA* or *exsC* promoter from *P. aeruginosa*, to construct reporter gene fusion systems. This is because the activity of such a promoter is directly controlled by the master regulator protein ExsA, and the activity of ExsA is positively correlated with the expression level of the entire T3SS gene cluster (McCaw et al., PNAS 2002).

“*exoS* promoter reporter” (line 179-180) is a typo, it is the *exsC* promoter, which is widely used to construct reporter gene fusion systems.

4. In vivo plasmid retention (Fig. 1g)

The dramatic reduction in mouse mortality with Dap2-expressing strains implies in vivo repression of T3SS. However, the authors should confirm that plasmids were retained and consider the possibility that plasmid loss during infection could have affected the results were retained and actively expressed during infection. Was plasmid maintenance tested or monitored in vivo? Were bacterial burdens in tissue quantified in this experiment?

Thank you for your suggestions, we collected strains in the phage therapy experiment, and we tested 23 colonies isolated from the liver of bacteria infected mice, and the presence of plasmid was confirmed by PCR, so the plasmid is stable in vivo.

The bacteria burdens in liver were quantified (Fig 6e)

5. Quantify plaque size and clarify EOP (Fig. 2)

Data related to plaque morphology and size (e.g., Figs. 2a, 2b, 4a) are used to make claims of impaired fitness, but these should be quantified using plaque diameter or area measurements and statistical analysis. In Fig. 2b, dilution spot assays are shown, but these are not equivalent to EOP values. The authors should include proper EOP calculations (e.g., PFU on test strain divided by PFU on reference strain).

We appreciate the reviewer's suggestion regarding plaque quantification. Indeed, we have performed these measurements using ImageJ. Ultimately, we elected not to include the quantification data to maintain a concise presentation, given the large volume of figures already in the manuscript.

6. Introduce HNH earlier

HNH is referenced multiple times before its function is explained (lines 276-280). A brief introduction to HNH as a genome packaging endonuclease and Lon target earlier in the manuscript would improve clarity.

We introduced HNH in the fourth section of the results.

7. Delayed burst time not apparent (Fig. 4c)

The claim that PaoP5Δdap1Δdap2 shows delayed burst timing needs clarification. The growth curves in Fig. 4c do not obviously support this. Please define the criteria for assessing burst time and describe the basis for this conclusion.

Thanks for pointing it out, we agree that the phages did not show a significance of delayed burst time, we deleted this description in the manuscript.

“the one-step growth curve demonstrated a markedly reduced burst size for PaoP5Δdap1Δdap2 (Fig. 4c).”

8. Fig. 4f gel labeling

The gel images in Fig. 4f are not clearly labeled, making interpretation difficult. Labeling each gel and clarifying what the asterisk indicates would help. The identity of bands, especially HNH, should be explicitly marked in each gel.

Thanks! We made the corrections.

9. Phylogenetic tree scale (Fig. 5a)

The tree lacks a scale bar. Readers cannot assess phylogenetic distance or confidence without it.

We added a scale bar in the Phylogenetic Tree (fig 6a in this version).

10. Figure citation correction (line 360)

The reference to Fig. 6b should be corrected to Fig. 5c.

Thanks. We corrected it.

11. Statistical analysis missing (Fig. 5d,e)

No statistical tests are reported for the phage/bacterial titers in mouse liver. This is critical for evaluating the claim of differential therapeutic efficacy.

Thanks. We made corrections.

12. Expand discussion with field context

The discussion primarily rephrases results without situating them in broader context and is also redundant with material presented in the introduction. It would be valuable to speculate why phages might inhibit virulence regulators like ExsA or T3SS—perhaps to reduce host inflammation, slow immune clearance, or alter bacterial physiology in ways that promote phage spread. The authors should also consider broader implications of cooperative anti-defense systems in phage evolution.

Thank you for your suggestions. The mechanism by which phages inhibit virulence regulators such as ExsA or the T3SS is indeed quite interesting. However, we have not yet reached a definitive conclusion on this matter. We have included the following discussion in the text.

“Burst size analysis demonstrated that PaoP5Δdap2 produced approximately 1.95-fold more progeny when infecting ΔexsA mutants compared to wild-type PAO1 (Fig S4d). The modest increase in viral yield implies that inhibiting the T3SS may represent a specific phage strategy to conserve host resources, thereby enhancing replication efficiency. Thus, while the effect of a single gene is limited, the cumulative contribution of multiple such phage genes could substantially enhance phage fitness—a possibility that requires further investigation.”

Minor Concerns

ADS abbreviation: Consider avoiding this abbreviation unless space is a concern—it adds unnecessary jargon.

We thank the reviewer for this suggestion. We have used the abbreviation "ADS" (anti-

defense system) throughout the manuscript for brevity, as it is a well-established term in the field (e.g., *Nature Reviews Genetics* 25, 237-254, 2024) and appears over 30 times in the text. However, to improve readability for a broad audience, we have now spelled out the term in its first occurrence and added the abbreviation in parentheses for subsequent uses.

Line 108: Include a reference when *Acb2* is first mentioned as an example of partial defense activity.

Thanks. We added the reference after the first mention of *Acb2*.

Line 147: Clarify that the comparison is between *Dap2* expression and an empty vector control.

Thanks for pointing this out.

"we analyzed the promoter activity of key T3SS effector genes (*exoS*, *exoY*, and *exoT*) in WT PAO1/pME6032 versus PAO1 harboring pME6032-*dap2*."

Line 241: Refer readers to the figure panel where the relevant plaque images or data are shown.

Thanks for pointing this out.

"Next, we employed the CRISPR-Cas9 system to delete the *dap2* gene in phage PaoP5, with the knockout another phage gene (*orf014*) which does not affect the plaque formation as a negative control (Fig. 2a)."

Line 266: "Degradation on" should be revised to "degradation by."

We made the correction.

Line 286: The claim about defective packaging in Δ *dap2* lysates should be supported by a figure reference.

We added Fig. 3f after this sentence.

Line 373: Refer readers to the figure/data showing phage titers in liver and related outcomes.

Thanks for pointing this out. we added Fig. 6d after this sentence.

Suggestions

Consider analyzing whether co-expression of Dap2 and Dap1 offers any fitness advantage in strains lacking Lon. This could help disentangle whether their function is specific to Lon inhibition or broader.

We tested the phenotype of *PaoP5Δdap1Δdap2* in Δlon , and could not fully restore the phenotype of size plaque and successful genome packaging, so even co-expression of Dap2 and Dap1 might not be able to provide more significant phenotype and more fitness advantage.

Discuss whether the Dap1/Dap2 pairing is representative of a broader strategy, and whether other such synergistic/overlapping systems exist in known phages.

Thanks for your suggestions and it is very interesting to test whether the Dap1/Dap2 pairing is representative of a broader strategy, we did identify this gene pair in 67 phage genomes through BLAST, however, due to the huge diversity of phage genomes, such synergistic/overlapping systems might exist in other phages which needs further research in the near future.

Dear Haihua,

Thank you for submitting the revised version of your manuscript to The EMBO Journal. The study has now been seen by two of the original referees, who find that the revisions have significantly improved the manuscript, but also indicate a few remaining technical and data presentation aspects that would need to be addressed in the revised version.

Additionally, there are a few editorial and formatting points that need to be implemented in the final version:

1. As part of the EMBO Press transparent editorial process, The EMBO Journal will publish online a Peer Review File to accompany accepted manuscripts. This file will be published in conjunction with your paper and will include the anonymous referee reports, your point-by-point response and all pertinent correspondence relating to the manuscript, including decision letters. Please note that the Author Checklist will be published at the end of the Peer Review File. Please let us know if you want to remove or not any figures or data from the Peer Review File prior to publication. Please note that retaining unpublished data in the Peer Review File means that these count as published and that the Peer Review File would need to be referenced in future publications.
2. Please ensure that the funding information is correct and identical both in the manuscript and our online system.
3. At EMBO Press we ask authors to provide source data for the main manuscript figures. You will receive a separate email with instructions for providing source data with your revised manuscript, including how to upload and organise the files.
4. Please remove all figures from the manuscript text, while leaving in the figure legends.
5. Please update the nomenclature of Expanded View figures into Figure EV1 etc. throughout the manuscript.
6. Please upload the EV figures as individual production quality figure files in the .eps, .tif, or .jpg format (one file per figure).
7. Please rename "Summary" into "Abstract".
8. There are references to Datasets 1-3 in the manuscript. Please upload them with the legends included in the corresponding files and names following the nomenclature Dataset EV1-3.
9. In the Data Availability section, please add a resolvable link to the dataset PRJNA1232226. More information about the format of this section can be found here: <https://link.springer.com/partners/embo-press/editorial-policies#Data%20availability%20statement>.
10. CRediT has replaced the traditional author contributions section because it offers a systematic, machine-readable author contributions format that allows for more effective research assessment. Please remove the Author Contributions from the manuscript and use the free text boxes beneath each contributing author's name in our online submission system to add specific details on the author's contribution. More information is available in our guide to authors.
11. Please rename "Declaration of interests" section into "Disclosure and competing interests statement" (further info: <https://www.embopress.org/page/journal/14602075/authorguide#conflictsofinterest>).
12. Please move "Acknowledgements" and "Disclosure and competing interests statement" sections after "Methods".
13. Please update references according to The EMBO Journal style - it should be alphabetical. Where there are more than 10 authors on a paper, the first 10 should be listed, followed by 'et al.'
14. Please download and fill our Reagents and Tools Table template (.docx), which you can find in our author guidelines: <https://media.springernature.com/original/springer-cms/rest/v1/content/27825802/data/v1>
The information currently provided in Tables EV1 and EV2 should be included in the Reagents and Tools Table. When submitting your revised manuscript, please do not include the Reagents and Tools Table in the Methods section of the manuscript but upload it as a separate file choosing the file type "Reagent Table".
15. During our standard image check, we noticed possible figure panel reuse between the following figures:
 - Figure 2A and Figure 3D
 - Figure 3F and Figure S7A
 - Figure 4B and Figure S7B
 - Figure S6B and Figure S7A
 - Figure S7A and Figure S7BPlease check and correct if needed. If this was intentional, please clearly indicate the reuse in the figure legends.
16. Our data editors have flagged the following issues in figure legends that need correcting:
 - Please provide the exact p-values in the legends of figures 1G, 2D, 3B, G; 4D, E; 5D, 6D, E.
 - Please indicate the statistical test used for data analysis in the legends of figures 1C, 4D.
 - Please provide the information on the number and nature of replicates in the legends of figures 1C, 2D, 4D, 5C.
 - Please define the error bars in the legends of figures 2D, 4D, 5C.
17. Papers published in The EMBO Journal are accompanied online by a 'Synopsis' to enhance discoverability of the manuscript. It consists of A) a short (1-2 sentences) summary of the findings and their significance, B) 3-4 bullet points highlighting key results and C) a synopsis image that is 550x300-600 pixels large (width x height, jpeg or png format). You can either show a model or key data in the synopsis image. Please note that the image size is rather small and that text needs to be readable at the final size. Please send us this information together with the revised manuscript.

Please feel free to contact me if have any questions regarding this final revision. Thank you again for giving us the chance to consider your manuscript for The EMBO Journal. I look forward to receiving the revised version.

With best wishes,

Ieva

Ieva Gailite, PhD
Senior Scientific Editor
The EMBO Journal
Meyerhofstrasse 1
D-69117 Heidelberg
Germany
Tel: +4962218891309
i.gailite@embojournal.org

We realize that it is difficult to revise to a specific deadline. In the interest of protecting the conceptual advance provided by the work, we recommend a revision within 3 months (16th Apr 2026). Please discuss the revision progress ahead of this time with the editor if you require more time to complete the revisions.

Referee #1:

This is the manuscript I reviewed earlier this year (Reviewer #1). The authors have adequately addressed my major concerns, including: (i) the mechanism by which Dap2 inhibits ExsA and Lon; (ii) the use of an animal model to validate the biological impact of Dap2 on PaoP5 phage therapy; and (iii) evidence that Lon proteolytic activity is broadly inhibited by Dap2. Overall, the revised manuscript is in good condition, and I recommend it for publication, subject to the following two minor suggestions.

Quantitative analysis of the protein expression and stability of the Dap2 mutants (V54K, A64K, N79A, the triple mutant VAN/KKA, E97A, and E99A) would be important to demonstrate that these variants are expressed at comparable levels in cells. This would help exclude the possibility that the observed phenotypes arise from differences in protein abundance or stability rather than from altered functional activity.

In Figure 4f, the Dap1 bands in the first and third panels are not consistently indicated by the brown arrows. As a result, it is difficult to unambiguously identify the Dap1 protein bands. Please revise the figure to ensure consistent and clear annotation across all panels.

Referee #2:

The authors have undertaken extensive experimental efforts to address all reviewer critiques. They were able to achieve genetic separation of the Lon-inhibiting and ExsA-inhibiting activities of Dap2, and provided data indicating that T3SS inhibition occurs in the context of phage infection. I now concur with the authors' assertion that the manuscript at hand achieves sufficient novelty relative to their previous publication. I do have a remaining minor concern regarding Fig. 5. Other than that, I think the paper is suitable for publication.

The main issue is with the newly added Fig 5c (one-step growth curve). I suspect something may have been mislabeled or perhaps miscalculated. The curves as labeled do not agree with the calculated phage burst sizes shown in 5d. The curves in 5c all show approximately a 100-fold burst size, with the uncomplemented $\Delta\text{dap1}\Delta\text{dap2}$ mutant exhibiting the strongest increase in PFU, while the burst size calculation in 5d makes it seem like this phage does not replicate at all. Please explain / resolve this issue.

I may be mistaken but do believe there is a typo in Fig. 5b. Panel 3 should be labeled "PaoP5 $\Delta\text{dap1}\Delta\text{dap2}$ + PAO1/p-dap2"

Fig. 5c I imagine y-axis should read "log₁₀ PFU/ml"

The title of Fig. 5 should probably be changed to "Dap2 and Dap1 cooperate to defend against Lon protease"

Referee #1:

This is the manuscript I reviewed earlier this year (Reviewer #1). The authors have adequately addressed my major concerns, including: (i) the mechanism by which Dap2 inhibits ExsA and Lon; (ii) the use of an animal model to validate the biological impact of Dap2 on PaoP5 phage therapy; and (iii) evidence that Lon proteolytic activity is broadly inhibited by Dap2. Overall, the revised manuscript is in good condition, and I recommend it for publication, subject to the following two minor suggestions.

Response: Thank you very much for your positive comments!

Quantitative analysis of the protein expression and stability of the Dap2 mutants (V54K, A64K, N79A, the triple mutant VAN/KKA, E97A, and E99A) would be important to demonstrate that these variants are expressed at comparable levels in cells. This would help exclude the possibility that the observed phenotypes arise from differences in protein abundance or stability rather than from altered functional activity.

Response: Thanks. To directly test whether the point mutations affect protein stability and expression, we expressed and purified all Dap2 mutants (V54K, A64K, N79A, V54K/A64K/N79A, E97A, E99A) alongside the wild-type protein in *E. coli* BL21(DE3). SDS-PAGE analysis of the purified proteins confirmed that all point mutants are expressed.

In Figure 4f, the Dap1 bands in the first and third panels are not consistently indicated by the brown arrows. As a result, it is difficult to unambiguously identify the Dap1 protein bands. Please revise the figure to ensure consistent and clear annotation across all panels.

Response: Thanks. We have now marked the protein positions more accurately and clearly.

Referee #2:

The authors have undertaken extensive experimental efforts to address all reviewer critiques. They were able to achieve genetic separation of the Lon-inhibiting and ExsA-inhibiting activities of Dap2, and provided data indicating that T3SS inhibition occurs in the context of phage infection. I now concur with the authors' assertion that the manuscript at hand achieves sufficient novelty relative to their previous publication. I do have a remaining minor concern regarding Fig. 5. Other than that, I think the paper is suitable for publication.

Response: Thank you very much for your positive comments!

The main issue is with the newly added Fig 5c (one-step growth curve). I suspect something may have been mislabeled or perhaps miscalculated. The curves as labeled do not agree with the calculated phage burst sizes shown in 5d. The curves in 5c all show approximately a 100-fold burst size, with the uncomplemented $\Delta\text{dap1}\Delta\text{dap2}$ mutant exhibiting the strongest increase in PFU, while the burst size calculation in 5d makes it seem like this phage does not replicate at all. Please explain / resolve this issue.

Response: Thank you for pointing this out. The labeling error in the one-step growth curve has been corrected.

I may be mistaken but do believe there is a typo in Fig. 5b. Panel 3 should be labeled "PaoP5 $\Delta dap1\Delta dap2$ + PAO1/p-dap2"

Response: Thanks. The labeling error has been corrected.

b

Fig. 5c I imagine y-axis should read "log10 PFU/ml"

Response: The labeling error has been corrected.

The title of Fig. 5 should probably be changed to "Dap2 and Dap1 cooperate to defend against Lon protease"

Response: Thank you for your suggestions, we made the corrections.

"Fig. 5. Dap2 and Dap1 cooperate to defend against Lon protease"

Dear Haihua,

Thank you for incorporating the final formatting requests in the manuscript. I am now pleased to inform you that your manuscript has been accepted for publication. Congratulations with a nice study!

Before we forward your manuscript to our publishers, we would like to propose some edits in the manuscript abstract and synopsis. We would also like to propose an alternative version of the title, with the goal to highlight the more general novelty of the findings. I have also written a short blurb that will accompany the title of your manuscript in our online system. Please take a look at the proposed text changes in the attached text file and let me know if any corrections are needed.

You may qualify for financial assistance for your publication charges - either via a Springer Nature fully open access agreement or an EMBO initiative. Check your eligibility: <https://link.springer.com/journal/44318/how-to-publish-with-us>

If you have any questions, please do not hesitate to contact the Editorial Office. Thank you for your contribution to The EMBO Journal!

Wishing you a happy Year of the Horse,

Ieva
